



# Deciphering the evolution of the Bleis Marscha rock glacier (Val d'Err, eastern Switzerland) with cosmogenic nuclide exposure dating, aerial image correlation, and finite-element modelling

Dominik Amschwand[1,3], Susan Ivy-Ochs[1,2], Marcel Frehner[1], Olivia Steinemann[2], Marcus Christl[2],
Christof Vockenhuber[2]

[1]Department of Earth Sciences, ETH Zurich, 8092, Zurich, Switzerland
[2]Laboratory of Ion Beam Physics, ETH Zurich, 8093, Zurich, Switzerland
[3]Now at: Department of Geosciences, University of Fribourg, 1700, Fribourg, Switzerland

*Correspondence to*: Dominik Amschwand (dominik.amschwand@unifr.ch) ORCID: 0000-0003-2179-1481

**Abstract.** We constrain the Holocene morphodynamic development of the Bleis Marscha rock glacier (Err-Julier area, eastern Swiss Alps) with fifteen cosmogenic nuclide exposure ages ([10]Be, [36]Cl), 2003/2012 horizontal surface creep rate quantification from orthophoto orientation correlation, and semi-quantitative ice-content estimates from finite-element modelling. The results suggest that the complex Bleis Marscha rock glacier formed during two activity phases, one in the early Holocene and one in the late Holocene, separated by a mid-Holocene period of inactivation. The now transitional-inactive low-elevation lobes (first generation) formed after the retreat of the Egesen cirque glacier in a pulse-like manner at 11.5–9.0 ka. Rock-glacier viscosities inverted with the finite-element model hint at ground ice in these lobes which is possibly as old as its early-Holocene debris cover. In contrast to the debris-conditioned rapid emplacement, the thermally controlled permafrost degradation is still ongoing, likely attenuated by thermal decoupling from the insulating coarse-debris boulder mantle. Nuclide loss from boulder erosion, affecting the nuclide inventory of boulders independently, led to a heterogeneous exposure age distribution on the transitional-inactive lobes. Exposure ages on such disturbed lobes record time elapsed since inactivation and are interpreted as (minimum) stabilisation ages. The inception of the active high-elevation lobes (second generation) at 2.8 ka is related to the late-Holocene cooling recorded at numerous sites across the Alps. Precise exposure ages of the last 1.2 ka correlate with down-stream distance and yield a long-term average surface speed coincident with 2003/2012 measurements. These long-term consistent surface creep rates indicate stable permafrost conditions and continuous rock-glacier growth despite the intermittent late-Holocene glacier cover of the Bleis Marscha cirque. The exposure ages on active, undisturbed lobes record time elapsed since boulder emergence at the rock-glacier root and are interpreted as travel time estimates. This work contributes to deciphering the past to quasi-present climate sensitivity of rock glaciers.



# 1 Introduction

In our current warming climate (Hock et al., 2019), active rock glaciers as the "visible expression of mountain permafrost" (Barsch, 1996) receive considerable attention. Their surface kinematics is considered as diagnostic for the thermal state of mountain permafrost (Delaloye et al., 2018), which is otherwise not directly observable. Rock glaciers are thought to store significant water resources as ground ice (Jones et al., 2019) and become more significant in the deglaciating mountains (Haeberli et al., 2017; Knight et al., 2019). However, in the literature, contradicting views on the climate sensitivity of rock

glaciers are proposed.

One concept is that rock glaciers respond attenuated and delayed to current warming because of the high thermal inertia of the ice-rich core and the thermal decoupling from external climate by the insulating effect of the boulder mantle (active layer) (e.g. Haeberli et al., 2017; Anderson et al., 2018) via the 'thermal semi-conductor' effect (Harris and Pedersen, 1998; Humlum, 1998; Hanson and Hoelzle, 2004; Guodong et al., 2007). The ground cooling effect of a coarse-debris mantle

(Schneider et al., 2012; Wicky and Hauck, 2017) favours a large negative thermal offset and permafrost conditions even at mean annual ground temperature (MAGT) close to 0 °C (Kellerer-Pirklbauer, 2019). Furthermore, the creep of millennia-old rock glaciers is tied to the ice-supersaturation of the debris and hence to the preservation of ground ice over their entire lifetime (Barsch, 1996; Haeberli et al., 2003).

Another concept is the synchronous, rapid response to warming, based on kinematic rock-glacier monitoring. "Almost all

rock glaciers across the Alps" show a common behaviour of surface creep rates with (sub-)seasonal fluctuations (Delaloye et al., 2010). These decennial to annual changes in surface creep rates respond within months to changing summer air temperature and snow cover timing (Noetzli et al., 2019). Debris pulses or "surge packages" (Kenner et al., 2014), as well as "significant acceleration", destabilisation (Marcer et al., 2019) up to "sudden collapse" (Bodin et al., 2016) are reported. Rock glacier formation can occur within centuries (Humlum, 1996), or under very specific topo-climatic conditions even

within decades (Scotti et al., 2017).

Active rock glaciers are defined as "lobate or tongue-shaped bodies of perennially frozen unconsolidated material supersaturated with interstitial ice and ice lenses that move downslope or downvalley by creep as a consequence of the deformation of ice contained in them and which are, thus, features of cohesive flow" (Barsch, 1996). Their active phase and development are conditioned by ground ice preservation, permafrost conditions (Haeberli et al., 2006), and debris supply

(Kenner and Magnusson, 2017). There are two paths to inactivity: (i) climatic inactivation and (ii) dynamic inactivation. In climatic inactivation ground ice melts out until its content falls below a critical saturation threshold where cohesive creep is no longer possible. The movement comes to a halt. This type of inactivation is a direct consequence of warming and a rising permafrost belt. In dynamic inactivation rock wall weathering rate and debris/ice incorporation becomes insufficient to sustain shear stresses required for the advancement, regardless of the rock-glacier thermal state and ground ice preservation.





Ultimately, when ground ice is lost completely, the rock glacier deposit settles, although maintaining its diagnostic micro-topography. These former rock glaciers are defined as relict (Barsch, 1996).

The response of rock glaciers to external forcings such as air temperature, precipitation and snow cover, weathering intensity, debris supply, and interactions with glaciers are insufficiently understood. Furthermore, their external response is difficult to disentangle from internal thermo-mechanical and topographic feedbacks. Historical records are too short

compared to typical rock glacier lifetimes, activity phases and response periods. Long-term effects on rock glacier development must remain unresolved (Kenner and Magnusson, 2017). To put the present-day morphology reflecting the lifelong dynamic history of active rock glaciers and (relict) rock glacier deposits (Frauenfelder and Kääb, 2000) in a climate-sensitivity context, their activity phases need to be placed in a chronological framework. Cosmogenic radionuclides exposure dating is a unique tool because it directly measures a chronometric, numerical exposure age of the landform surface (Ivy-

Ochs and Kober, 2008). The technique has been applied on (relict) rock glaciers deposits or related periglacial landforms by Ivy-Ochs et al. (2009), Böhlert (2011a, b), Moran et al. (2016), Denn et al. (2017, and references therein), and Steinemann et al. (2020). Relict and active rock-glacier deposits in Iceland were exposure dated by Férnandez-Férnandez et al. (2020).

In this study, our focus is on the Bleis Marscha rock glacier located in the Err-Julier area, eastern Swiss Alps (Fig. 1). It is a 1100 m long, tongue-shaped rock glacier at an elevation range of 2400–2700 m a.s.l. Previous studies used relative dating

techniques (Frauenfelder et al., 2001; 2005; Laustela et al., 2003). We exposure date 15 boulders along a longitudinal transect from the lowermost front up to the transition towards the talus with the cosmogenic radionuclides $^{10}$Be and $^{36}$Cl. This is the first study that exposure dates boulders on an active, presently moving rock glacier lobe in the Alps. The exposure ages are interpreted in light of field observations, present surface creep quantification through image correlation, and numerical finite-element (FE) modelling to unveil periods of activity and the morphodynamic development of the Bleis

Marscha rock glacier. This work contributes to deciphering the past to quasi-present climate sensitivity of rock glaciers in a high-mountain environment.

## 2 Study site

The studied Bleis Marscha rock glacier (WGS 84: 46°34′18″N, 9°42′10″E; CH1903 / LV03: 773553, 160325) lies on the eastern slope of the upper Val d'Err, a side valley of the Surses (Oberhalbstein) in the Err-Julier area, Grisons, eastern Swiss

Alps (Fig. 1). The rock glacier originates in a NNW-facing cirque of Piz Bleis Marscha (3127 m a.s.l.).

The Err-Julier area lies in the rain shadow of the Lepontine, Bernese and Glarus Alps resulting in a dry-cold, continental-type climate. Frauenfelder et al. (2001) report low mean annual precipitation (MAP) of 900–1000 mm a$^{-1}$ (1971–1990), a regional lapse rate of 0.55°C/100 m and a mean annual 0 °C-isotherm at c. 2180 m a.s.l. Val d'Err is a high valley with the valley floor at an elevation above 2000 m a.s.l. and surrounding peaks rising to over 3000 m a.s.l. The present-day lower

limit of permafrost occurrence is roughly at 2400 m a.s.l. (Gruber et al., 2006; Boeckli et al., 2012).





The north-northwest to south-southeast oriented valley lies in a tectonically complex zone between the Upper Penninic Platta nappe overlain by the Lower Austroalpine Err nappe. The debris-supplying headwalls (in Tschirpen unit of the Err nappe) are composed of post-Variscan granitoids and Permo-Triassic to Lower Cretaceous sediments (mostly slate and carbonates), separated by a thrust outcropping subhorizontally along the cirque walls (Cornelius, 1935, 1950; Frauenfelder et al., 2005).

During the Last Glacial Maximum (LGM) around 24 ka (Ivy-Ochs, 2015; Wirsig et al., 2016), the area was covered up to an elevation of ~2800 m a.s.l. by ice flowing northwards from the nearby Engadine ice dome (Inn River catchment) into the Rhine glacier system (Bini et al., 2009). Prominent presumed Lateglacial (Egesen) moraines along the valley flanks (Frauenfelder et al., 2001) suggest that the upper Val d'Err, including our study site, was likely last occupied by a glacier during the Egesen stadial (12.9–11.7 ka (Ivy-Ochs, 2015)). The Bleis Marscha cirque was occupied by a glacieret during the

Little Ice Age (LIA) and up until recently (Dufour, 1853; Frauenfelder et al., 2001).

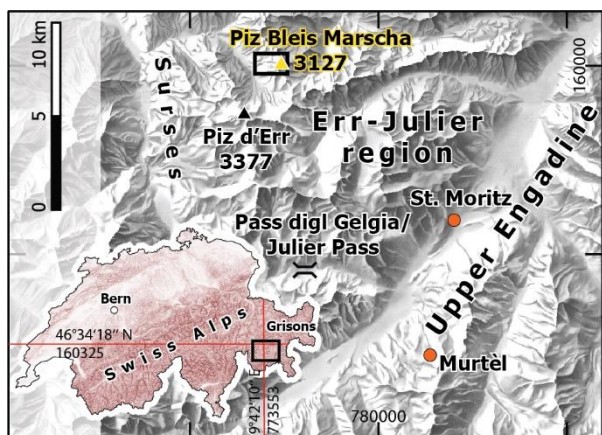

**Figure 1: Location of the Bleis Marscha rock glacier to the west below Piz Bleis Marscha in the Err-Julier region, eastern Swiss Alps. The rectangle shows the area covered in Fig. 3. Inset map: Location and extent (black rectangle) of the Err-Julier regional map within Switzerland and the coordinates of the lowermost front of the Bleis Marscha rock glacier (Maps reproduced with the authorization of the Swiss Federal Office of Topography swisstopo).**

**3 Material and Methods**

Topographic maps, aerial photographs, digital terrain models, geomorphological field mapping, surface-exposure dating, and FEM were combined to reconstruct the development of the Bleis Marscha rock glacier.

**3.1 Field work and landform analysis**

Fieldwork including mapping at a scale of 1:5000 and sampling was carried out in August 2017. Orthorectified aerial images
(0.25×0.25 m resolution) and the high-resolution digital terrain model (DTM) "swissALTI3D" provided by the Swiss Federal Office of Topography (swisstopo) served for topographic analysis and visualisation. The DTM, acquired in 2016, is gridded on 2×2 m cells (resolution) and has an average error of ± 1–3 m (1σ level accuracy) for areas above 2000 m a.s.l.



Extraction of swath profiles and morphometric calculations were carried out with the Matlab toolbox TopoToolbox 2 (Schwanghart and Scherler, 2014) and the open-source software QGIS.

## 3.2 $^{10}$Be and $^{36}$Cl exposure dating

The numerical ages of 15 boulders of the rock-glacier deposit were determined with the surface exposure dating method using the cosmogenic radionuclides $^{10}$Be and $^{36}$Cl.

Ideally, the sampled rock surface has undergone undisturbed, single-stage (no pre-exposure or inheritance), continuous exposure (no covering) in the same position (not shifted or toppled) since deposition (Ivy-Ochs and Kober, 2008). Suitable boulders are large (>1.5 m) and in a stable position; suitable rock surfaces do not show signs of fast weathering or spalling. Boulders at the rock glacier surface suitable for exposure dating were sampled with hammer, chisel, and battery-powered saw according to guidelines of Ivy-Ochs and Kober (2008) and field observations. The samples (~0.5 kg of rock material each) were collected along the central flow line from the lowermost front up to the high-elevation active lobes, preferentially on ridges to minimise topographic and snow shielding (Böhlert et al., 2011b) and towards the frontal upper edge of each morphologically identified lobe (Haeberli et al., 2003; Steinemann et al., 2020). 14 samples were Julier granodiorite (quartz, $^{10}$Be; Table 1); only one dolomite boulder for $^{36}$Cl met the sampling criteria (Table 2).

$^{10}$Be sample preparation followed Kronig et al. (2018). The ratio of $^{10}$Be/$^{9}$Be is measured with the 600 kV Tandy at the ETH Zürich Accelerator Mass Spectrometry (AMS) facility (Christl et al., 2013). The in-house standard S2007N, which is calibrated against the 07KNSTD was used. For $^{36}$Cl extraction from the dolomite sample (Err8), the method of isotope dilution was employed (Ivy-Ochs et al., 2004). Concentrations of major and trace elements were measured by ICP-MS (Inductively Coupled Plasma Mass Spectrometry) at Actlabs (Ontario, Canada) (Table 3). AMS measurements were conducted with the 6 MV Tandem accelerator (Synal et al., 1997; Vockenhuber et al., 2019) of the Laboratory of Ion Beam Physics, ETH Zurich.

The $^{10}$Be surface exposure ages were calculated from the blank-corrected data (long-time laboratory blank of $^{10}$Be/$^{9}$Be = (3.2 ± 1.4) × 10$^{-15}$) using the CRONUS-EARTH online calculator (Balco et al. 2008) with the North-eastern North America (NENA) $^{10}$Be production rate of 3.87 ± 0.19 atoms g$^{-1}$ a$^{-1}$ and the scaling model by Lal (1991)/Stone (2000). The NENA production rate has been shown to be well applicable for the Alpine area (Claude et al., 2014). The $^{36}$Cl surface exposure age was computed with an ETH Laboratory of Ion Beam Physics (LIP) in-house developed MATLAB program based on the equations and constants given in Alfimov and Ivy-Ochs (2009, and references therein). $^{36}$Cl production in dolomite is dominated by spallation of Ca, muon interactions with Ca and low-energy neutron capture reflecting the high natural Cl concentration (Err8: 49.6 ± 0.1 ppm, Table 2). The following production rates were used: 48.8 ± 3.4 $^{36}$Cl atoms g$^{-1}$ a$^{-1}$ for spallation in Ca and 5.3 ± 1.0 $^{36}$Cl atoms g$^{-1}$ a$^{-1}$ for muon capture in Ca at the rock surface. A neutron capture rate of 760



± 150 neutrons/$g_{air}$ (Alfimov and Ivy-Ochs, 2009) was implemented. We used the Lal/Stone scaling of the production rates to the site latitude, longitude, and elevation (Balco et al., 2008).

The shielding parameters were calculated with the "online calculators formerly known as the CRONUS-Earth online calculators" (Balco et al., 2008, http://hess.ess.washington.edu/math). We report and discuss exposure ages with an erosion rate of 1 mm kyr$^{-1}$ for the crystalline samples ($^{10}$Be) and 5 mm kyr$^{-1}$ for the dolomite sample ($^{36}$Cl). Snow-cover corrections are omitted as such corrections would change the exposure ages by only a few percent. The reported errors are at the 1σ level including analytical uncertainties of the AMS measurements and the blank correction (internal errors).

**3.3 Orthophoto orientation correlation**

We quantify the horizontal surface creep rate with the cross-correlation of orientation images (Fitch et al., 2002) derived from bi-temporal ortho-images with the Matlab tool ImGRAFT (Messerli and Grinsted, 2015). The used orthophoto mosaic *swissimage 25 cm* is a composite of orthorectified digital color aerial photographs, provided by the Swiss Federal Office of Topography (swisstopo). Ground resolution is given as 0.25 m, positional accuracy as ± 0.25 m. We derived the orientation
images from the R band of the RGB images.

The orientation correlation method developed by Fitch et al. (2012) is a feature-based method of translatory image matching that matches the orientation of the image intensity gradients. Orientation images are normalized and invariant to pixel brightness, making the method more robust and less susceptible to different illumination in the images.

The post-processing steps are noise filtering to remove erroneous matches and smoothing to attenuate small-scale and thus
likely short-lived velocity variations. A minimum correlation coefficient of 0.6 and a conservative signal-to-noise ratio threshold $SRN_{min} = 6$ sufficed to remove incoherent and poor-quality displacement vectors. The optimal template size of 51×51 pixels was found using a procedure after Debella-Gilo and Kääb (2012). The search windows size of 211×211 pixels was defined with the recommendations of Messerli and Grinsted (2015). We estimate the uncertainty by correlating a reference area in the valley floor considered as stable. The modal displacement defines the significance level, that is the
threshold below which any measured displacement is not distinguishable from immobility.

Streamlines depict the trajectory and travel time for a particle travelling at the rock glacier surface, if the underlying surface flow field remains unchanged during the entire travel period (Kääb, Haeberli and Gudmundsson, 1997). Streamlines interpolated up-stream from the sampled boulders yield theoretical steady-state travel time estimates. We numerically integrate the defining ordinary differential equation with a Runge-Kutta 4$^{th}$ order scheme. Since streamline interpolation is
noise sensitive, the input velocity field is additionally smoothed. Data gaps are interpolated with the Matlab tool "inpaint_nans" (D'Errico, 2012).

The horizontal surface deformation pattern is analysed by means of the strain-rate tensor (Kääb, Haeberli and Gudmundsson, 1997) defined as the symmetric part of the velocity gradient tensor. Velocity gradients are computed with the central finite





difference scheme. Short-wavelength noise that lead to errors larger than typical strain rate values are filtered out by an
additional circular averaging filter (low-pass) filter. The strain-rate orientation is visualised by the direction of the principal
strain rate axes, obtained via the diagonalization of the strain rate tensor. In this particular representation of the strain rate
tensor, the shear strain components vanish and the two non-zero components represent pure extension or contraction.

### 3.4 Finite-element modelling

The surface movement of a rock glacier integrates the overall vertical deformation profile (Arenson et al., 2016). An
appropriate flow law – a mathematical formulation of the governing deformation process – allows in principle to infer from
(known, observable) surface properties to (unknown) effective material properties and structures at depth that cause the
observed deformation.

The long-term deformation of rock glaciers is governed by gravity-driven steady-state creep of its ice-rich core (Arenson et
al., 2016). In case of ice-supersaturation, the deformable (excess) ice leads to stress transfer in space and time and therefore
to a coherent velocity field, a diagnostic feature of active rock glaciers. Creep of permafrost can therefore be approximately
described by Glen's flow law for polycrystalline ice (Haeberli et al, 2006), establishing a constitutive power-law relationship
between shear stresses $\tau$ [Pa] and shear strain rates $\dot{\varepsilon}$ [s$^{-1}$]. The surface speed $u_s$ [m s$^{-1}$] of an infinite, parallel-sided slab
using Glen's flow law is

$$u_s = \frac{2A}{n+1} (\rho g \sin \bar{\alpha})^n H^{n+1},  \tag{1}$$

with flow rate factor $A$ [Pa$^{-n}$ s$^{-1}$] related to dynamic viscosity $\mu := \frac{\tau}{2\dot{\varepsilon}} = (2A\tau^{n-1})^{-1}$ [Pa s], stress exponent $n$ [-], density $\rho$
[kg m$^{-3}$], average surface slope $\bar{\alpha}$ [°], thickness $H$ [m], and gravitational acceleration $g$ [9.81 m s$^{-2}$]. However, Eq. 1 is
underdetermined, and no unique solution exists. A simultaneous determination of material properties (density, viscosity) and
structure (thickness) is not possible. This inverse problem requires regularisation to become solvable, i.e. all but one of these
parameters need to be estimated independently.

The rock-glacier model is mechanically described by the model parameters $H$, $\bar{\alpha}$, $\rho$, and $\mu$. We invert for effective dynamic
viscosity $\mu$, from known surface slope $\bar{\alpha}$, and surface speed data $u_s$. We regularise the inverse problem by a-priori
prescribing density $\rho$, and thickness $H$, parameters that can be reasonably well estimated from literature knowledge and field
observations (Fig. 2a). In absence of any Bleis Marscha rheological borehole data to constrain the power-law exponent $n$ of
the effective viscous flow law, we proceed by a forward operator that implements the simplest linear viscous (Newtonian)
material ($n = 1$). The system of force-balance and isothermal constitutive equations are solved numerically in two
dimensions with the numerical FE code presented in Frehner et al. (2015).

Direct evidence from over-steepened rock glacier fronts, borehole deformation measurements (Arenson et al., 2002) and
indirect geophysical investigations (Springman et al., 2012) suggest a pronounced mechanical layering of rock glaciers. A
robust finding is a sequence of three main layers (Haeberli et al., 2006; Frehner et al, 2015). (1) The ice-free surface layer





consists of a matrix-poor, clast-supported framework of large, interlocked boulders. (2) Beneath the boulder mantle, the ice-rich permafrost layer consists of a frozen mixture of ice, debris, and fine material. This rock-glacier core accommodates the horizontal deformation, often concentrated in localized shear zones or décollements. (3) Boulder mantle and core lie on a stiff, immobile substratum of ice-poor, frozen debris, which does not contribute to the observed dynamic deformation of the rock glacier (Arenson et al., 2002). Field observations suggest that the general rock-glacier stratigraphy is valid for the

studied Bleis Marscha rock glacier (Sect. 4.1). We approximate the deforming, dynamic part of the rock glacier as a three-layer system consisting of a 5 m thick boulder mantle (constant along profile), ~30–50 m thick core (variable) and a 3 m thick basal low-viscosity shear zone (constant). The (variable) cumulative thickness is estimated as follows: The model top boundary is the rock-glacier surface given by the DTM, implemented as a free surface (Fig. 2b). The bottom boundary is defined by a fixed, no-slip boundary to the immobile substratum. Its elevation is projected from the adjacent terrain at the

rock-glacier front where the rock glacier rises from the Salteras terrace (Fig. 2a; cf. Kääb and Reichmuth, 2005; Scapozza et al., 2014) and parallel to the average rock-glacier surface in the upper stretches. All layers including the ice-free boulder mantle effectively obey a viscous flow law and are separated by no-slip boundaries (dynamically coupled, "welded") (Arenson et al., 2002; Springman et al., 2012; Frehner et al., 2015).

We estimate the density of the rock glacier materials as the weighted average of the density of its constituents, namely debris

(Err granodiorite, $\rho_{debris} = 2700$ kg m$^{-3}$), ice ($\rho_{ice} = 910$ kg m$^{-3}$), and air ($\rho_{air} = 1$ kg m$^{-3}$) (Eq. 10 in Müller et al., 2016). For the approximately void-free rock-glacier core with 60 vol% ice, we obtain $\rho_c = 1626$ kg m$^{-3}$, and for the ice-free boulder mantle with 30 vol% air $\rho_m = 1890$ kg m$^{-3}$ (Barsch, 1996; Fig. 2b). The effect of water on density is insignificant. The prescribed viscosity ratios between the different layers are: rock-glacier core to high-viscosity boulder mantle 1:20 (estimated from dominant wavelength of furrow-and-ridge microtopography, cf. Sect. 4.1), and core to low-viscosity basal

shear layer 10:1 (conservative estimate; Fig. 2b).

The susceptibility for steady-state creep of debris-ice mixtures depends on debris-ice proportions (specifically the degree of ice supersaturation, i.e. the ice volume exceeding the pore volume), fabric, particle size, temperature and water content (Moore, 2014). Since the influence of each parameter is difficult to disentangle and the material properties and composition are not known at this level of detail, we absorb these contributions by the effective viscosity, $\mu$, and estimate it by solving the

inverse problem. We depart from an initial rock-glacier model with a uniform viscosity of pure ice for the entire rock-glacier core, $\mu_c^0 = 2 \times 10^{13}$ Pa s as an initial guess. Next, we compute synthetic surface velocity data by means of the discretized forward operator, carried out by the numerical FE code (Fig. 2c). The synthetic data predicted by the current model is visually compared to the measured data. For rock-glacier parts where the misfit between synthetic/predicted and measured data exceeds the data uncertainty, the viscosity is either increased if synthetic velocities were too high compared to measured

values ($d_{pred} > d_{meas}$) or decreased in the opposite case ($d_{pred} < d_{meas}$, estimation problem). The obtained plausible rock-glacier viscosity distribution, $\tilde{m}$, that explains the measured surface velocities, $d_{meas}$, is one solution to the inverse problem.



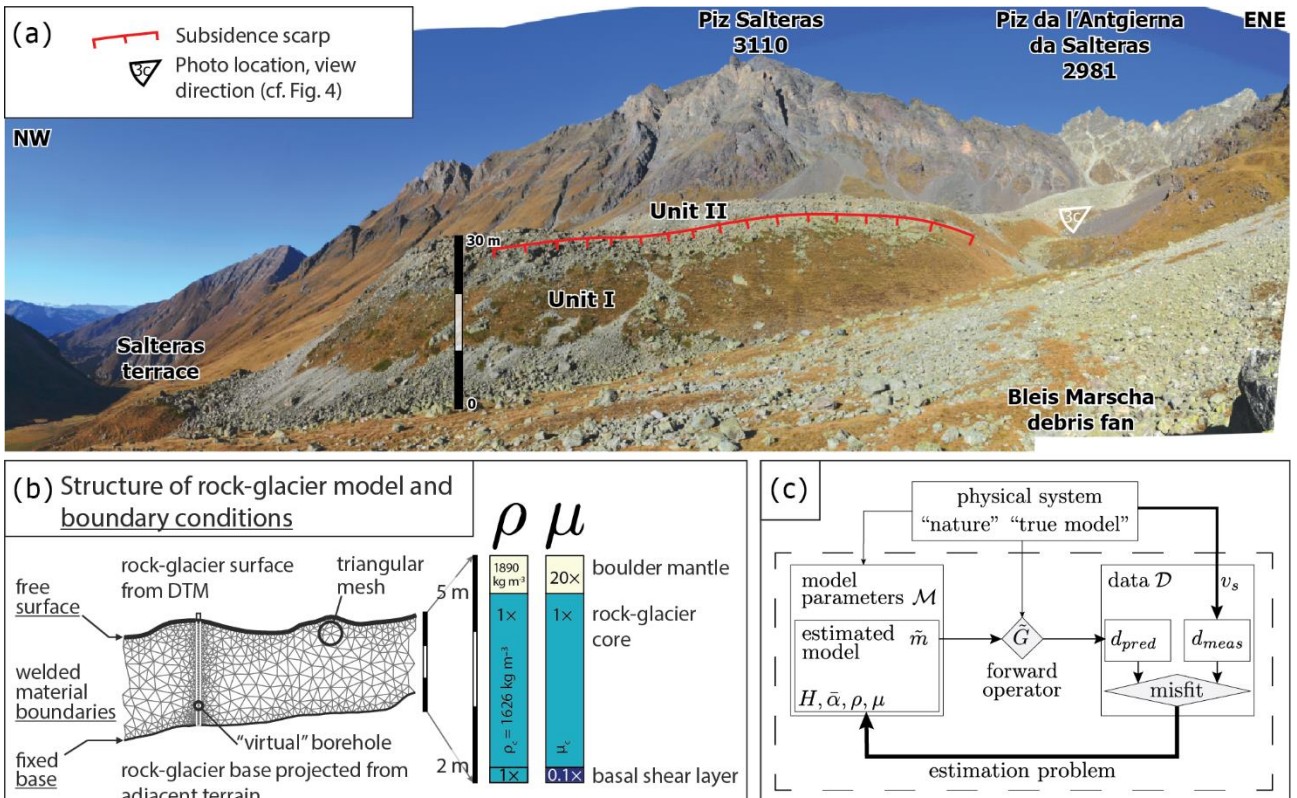

**Figure 2: (a) Side view of the lowermost, transitional-inactive front, rising 30–50 m above the Salteras terrace (cf. Fig. 4 for photo location and view direction). Part of the southern front is subsiding, possibly due to ground ice loss. The vertical throw of the subsidence scarp is 2–3 m. (b) Finite-element (FE) model setup, triangular mesh (Shewchuk, 1996), boundary conditions, and prescribed ratios of density ρ and dynamic viscosity μ (profiles). The surface boundary is the well-constrained elevation along a longitudinal profile, extracted from the DTM as a swath profile with TopoToolbox. The a-priori prescribed density and viscosity ratios create the characteristic mechanical layering of rock glaciers with a stiff, high-viscosity boulder mantle, deformable core, and basal weak, low-viscosity layer. (c) Inverse problem structure with the model parameters that are linked to the observable data via the forward operator $\widetilde{G}$, the FE model. The model predicts a synthetic surface velocity from the mechanical parameters that control the rock-glacier creep: thickness H, average surface slope $\bar{\alpha}$, density ρ, and dynamic viscosity μ.**

## 4 Results and interpretation: Geomorphology, age, and kinematics of the Bleis Marscha rock glacier

### 4.1 Field observations

The Bleis Marscha rock glacier is a tongue-shaped multi-unit talus rock glacier at an elevation range of 2400–2700 m a.s.l. (Fig. 3). With a length of 1100 m, a width of 150–200 m, a surface area of ~$2.4 \times 10^5$ m², and a source area of ~$3.1 \times 10^5$ m², it ranks among the largest rock glaciers in the Err-Julier area. We estimate a total volume of ~$(7–10) \times 10^6$ m³. Internal steep frontal scarps (slope angle >30°) and other morphological indicators separate the complex rock glacier into different units sensu Barsch (1996), each apparently with its own activity phase and status. We subdivide the Bleis Marscha rock glacier



into a two-part sequence comprised of two active upper lobes (units IV–V; 2550–2700 m a.s.l.) that override three transitional to inactive lower lobes (units I–III; 2400–2550 m a.s.l.; Figs. 3, 4).

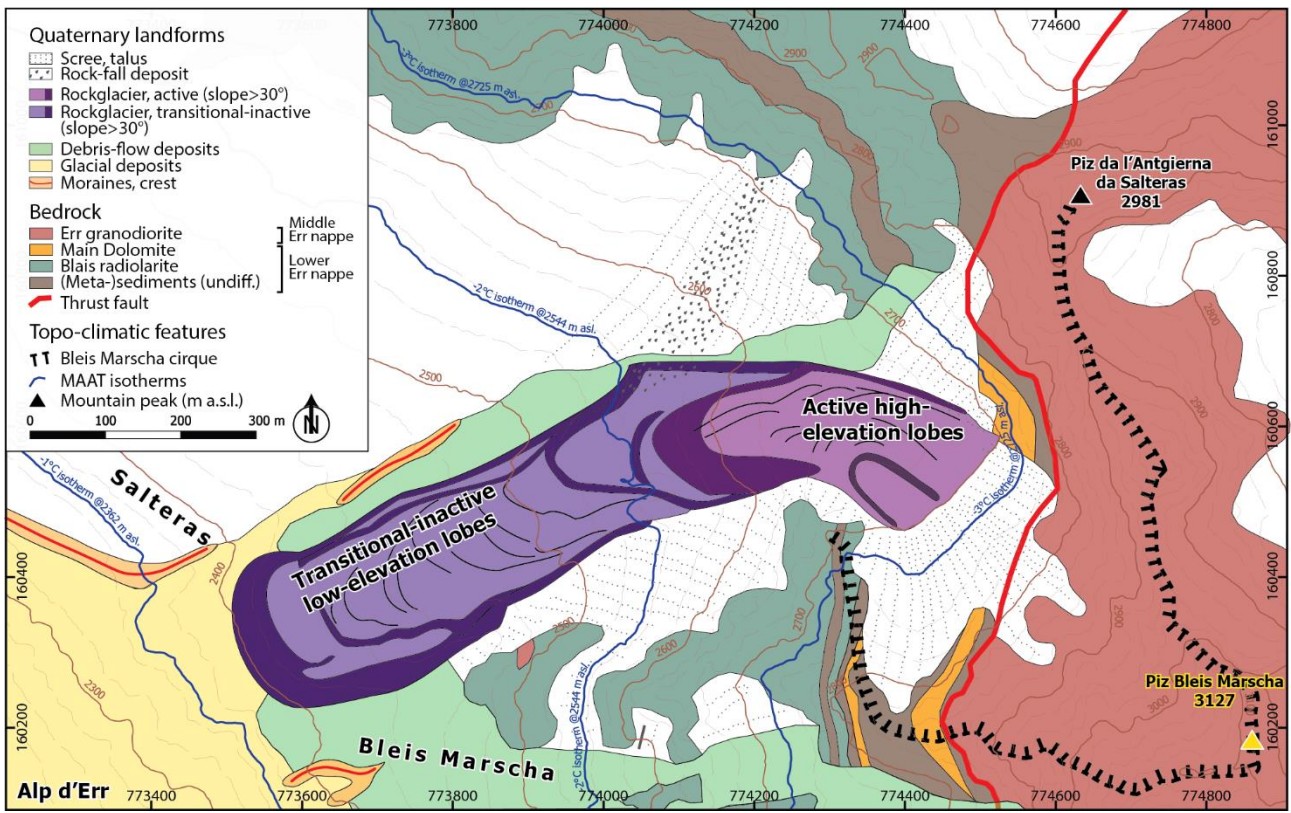

**Figure 3: Quaternary features and main lithologies in the Bleis Marscha cirque. Bedrock map modified from Cornelius (1935, 1950). A subhorizontal thrust separates the crystalline Middle Err unit from the sedimentary Lower Err unit. Present-day MAAT isotherms are drawn according to Frauenfelder et al. (2001).**

The rock glacier boulder mantle consists of three main rock types, each with its well-localised source in the cirque headwall (Fig. 3, Cornelius (1935, 1950)): prevalently weathering-resistant Err granodiorite, followed by Blais radiolarite and 260 tectonically fractured Main dolomite. The angular blocks have typical volumes of 0.5–50 m³, exceptionally with >10 m edge length. The dominance of Err granodiorite (>80 vol%) increases towards the lowermost front. Therefore, the lower lobes were originally connected to the talus and must extend upslope beneath the now-active lobes. Weathering-prone lithologies such as rauhwacke (cellular dolomite), conglomerate, limestone and shales occur only in lenses or thin seams in the headwall and are only rarely found on the rock-glacier surface. The rock glacier coarse-debris transport system is downstream 265 disconnected from the laterally adjacent terrain.

Despite the very recent glaciation of the cirque (1853 Dufour, 1887 Siegfried maps: Dufour, 1853; Siegfried, 1887), we did not observe signs of glacier activity nor any surface ice. The small ice patch mentioned by Frauenfelder et al. (2005) had completely disappeared by 2017. The smooth, unstructured talus passes directly into the broad (180 m) rock glacier root



zone (unit IV) at an elevation of 2700 m a.s.l. (Fig. 5a). No geomorphological evidence of the LIA glacier remains. A small
rock glacier lobe (unit V, width 80 m) is well visible in the root zone emanating from the talus (Fig. 4). In the bend at the lip
of the cirque, the micro-topography becomes more accentuated with a set of parallel longitudinal, asymmetric ridges (steep
side facing outwards), whose amplitude however rarely exceeds 2 m. This lobe rests on an older, apparently inactive body
(unit III, Fig. 5a) that displays incipient soil development, vegetation patches and a most likely rock fall-derived radiolarite-
debris covered protrusion (Fig. 5a). The steep (>20°) terrain is cut by three major scarps where the two-layered stratigraphy
of a coarse, clast-supported blocky mantle (thickness of 3–5 m) over a finer, matrix-supported core is exposed. These inner
scarps are expressions of faults and form the frontal boundary of units III and IV (Fig. 5b). The right-lateral boulder apron
along unit III between the rock-fall deposits (protrusion) and its front formed by debris slides exposing fresh material
suggests recent movement despite morphological signs of inactivity (Fig. 5c).

Unit II (the one being overridden by unit III) appears distinctly older and inactive. The lobe is marked by signs of settling
and weathering such as vegetation patches, lichen-covered boulders, iron staining, weathering rinds and a frontal apron. The
surface is gently inclined (10°), with a well-developed furrow-and-ridge micro-topography of crescent-shaped transverse
ridges separated by <5 m deep furrows (Fig. 5b). The lobe rests on an older, inactive to possibly relict body (unit I), recessed
by ~5 m and forming a ledge along the sides (Fig. 5c). The transitional-inactive lowermost lobes (units I–III) are bordered by
a stable outer ridge which is densely vegetated and has a thick soil cover (Figs. 5b, c). It extends upslope until the body
disappears beneath the younger unit IV (Fig. 2a). We interpret this grassy ridge as the margin of the oldest, possibly relict
body (unit I), and not as lateral moraines (Frauenfelder et al., 2001; cf. Barsch 1996). The front of unit I, mantled by a fall-
sorted boulder apron, rises roughly 30–50 m from the edge of the Salteras terrace at an elevation of 2380 m a.s.l. (Fig. 2a).
This lowermost front of the Bleis Marscha rock glacier has come to a halt just at the edge of the trough shoulder with a
(Egesen?) moraine ridge. The moraines and the rock glacier are nowhere in contact (Fig. 3).

A WSW-ENE running scarp, indicated by exposed fresh material, dissects both units I and II and is therefore a younger
feature (Figs. 4, 2a). A part of the lowermost lobes (300 m × 30 m) is slowly collapsing and subsiding southwards,
perpendicular to the former creep direction. No bulges, debris slides or debris aprons are observed, i.e. the deformation must
be slow and distributed. The body is sagging instead of laterally advancing, which hints at ground ice loss.

The mean surface slope and presumably also the basal slope drops from 20° to 10° at an elevation of 2500 m a.s.l., causing
the lower lobes to decelerate and to buttress against the upper lobes. The break in slope induced along-flow compressive
stresses (Fig. 5b). Buckle folding of the layered rock glacier (boulder mantle, ice-rich core) in response to compressive flow
likely was the dominant formation mechanism of the transverse furrow-and-ridge micro-topography in the lower part (cf.
Frehner et al., 2015). We use an analytical formula for the effective viscosity ratio $R$ between the rock glacier mantle and
core (equation modified for one-sided support from Biot (1961)):




$$R := \frac{\mu_m}{\mu_c} = \frac{3}{8}\left[\frac{\lambda_d}{\pi h_m}\right]^3.$$ (2)

The viscosity ratio $R$ can be calculated using the dominant wavelength $\lambda_d$ and rock glacier mantle thickness $h_m$. Inserting $\lambda_d$ = 54 m obtained via Fourier analysis and $h_m$ = 4 to 5 m, we estimate $R$ = 30 to 15.



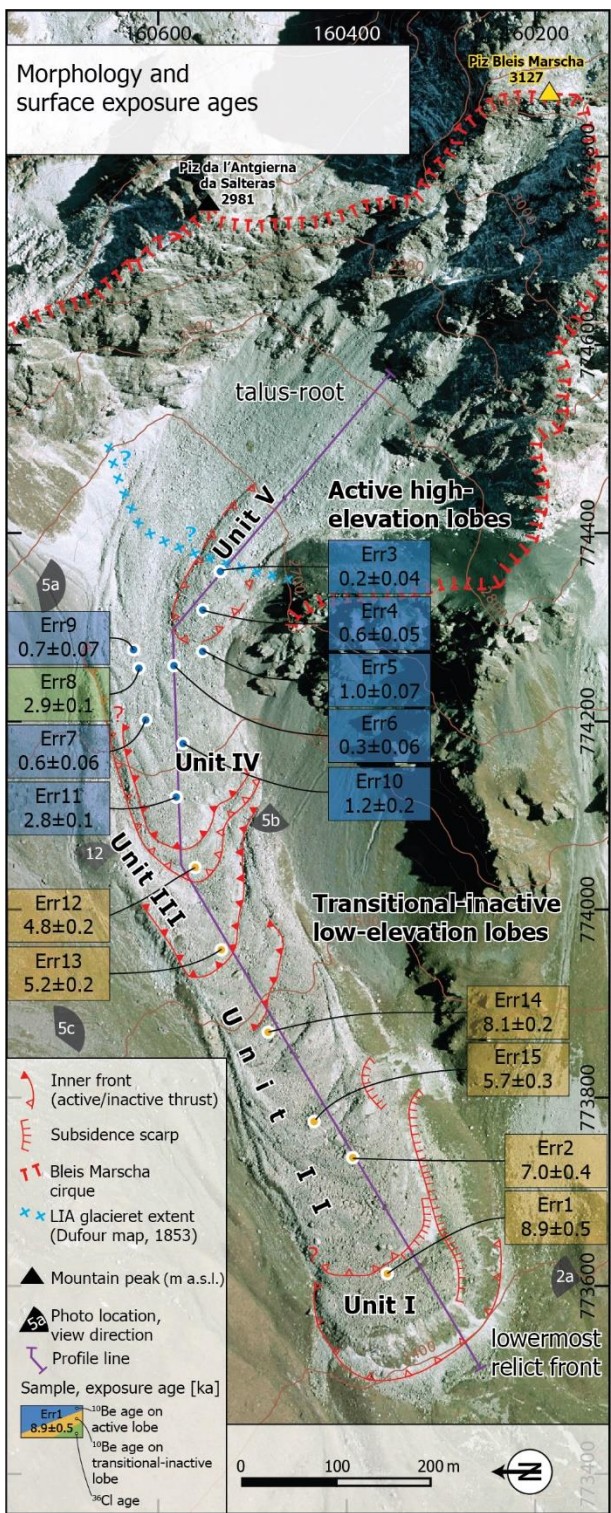

**Figure 4:** Plan view of morphological domains on 2003 orthophoto showing the morphological discontinuities (inner fronts, scarps, slope breaks), sampling locations and exposure ages (uncertainties are internal errors; cf. Fig. 7 for kinematic domains). Units I–III are transitional-inactive lobes overridden by the active units IV–V. Historical maps suggest that the Holocene maximum glacier extent reached during the Little Ice Age (LIA) remained within the Bleis Marscha cirque. (Orthophoto reproduced with the authorization of the Swiss Federal Office of Topography swisstopo).



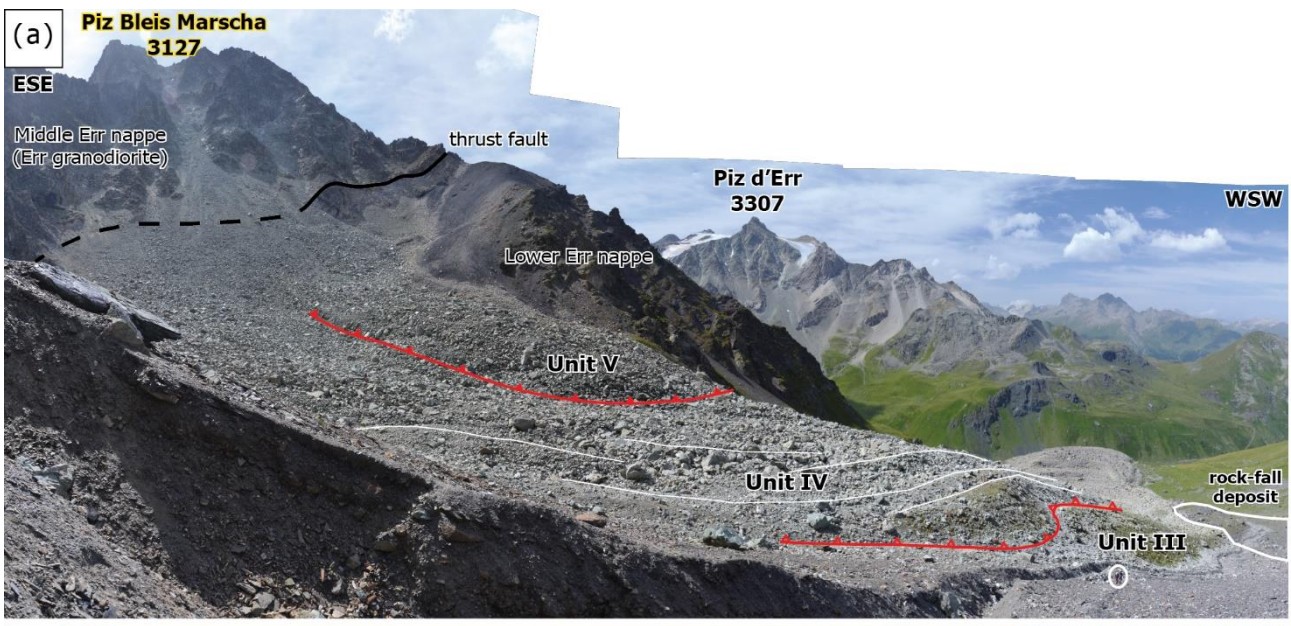

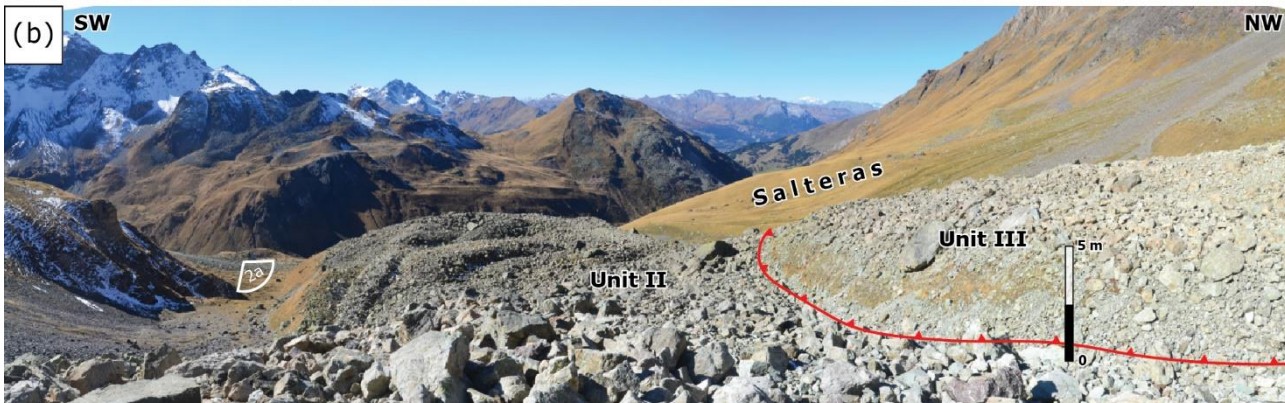

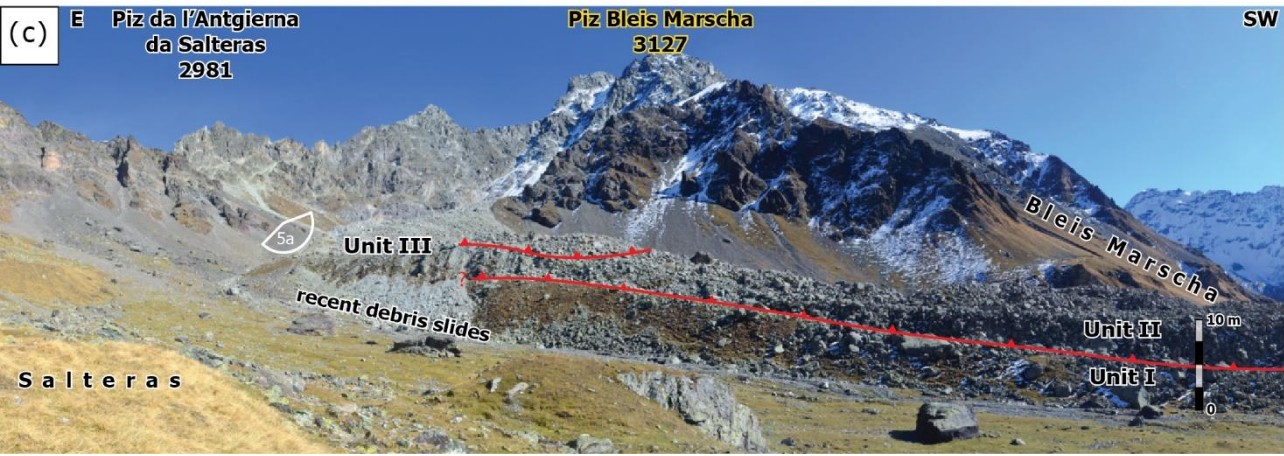



**Figure 5: Panorama images of the Bleis Marscha rock glacier (cf. Fig. 4 for photo locations). (a) View to SE into the root zone in Piz Bleis Marscha cirque. Most of the rock-glacier material is sourced in the back of the cirque from the crystalline Middle Err nappe (Err granodiorite outcropping above thrust fault). Note the debris-filled cirque with the prograding talus slope (transport-limited system) and the sudden terrain drop-off just below the cirque lip. (b) View to W onto lower rock-glacier part with furrow-and-ridge micro-topography (unit II). The currently advancing unit III is outlined by the prominent inner scarp. (c) Side view to SE of the middle rock-glacier part, evidencing the Bleis Marscha rock glacier as a multi-unit debris stream composed of multiple stacked lobes. The fresh lateral debris apron along the orographic right side of unit III hints at ongoing activity and likely recent reactivation. Unit I at the base of the stack is discernible by the vegetated, stable outer ridge and the ledge beneath unit II. The different coloring and composition of the autumnal vegetation is related to the substrate lithology: While the brownish grass in the foreground (moraine ridge, likely Egesen stadium of Bleis Marscha cirque glacier) grows on Err granodiorite, the depression between the rock glacier and the parallel-running moraine is filled with down-washed fines of mostly Blais radiolarite, schists and carbonates (higher moisture retention capacity).**

## 4.2 Boulder exposure ages

The measured $^{10}$Be concentrations are in the range of $(0.7 \pm 0.1$ to $23.6 \pm 1.2) \times 10^4$ at g$^{-1}$, the $^{36}$Cl concentration is $(3.1 \pm 0.1) \times 10^5$ at g$^{-1}$ (Tables 1, 2). The cosmogenic nuclide exposure ages ($^{10}$Be and $^{36}$Cl) range from $0.23 \pm 0.04$ ka to $8.95 \pm 0.47$ ka (Figs. 4, 6; Tables 1, 2). Two distinct age populations are distinguished: (i) a late Holocene group with precise ages ranging from 0.2 to ~1.2 ka located on the high-elevation active rock glacier lobes (samples Err3, 4, 5, 6, 7, 9, 10), and (ii) an early-to-middle Holocene group with dispersed (overlapping $\pm1\sigma$ range) ages ranging from ~4.8 to ~9.5 ka on the lower inactive lobes (samples Err1, 2, 12, 13, 14, 15). The $^{10}$Be and $^{36}$Cl ages of Err11 and Err8, respectively, defy this first-order classification.

The exposure ages in general anticorrelate with elevation and correlate with down-flow distance (Fig. 6). A linear regression between exposure age and distance from the rock glacier root is fitted to the samples on the presently active, high-elevation lobes (units IV–V). As all boulders were sourced in the root zone, the inverse slope can be interpreted as a mean transport rate (cf. Denn et al., 2017), yielding a plausible long-term surface speed of $v_s \approx 30$ cm a$^{-1}$. The time-axis intercept (Fig. 6) coincides with the transition to the talus. The boulders remained exposed at the surface and were passively transported at the rock glacier surface between talus and top of front slope (Frauenfelder et al., 2005; Scapozza et al., 2014; Winkler and Lambiel, 2018). We conclude that the exposure ages on the active lobe have negligible systematic errors and no overall age shift from inheritance/pre-exposure (systematically "too old") or nuclide loss/incomplete exposure (systematically "too young"). This implies that the transit time in the talus is not recorded by the exposure ages, possibly because it is shorter than the analytical uncertainty and the time "lost" due to snow shielding (not more than decades), or because the boulders were covered in the talus and surfaced only on the rock glacier.

An exception is the $^{36}$Cl age (Err8) that is roughly 2 ka older compared to its neighbouring $^{10}$Be ages (Fig. 4). As shielding by snow would make the age even older, the simplest explanation is that Err8 was pre-exposed in the rock wall prior to falling onto the talus and incorporation into the rock glacier. This age is an outlier and is not discussed further.

On the lower, transitional-inactive lobes (units I–III), the $^{10}$Be ages of Err2 and Err15 are significantly younger than the further up-slope sampled Err14. It is unlikely that this apparent age inversion reflects internal creep, as the laminar flow





behaviour of a rock glacier makes mixing impossible and contradicts our finding of linear relationship between exposure age and down-flow distance on the active lobes. Therefore, the exposure-age scatter must be introduced by surface processes in the boulder mantle that affect the nuclide inventory of boulders selectively (e.g. Böhlert et al., 2011a; Heyman et al., 2011). On these transitional-inactive lobes, the boulders might have rotated during settling and inactivation, resulting in too low nuclide concentrations and too young ages for boulders Err2 and Err15 ($7.03 \pm 0.37$ and $5.70 \pm 0.30$ ka, respectively). We

consider the ages of $8.08 \pm 0.24$ to $8.95 \pm 0.47$ ka (samples Err14, 1) as more representative for the low-elevation lobes.

**Table 1:** [10]Be sample properties and calculated exposure ages (Err granodiorite).

| Sample name | Latitude | Longitude | Elevation | Sample thickness | Shielding factor[a] | [10]Be concentration[b] | Exposure age[c] (erosion corrected) |
|---|---|---|---|---|---|---|---|
| | °N | °E | m a.s.l. | cm | - | $10^4$ at g$^{-1}$ | a |
| Err1 | 46.5719 | 9.7036 | 2448.7 | 1.5 | 0.9739 | $23.584 \pm 1.216$ | $8948 \pm 466 \ (636)$ |
| Err2 | 46.5722 | 9.7052 | 2481.5 | 1.5 | 0.9751 | $18.992 \pm 1.004$ | $7027 \pm 374 \ (505)$ |
| Err15 | 46.5726 | 9.7057 | 2485.3 | 1.0 | 0.9582 | $15.257 \pm 0.771$ | $5697 \pm 290 \ (399)$ |
| Err14 | 46.5730 | 9.7070 | 2498.3 | 2.0 | 0.9484 | $21.366 \pm 0.638$ | $8076 \pm 243 \ (460)$ |
| Err13 | 46.5734 | 9.7081 | 2529.3 | 2.0 | 0.9502 | $14.027 \pm 0.659$ | $5169 \pm 244 \ (349)$ |
| Err12 | 46.5737 | 9.7093 | 2551.7 | 1.5 | 0.9476 | $13.234 \pm 0.654$ | $4796 \pm 238 \ (332)$ |
| Err11 | 46.5738 | 9.7102 | 2584.9 | 1.5 | 0.9450 | $7.819 \pm 0.277$ | $2774 \pm 99 \ (166)$ |
| Err10 | 46.5737 | 9.7110 | 2611.7 | 1.5 | 0.9211 | $3.308 \pm 0.425$ | $1181 \pm 152 \ (162)$ |
| Err7 | 46.5741 | 9.7113 | 2623.2 | 1.5 | 0.9365 | $1.854 \pm 0.161$ | $646 \pm 56 \ (64)$ |
| Err6 | 46.5738 | 9.7121 | 2644.1 | 2.0 | 0.9406 | $0.860 \pm 0.183$ | $295 \pm 63 \ (64)$ |
| Err9 | 46.5742 | 9.7123 | 2647.8 | 1.5 | 0.9204 | $2.083 \pm 0.213$ | $727 \pm 74 \ (82)$ |
| Err5 | 46.5735 | 9.7123 | 2648.9 | 1.5 | 0.9428 | $2.940 \pm 0.202$ | $1001 \pm 69 \ (84)$ |
| Err4 | 46.5735 | 9.7128 | 2667.5 | 2.0 | 0.9392 | $1.642 \pm 0.154$ | $556 \pm 52 \ (59)$ |
| Err3 | 46.5733 | 9.7134 | 2679.8 | 1.8 | 0.9259 | $0.673 \pm 0.115$ | $229 \pm 39 \ (41)$ |

a Shielding correction includes the topographic shielding due to surrounding landscape and the dip of the sampled surface.

b AMS measurement errors are at 1σ level and include the AMS analytical uncertainties and the error of the subtracted blank. Measured ratios were calibrated to the 07KNSTD standard.

c Exposure age errors are internal errors (uncertainties represent 1σ confidence range comprising AMS counting errors and errors based on the normalization to blanks and standards). External errors are given in parentheses. Erosion correction for a surface erosion rate of 1 mm kyr$^{-1}$.





**Table 2:** [36]Cl sample properties and calculated exposure age (dolomite).

| Sample name | Latitude | Longitude | Elevation | Sample thickness | Shielding factor[a] | Cl in rock | [36]Cl concentration[a,b] | Exposure age[c] (erosion corrected) |
|---|---|---|---|---|---|---|---|---|
| | °N | °E | m a.s.l. | cm | - | ppm | $10^6$ at g$^{-1}$ | a |
| Err8 | 46.5741 | 9.7120 | 2642.6 | 1.5 | 0.937 | 49.63 ± 0.09 | 0.313 ± 0.014 | 2890 ± 130 (181) |

a Concentration measured against [36]Cl/Cl standard K382/4N (Christl et al., 2013; Vockenhuber et al., 2019).

b Sample ratio corrected for laboratory blank of $(2.5 ± 0.4) × 10^{-15}$ [36]Cl/[35]Cl.

c Production rates as in Alfimov and Ivy-Ochs (2009) and references therein. Erosion correction for a surface erosion rate of 5 mm kyr[1] (karst weathering/corrosion).

**Table 3:** Elemental composition of the dolomite sample Err8.

| $SiO_2$ | $Al_2O_3$ | $Fe_2O_3$ | MnO | MgO | CaO | $Na_2O$ | $K_2O$ | $TiO_2$ | $P_2O_5$ | LoI | Sm | Gd | Th | U |
|---|---|---|---|---|---|---|---|---|---|---|---|---|---|---|
| % | % | % | % | % | % | % | % | % | % | % | ppm | ppm | ppm | ppm |
| 0.45 | 0.35 | 0.26 | 0.06 | 20.39 | 31.88 | 0.06 | 0.10 | 0.01 | 0.01 | 46.12 | 0.20 | 0.10 | 0.20 | 6.70 |

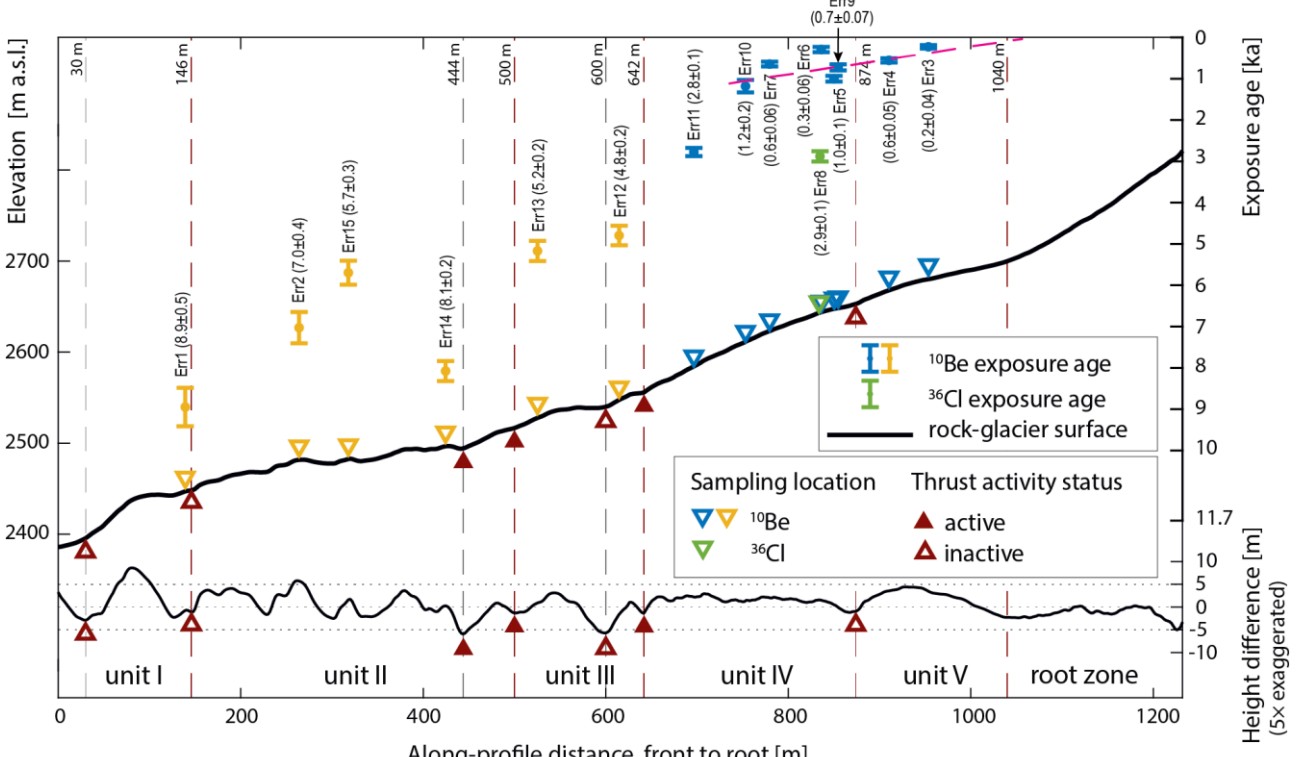

**Figure 6: Longitudinal section of rock-glacier surface, exposure ages and small-scale topography (trace shown in Figs. 4, 6). Note the "stratigraphically" directed time axis to emphasize the anti-correlation of the exposure ages with altitude. Exposure ages and sampling location projected onto the longitudinal profile. Linear regression (pink dashed line) of [10]Be samples Err3–7, 9, 10, that**



**lie along the central flow line, yield $t_{exp} = -3.28x + 3448$, $R^2 = 0.43$, whereas $x$ represents the down-profile distance. This regression line yields $t_{exp} = 0$ a at $x = 1050$ m, coinciding with talus–rock glacier transition. This suggests that pre-travel nuclide concentrations are negligible. The small-scale topography is a high-pass filtered topography computed by subtracting a 100 m running mean from the altitude at each point. Active thrusts coincide with sharp velocity gradients (cf. Fig. 7); this differential movement results in overriding lobes. Inactive thrusts are inner scarps without pronounced velocity changes.**

**4.3 2003/2012 surface creep rates**

Two orthophotos from late summer 2003 and 2012 are matched. The result is a noise-filtered horizontal surface velocity field and the orientation of the principal strain-rate axes for the Bleis Marscha rock glacier and its immediate surroundings (Figs. 7a, b). Typical speeds are $v_s \approx 30$ cm a$^{-1}$, with peak speeds up to 60 cm a$^{-1}$. There are a few data gaps where the image correlation failed due to decorrelation (non-translational movement, toppling, or vegetation), or missing data (snow fields).

The median or modal velocity of the non-moving, stable valley floor is a robust estimate for the magnitude of pre-processing errors (Fig. 8a). The significance level is 5.3 cm a$^{-1}$, i.e. speeds lower than this threshold are statistically non-significant and respective areas are classified as non-moving (pink areas in Fig. 7).

The kinematic data suggests the following subdivision of the Bleis Marscha rock glacier in four kinematic domains (front to root, boundaries along isotachs; cf. slope-velocity plot Fig. 8b for the elevation bands and the longitudinal velocity profile in

Fig. 9b):

• *Transitional-inactive front*, $v_s < 20$ cm a$^{-1}$ (elevation band 2400–2475 m a.s.l., units I–II lower part): The lowermost part is characterized by an irregular, "patchy" flow field both in terms of direction and magnitude at an overall low speed. Dominant strain type (shortening ~ compressional stress, stretching ~ extensional stress, slippage ~ shear stress) as well as the effective strain rate varies according to the small-scale topography, and the principal directions are not aligned with the

general WSW–NNO rock glacier orientation.

• *Transition zone*, 20 cm a$^{-1}$ < $v_s$ < 40 cm a$^{-1}$ (2475–2550 m a.s.l.; unit II upper part–unit III): In this part bounded by the 20 cm a$^{-1}$ isotach, a laterally clearly confined surface velocity field emerges. It is smooth with no sharp gradients except along the lateral margins. The speed gradually increases upslope to 40 cm a$^{-1}$, and the creep direction follows the large-scale topography. The strain is concentrated at the lateral margins, implying that the rock glacier body creeps en bloc. At the

margins, the principal axes are rotated by roughly 45° with respect to the general creep direction, typical for simple shear.

• *Rapid lobe*, 40 cm a$^{-1}$ < $v_s$ < 65 cm a$^{-1}$ (2550–2650 m a.s.l.; frontal/lower part of unit IV): The lower boundary is marked by a stark speed increase ($\Delta v_s = 10$ cm a$^{-1}$, data gap) and a prominent internal front scarp. Above an elevation of 2600 m a.s.l., the velocity decreases again. The velocity field remains smooth and aligned with the large-scale surface slope. Apart from the continuing lateral strain concentration, the deformation is characterized by strong compressive overriding of the lobes in

their frontal part and extensional flow behind. The principal axes are aligned with the general creep direction. The shortening is associated with speed decrease (in creep direction) or decelerating creep; stretching/elongation is associated with speed increase or accelerating creep.





- *Uppermost lobes*, 10 cm a$^{-1}$ < v$_s$ < 35 cm a$^{-1}$ (2650–2760 m a.s.l.; unit V and neighbouring upper part of unit IV): This lobe is individuated based on the slope-velocity relation (Fig. 8b) with a latero-frontal boundary following the morphology. The

defining character is the strongly fluctuating surface velocity that becomes gradually decoupled from the smoothed surface slope towards the talus. Strain rates appear random due to the velocity data gaps and decorrelation, likely due to independent, non-translational movement of single boulders. The boundary between root zone and talus is poorly constrained.

The velocity histogram (Fig. 8a) suggests that the discrimination of the Bleis Marscha activity status into low-elevation inactive versus high-elevation active lobes is at the 20 cm a$^{-1}$ isotach at an elevation of 2475 m a.s.l., implying that

(morphological) inactivity is not equivalent to (kinematic) immobility. Data points from the entire rock glacier surface drawn in a slope angle–surface velocity scatter plot (Fig. 8b, grey dots) are clustered in a wedge-shaped domain; the few outliers are due to non-translational movement (e.g. sliding, tumbling) and not due to large-area cohesive creep. Higher velocities are reached at higher surface slope angles, in accordance with the concept of shear-stress driven creep. The maximum slope angle slightly decreases with increasing velocity. Possibly, steep slopes at high deformation rates cannot be sustained by the

viscous debris-ice mixture over long time periods but are smoothed out by diffusion. The slope-velocity correlation is clearer when only a 20 m wide stripe along the central profile is considered, unaffected by boundary effects such as lateral drag and non-translational movements. The above defined kinematic domains have their distinct pattern. (i) The velocity of the lowermost, transitional-inactive part is low and independent of the 100 m averaged slope (yellow dots). (ii) The 'transition zone' (green) and 'rapid lobe' (dark blue) show a clear correlation between average surface slope and velocity and therefore

a strong large-scale topographic control on horizontal surface velocity, in agreement with our understanding of viscous creep. (iii) The correlation breaks down at the uppermost lobe towards the talus (light blue).

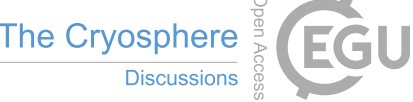

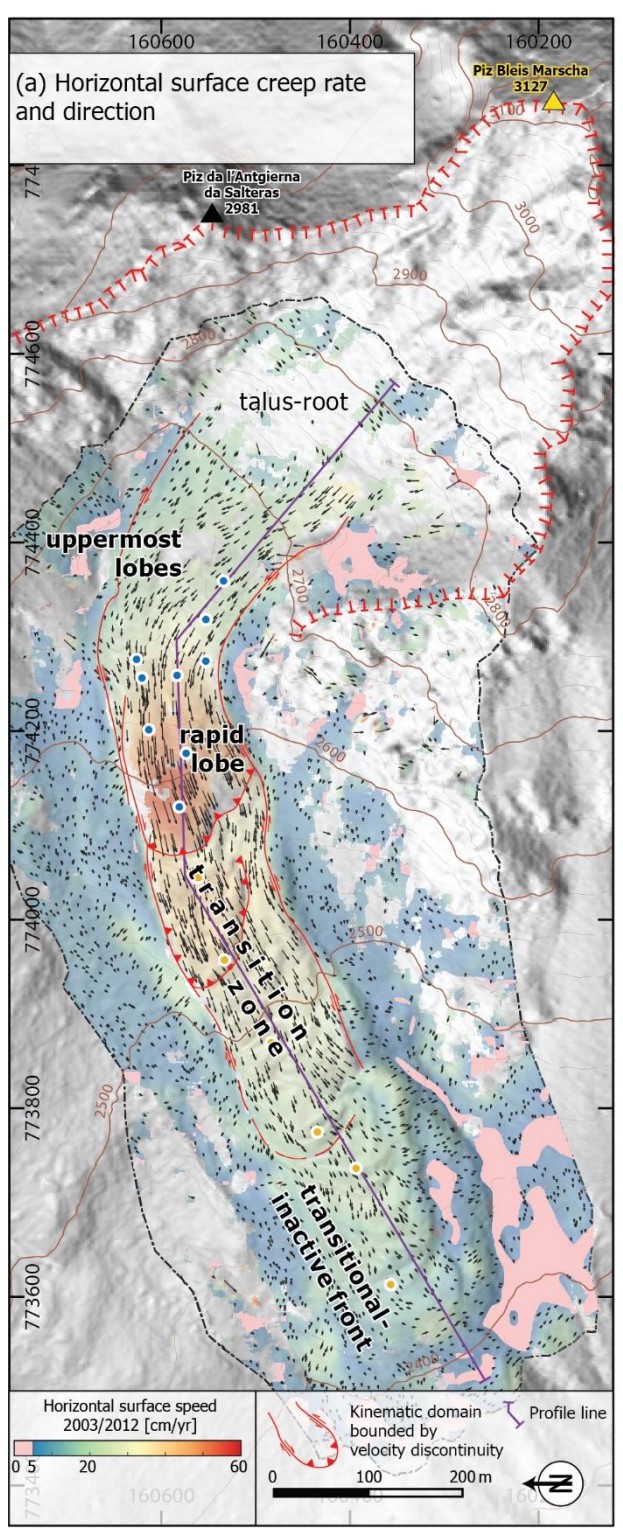

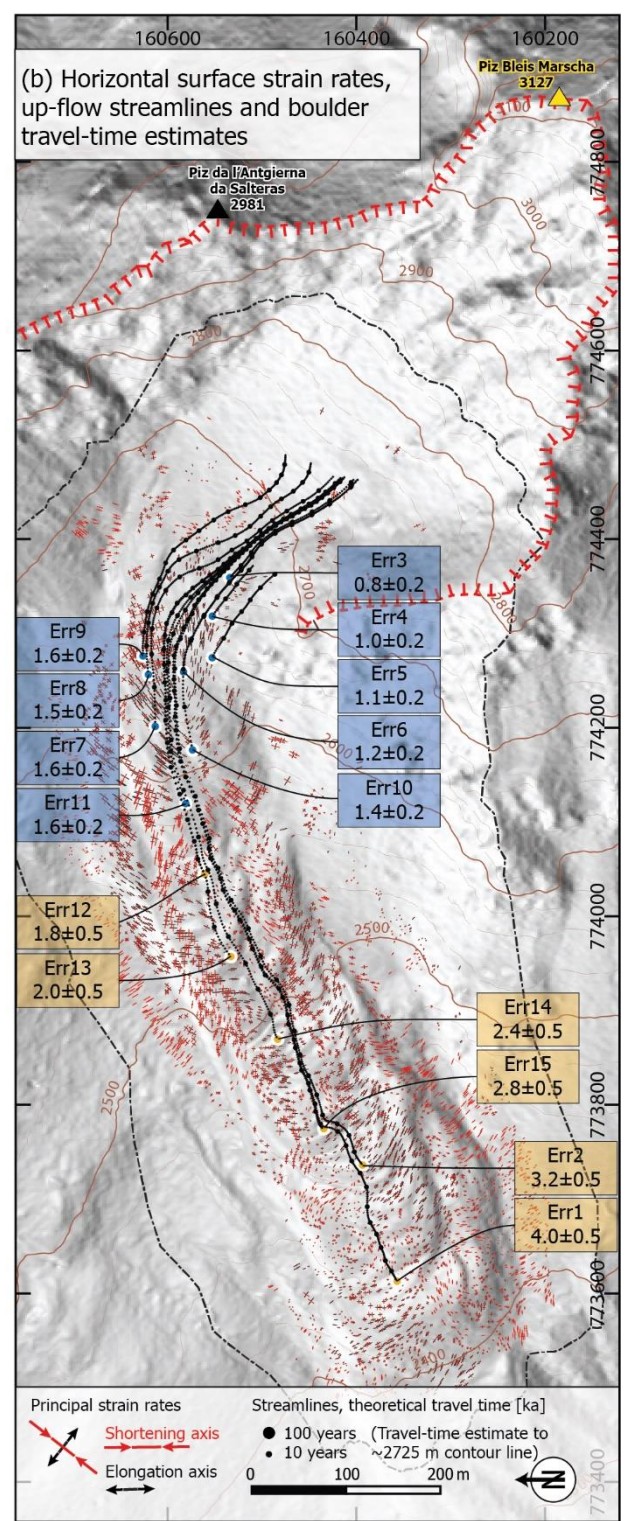





**Figure 7: (a) Plan view of kinematic domains separated by discontinuities such as decorrelation gaps and velocity jumps (cf. Fig. 4 for morphological domains). The noise-filtered horizontal surface velocity field of the Bleis Marscha rock glacier is derived from**
**two orthophotos from 2003 and 2012. Kinematic domains are labelled (cf. Fig. 9b). Significance level is 5 cm a⁻¹. The long temporal baseline of nine years averages inter-annual variabilities in rock glacier creep rates, and is a robust estimate of present-day, short-term surface kinematics. Morphological and kinematic discontinuities largely coincide. (b) Horizontal principal strain rates, computed with a spacing of 100 m (finite differences), show the direction of maximum shortening and elongation. The principle axes of strain rate and stress coincide. Up-flow streamlines from the sampled boulders to the ~2725 m contour line give a**
**theoretical trajectory and travel time estimate. Uncertainties of the estimated travel time $\Delta t_{trv}$ are estimated as 200 years for the active high-elevation lobes (blue rectangles) and 500 years for the transitional-inactive low-elevation lobes (yellow, cf. Fig. 10). (Hillshade background map reproduced with the authorization of the Swiss Federal Office of Topography swisstopo).**

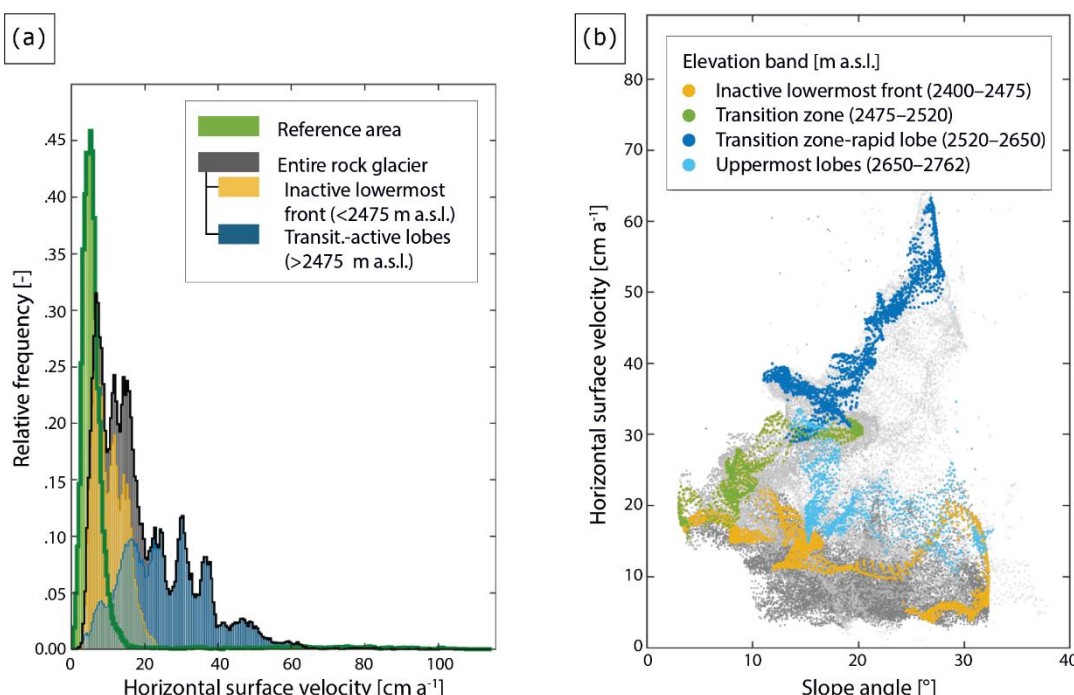

**Figure 8: (a) Velocity histogram: The histogram shows the relative occurrence frequency of surface velocities in the reference area (green) and on the Bleis Marscha rock glacier (dark grey). The rock glacier is further subdivided based along the 20 cm a⁻¹ isotach**
**in a lower transitional-inactive part (yellow) and an upper active part (blue). (b) Slope–velocity relation: Scatter plot showing the correlation between the 100 m averaged slope (from smoothed DTM) and the horizontal surface velocity for different kinematic domains of the Bleis Marscha rock glacier. The grey dots are data points on the entire rock glacier (lower part in dark grey/inactive <20 cm a⁻¹, upper active part in light grey). In color are the data points on a 20 m stripe along the central profile, where the effect of lateral drag is minimal.**

**4.4 Semi-qualitative ground-ice content estimates from mechanical modelling**

For a given rock glacier thickness, layering, and density, the effective viscosity distribution is fitted to the 2003/2012 surface velocity field. We obtain a simplified, but plausible rock-glacier structure (Fig. 9a) that reproduces the velocity data within their uncertainty (Fig. 9b). The result suggests that the viscosity in the apparently transitional-inactive low-elevation lobes are only slightly higher than in the active high-elevation lobes (Fig. 9a). The model fails to reproduce the observed velocity

jumps between individual lobes (profile at 420 and 620 m) and the differential movement of lobes (Fig. 9b). Vertical profiles





of the modelled horizontal displacement ("virtual boreholes") mimic borehole inclinometer measurements of other rock glaciers (Arenson et al., 2002) (Fig. 9a, insets). No boreholes have been drilled on Bleis Marscha rock glacier.

Rock glaciers are composed of a mixture of debris and ice, each material with a very different mechanical behaviour. However, the continuum approximation averages material parameters in space. Non-viscous processes and interactions
between constituents are neglected. Temperature-dependent viscosity, the effect of liquid water, processes at the debris-ice interfaces or grain-to-grain frictional contacts are not captured with the chosen isothermal linear-viscous rheology (Arenson et al., 2002; Moore, 2014; Frehner et al., 2015). Instead, we aim for a rough estimate of multi-annual, largely unchanging material properties. Short-term (daily to seasonal) processes like oscillating temperature or highly variable water flow leading to seasonal deformation variations are neither captured by the numerical model nor expressed in the 9 year averaged
surface velocity data set. The effective viscosity is a scale-dependent, homogenized, effective material parameter valid for the bulk material on a decametre-multiyear scale. The only deformation-relevant parameter that hardly changes over a decade must be related to ground ice content because of the high thermal inertia of the ice-rich body. We can therefore treat the inferred effective viscosity as a proxy for ground ice content. Since we have no data on Bleis Marscha subsurface ice content with which we could calibrate the inferred viscosity, we cannot give absolute estimates. Instead, we use the
modelling findings to compare the inferred relative ice content in different rock-glacier parts.

We used the FE code presented in Frehner et al. (2015). The aspects discussed therein (incompressibility, boundary effects) apply analogously in this study. To describe rock glacier creep mathematically, complex rheological constitutive relationships such as viscous power-law relationships have commonly been adopted. However, the roughly parabolic shape of horizontal deformation profiles as measured in several boreholes (Arenson et al., 2002) suggests that the deformation
above the shear zone is well represented by a Newtonian flow law (Frehner et al., 2015). In fact, our synthetic vertical deformation profiles ("virtual boreholes" in Fig. 9a, insets) resemble deformation measurements measured on other rock glaciers. The simple Newtonian model with vertical viscosity layering (low-viscosity basal layer) emulates more complex, non-linear deformation mechanisms as observed in rock glaciers.

The inferred rock glacier structure $\tilde{m}$ differs from the (unknown) true structure because of (i) non-uniqueness/ambiguity and
(ii) error propagation (Snieder and Trampert, 1999). First, non-uniqueness means that (possibly infinitely) many models explain the data equally well, even though the forward operator (Stokes flow) is unique. There is a fundamental ambiguity attached to the physical forward problem formulation. For example, higher surface speed can equally well be achieved with either greater thickness, steeper surface slope, or lower viscosity. However, the driving stress and ultimately the deformation is largely governed by the well-constrained average surface slope (not by the basal slope (Nye, 1952)) and the reasonably
assessable rock-glacier thickness and density. This is corroborated by the good correlation between surface velocity and averaged surface slope in the sections where viscous creep dominates (Fig. 9b). The remaining unknown viscosity can be estimated within an acceptable uncertainty. Second, both the observed data and the forward operator are contaminated with

errors that are carried into the estimated model. The inferred structure is drawn towards a rock-glacier state for the years 2003/2012 because the model is fitted to data from this measurement period. Due to the long 9 year baseline, an unusual

behaviour of a single year is not carried into the model (e.g. the warm summer 2003 with exceptionally high creep rates (Delaloye et al., 2010)).



**Figure 9: (a) Inferred effective viscosity distribution on the longitudinal transect (trace shown in Figs. 4, 6). With this viscosity distribution, the predicted surface velocity (Fig. 9b) of this estimated model, $\tilde{m}$, match the measured one within the data uncertainty: $\tilde{m}$ is one solution to the inverse problem. Insets: Modelled yearly horizontal displacements in "virtual boreholes". The**
**strain localisation near the base despite the linear stress–strain rate relationship arises from the rock-glacier parameterisation with the three-part mechanical layering (boulder mantle, core, basal shear layer). The simple Newtonian model with depth-varying viscosity generates synthetic velocity profiles that resemble borehole deformations measured on other rock glaciers. (b) Longitudinal section of observed (blue line), modelled/synthetic (orange) horizontal surface velocity and the 100 m filtered slope (yellow). Surface velocity on sections dominated by viscous creep closely follow the average surface slope. Correlation is lost**
**at the stabilised lowermost front and in the talus. Kinematic domains are delimited by sharp speed gradients (dark red triangles, cf. Fig. 6).**





## 5 Discussion: Bleis Marscha landform history

By examining the exposure ages and considering the image correlation results we decipher the history of the Bleis Marscha rock glacier over the past 9 ka. As the regional climate history is reasonably known (Frauenfelder et al., 2001, 2005; Ivy-Ochs et al., 2009; Böhlert et al., 2011a), we attempt to disentangle internal thermo-mechanical/topographic feedbacks from external forcing such as climatic conditions, debris supply and interactions with glaciers.

The morphologic and kinematic domains coincide and separate the Bleis Marscha rock glacier into two zones of different age and activity status (Figs. 4, 7). Steep scarps outlining the lobes are expressed in the kinematic data as velocity jumps, data gaps and decorrelated areas. The active high-elevation lobes (2550–2700 m a.s.l.) with surface speeds up to 60 cm a$^{-1}$ display clustered exposure ages within $0.23 \pm 0.04$ to $2.8 \pm 0.1$ ka. The morphologically transitional-inactive low-elevation lobes (2400-2550 m a.s.l.) show surface speeds below 40 cm a$^{-1}$ and strongly dispersed exposure ages ranging from $4.8 \pm 0.2$ to $8.9 \pm 0.5$ ka.

### 5.1 Surface kinematics across time scales

In order to understand what the exposure ages represent, we compare the measured exposure age $t_{exp}$ with travel time $\Delta t_{trv}$ computed from streamlines of the present, short-term (2003/2012) surface velocity field (Fig. 10). Theoretically, boulder residence time on the rock glacier surface is the sum of the travel time (transportation during active period) and time since cessation of advance (transitional-inactive period) (Haeberli et al., 1998, 2003). Conceptually, boulders move along the 1:1 line during active periods where boulder surface residence time equals both exposure time (i.e. ideal, complete exposure) and travel time (i.e. representative velocity field, steady-creep conditions (Haeberli et al., 2003)). There are two main clusters in the travel time–exposure age space: (i) Below the line of equal time estimates ("1:1 line") group the high-elevation samples with young exposure ages $t_{exp}<1.2 \pm 0.2$ ka, and (ii) above the 1:1 line the low-elevation samples with older exposure ages.

$t_{exp} < \Delta t_{trv}$, below 1:1 line (Fig. 10): The exposure ages for the samples located on the presently active upper lobes are slightly less than the travel times by about 10–100 years, which is within the uncertainty margins. The measured surface velocity field is representative, and the discrepancy is smaller the closer the samples are to the central flow line (Err3, 4, 5, 10), where the influence of lateral viscous drag is smallest.

The exposure ages from the active lobe suggest a long-term averaged surface velocity of not more than 30 cm a$^{-1}$ over the last ~1000 years on the uppermost lobes, a value that is surprisingly consistent with the 2003/2012 short-term horizontal surface speeds (moderate v$_s$ ~ 30 cm a$^{-1}$, <65 cm a$^{-1}$). Topographic gradient, surface velocity and axes of principal strain rates are aligned on a 100 m scale (Fig. 7). Creep direction and magnitude are governed by the average surface slope. The consistency between present velocity field with long-term kinematics (in terms of average surface speed) and present 100 m scale topography (in terms of creep direction, speed, and strain rates) of the upper lobes indicates that the permafrost body is in equilibrium.





Our data on the active lobe of Bleis Marscha rock glacier show that the exposure ages for an undisturbed rock-glacier lobe can be addressed as (minimum) travel times. While travelling during the active phase, boulders remain mutually interlocked

within the clast-supported framework and the age dispersion remains small. Notably, it also shows that pre-travel nuclide build-up, inheritance, and pre-exposure in the rock wall or on the talus are insignificant.

$t_{exp} > \Delta t_{trv}$, above 1:1 line (Fig. 10): The exposure ages for the samples located on the presently transitional-inactive lower rock-glacier lobes considerably exceed the travel time by up to several thousand years. Although present-day surface velocity field is likely not representative for their active period, already the morphologically inactive appearance, exposure

age inversions, the large age dispersion exceeding analytical uncertainty, and the quotient of rock glacier length (1100 m) to typical surface speed (~30 cm a$^{-1}$) clearly indicates that the exposure ages (<8.9 ka) are much too old to be addressed as travel time estimates.

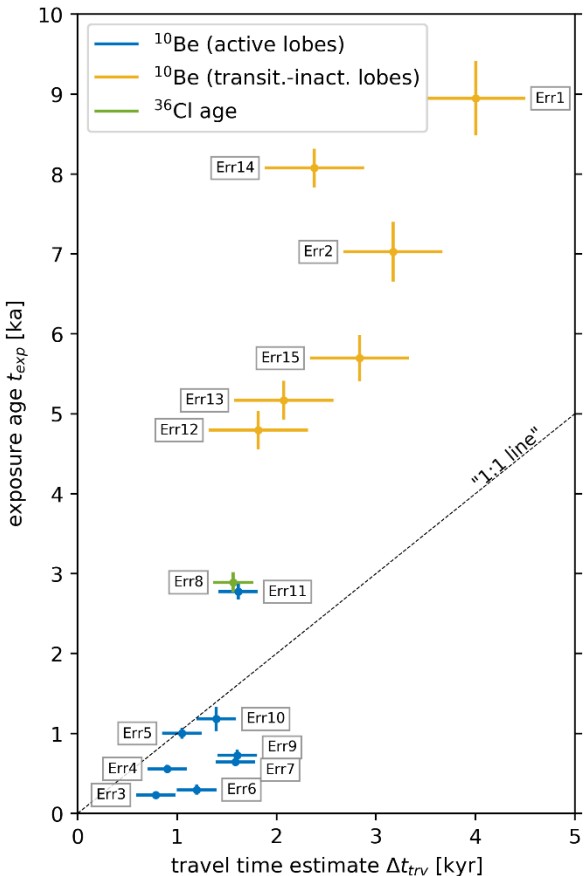

**Figure 10: Travel time estimate $\Delta t_{trv}$ versus exposure ages $t_{exp}$. By comparing the kinematic signal from long-term exposure ages and short-term image correlation results, we construct a framework on how to interpret the exposure ages. Travel time estimate**

**$\Delta t_{trv}$ for each sampled boulder is derived from the image analysis as the travel time between the current boulder position and the 2725 m a.s.l. contour line (Fig. 7b). Uncertainties of the exposure ages $t_{exp}$ are the analytical uncertainties (internal errors). Cf. Fig. 7b for travel time uncertainty estimates. In a hypothetical steady-state case with perfectly representative velocities and ideal**



**exposure ages, the boulders move along the line of equal time estimates ("1:1 line"). This "1:1 line" separates the boulder exposure ages on the active lobes (below) from the ages on the transitional-inactive lobes (above, long inactivity time). The exposure age of**
**Err8 is possibly "too high" because of nuclide inheritance. The travel time estimate of Err11 is likely affected by internal topographic feedbacks (kinematic wave and sudden terrain drop-off, cf. Fig. 5a).**

The surface deformation pattern on the Bleis Marscha rock glacier changes below an elevation of 2500 m a.s.l. (Fig. 7b). The transitional-inactive, slowly collapsing lower lobes are characterised by an irregular surface velocity field that is strongly coupled to the small-scale topography. Creep direction and magnitude are governed by the local surface slope, i.e. the ridges
settle and collapse. Intermittent boulder instability during inactivation has been affecting the nuclide inventory of each boulder individually, leading to incomplete exposure by self-shielding and nuclide loss by weathering or spalling, and ultimately to tendentially underestimated true exposure times ('apparent rejuvenation'). This results in a large dispersion of exposure ages and exposure age inversions on the low-elevation rock-glacier lobes.

Buckle folding in response to compressive flow of a layered medium likely was the dominant formation mechanism for the
transverse furrow-and-ridge micro-topography (cf. Frehner et al., 2015). However, the 2003/2012 strain-rate data do not show along-flow shortening and compression (Fig. 7b). Therefore, the observed micro-topography on the lower rock glacier lobes is an expression of palaeo-stress conditions that are different from the present-day stress field. The morphology memorises the cumulative deformation history over the lifetime of the rock glacier and is largely preserved during inactivation (Frauenfelder and Kääb, 2000).

The field above the 1:1 line is characteristic for rock-glacier lobes whose activity phase has ceased. A long time has passed since surface advection came to a halt, and the travel time is within the uncertainty of the exposure age. The exposure ages are rather inactivity or stabilisation ages than travel time estimates, as previously reported (Moran et al., 2016; Steinemann et al., 2020).

We find that exposure ages need to be interpreted according to the dynamic history of the sampled rock-glacier lobe. On
undisturbed lobes where the current surface creep is concordant with the micro-topography, exposure ages represent (minimum) travel times. This applies to active lobes without destabilisation. On 'discordant' lobes where micro-topography reflects different stress conditions and must have undergone disruptive dynamic changes, the exposure ages represent time elapsed since the major dynamic change. Inactivation or destabilisation are such disruptive events. With this interpretation scheme for exposure ages on active to relict rock-glacier lobes, we reconstruct the Bleis Marscha rock-glacier development
in three major phases, early, middle, and late Holocene (Fig. 11).



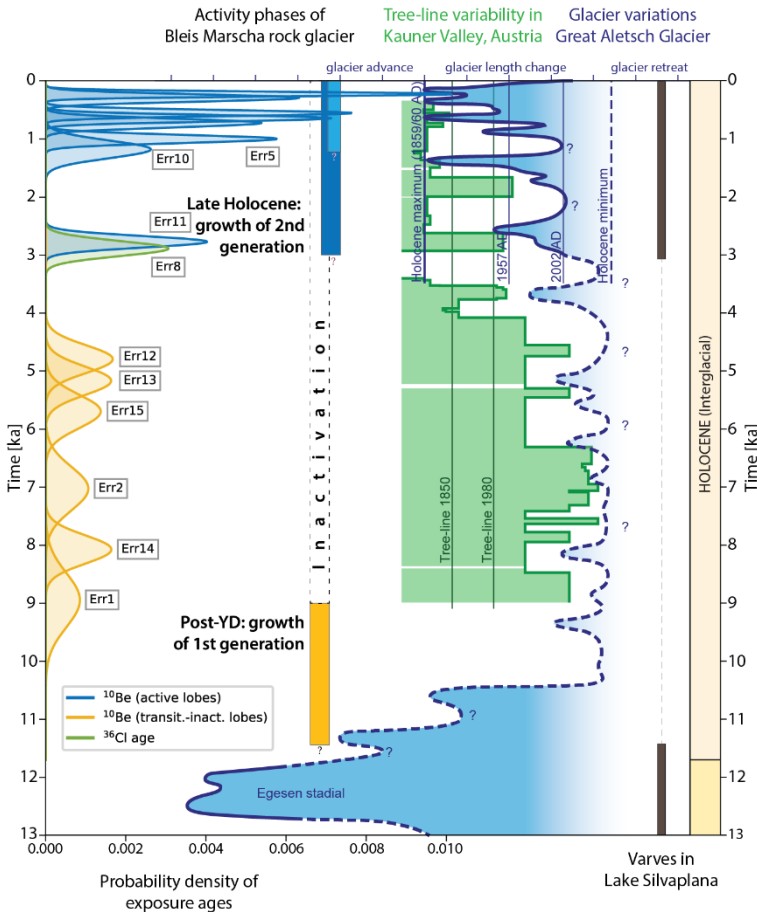

**Figure 11: Bleis Marscha rock-glacier activity phases in the framework of the general climate evolution in the Central-Eastern Alps during the Younger Dryas and the Holocene (modified from Ivy-Ochs et al. (2009) and Singeisen et al. (2020)). Climate proxies are Great Aletsch glacier length variations (modified from Holzhauser et al., 2005), Kauner valley tree-line variations (Nicolussi et al., 2005) and varve deposition in Lake Silvaplana (Upper Engadine; Leemann and Niessen, 1994). The probability densities of the exposure ages show two distinct generations separated by a millennia-long phase of inactivation: The currently transitional-inactive first generation developed shortly after the Egesen cirque glacier retreat and became inactive in the early Holocene. The disturbance associated with inactivation (settling, boulder weathering and spalling) resulted in nuclide loss and the observed large age dispersion. The exposure ages record the (minimum) time elapsed since inactivation. The currently active second generation developed coeval with the late-Holocene cooling and glacier re-advances. The precise exposure ages of the last 1.2 ka correlate with distance from the source, suggesting stable permafrost conditions. These undisturbed exposure ages record time elapsed since boulder emergence from the talus and boulder travel time.**

### 5.2 Early Holocene: Rapid debris-conditioned growth

A set of lateral moraines descending on both trough shoulders of the upper Val d'Err, short moraine segments parallel to the lower lobes of the Bleis Marscha rock glacier (Fig. 3) and hummocky moraines at the valley bottom near Alp d'Err were assigned to the Egesen stadium by Frauenfelder et al. (2001) and suggest that upper Val d'Err was still heavily glacierised during the early Younger Dryas (YD). However, during the late YD, the climate shifted to more continental, dry-cold conditions, and the Egesen glaciers starved (Kerschner and Ivy-Ochs, 2008; Ivy-Ochs et al., 2009, Ivy-Ochs, 2015) (Fig. 11).





Palaeoclimatic reconstructions by Frauenfelder et al. (2001) indicate that the lower permafrost boundary was depressed
several hundred meters more than the glacier equilibrium line altitude (ELA), opening a wide periglacial altitude belt (cf. Steinemann et al., 2020). This dry-cold permafrost phase lasted from late YD until about 10 ka.

The low-elevation, older-generation part of the rock glacier (units I–III) is confined within, but not connected to Egesen lateral moraines and must have derived its Err granodiorite debris from the previous Egesen Bleis Marscha cirque glacieret accumulation area (in the back of cirque), i.e. the rock glacier must have formed *after* the Egesen cirque glacier starvation 595 (Frauenfelder et al., 2001). From this crosscutting relationship and the oldest and lowermost stabilisation age atop its front of $8.9 \pm 0.5$ ka (sample Err1) we narrow the emplacement to a short activity period within about 11.5 and 9.0 ka. This rapid, post-YD to earliest Holocene formation agrees with reconstructions in the Julier area by Böhlert et al. (2011a) and the findings of Frauenfelder et al. (2001).

Estimating a characteristic advance rate of 1100 m in 2500 years, a rock-glacier width of 150 m (<200 m), a height of 30 m 600 (<40 m), and a debris fraction of 0.6, we arrive at a debris flux $q \approx$ (150 m $\times$ 30 m $\times$ 0.6 $\times$ 1100 m) / 2500 a = 1190 m$^3$ a$^{-1}$ (<2110 m$^3$ a$^{-1}$). Spread over the source area, this debris flux theoretically corresponds to head-wall erosion rates of ~3.8–6.8 mm a$^{-1}$. We hypothesize that such a high debris flux was most likely not in balance with long-term average debris supply rate. Instead, the rapid advance of the debris-ice lobe was enabled and conditioned by debris supplied by the mobilisation of reworked Egesen cirque glacier-derived material (Frauenfelder et al., 2001, 2005) and the warming-induced rapid head wall 605 weathering during the early Holocene (Kenner and Magnusson, 2017). We imagine the lower elevation part of Bleis Marscha rock glacier as a rapidly down-rushing, short-lived debris pulse (cf. Kirkbride and Brazier, 1995). Our findings of a high rock glacier-headwall ratio (Kenner and Magnusson, 2017) despite relatively short activity phase of early-Holocene relict-inactive rock glaciers (Böhlert et al., 2011a) seems to be representative for the Julier-Albula region.

### 5.3 Middle Holocene: Slow thermal degradation

After 10 ka, the climate shifted towards warmer conditions (Schimmelpfennig et al., 2014; Solomina et al., 2015) and the lower permafrost boundary began to rise above the rock-glacier front at 2400 m a.s.l. (Frauenfelder et al., 2001). By the middle Holocene, the lower continuous permafrost boundary was at least as high as today, likely above 2600 m a.s.l. (present permafrost distribution/-2°C isotherm, Fig. 3) and likely located near the topographically constrained talus-rock glacier transition in the back of the cirque. The tree line was higher than today (Nicolussi et al., 2005) (Fig. 11).

From the regional climate history, both climatic and dynamic inactivation are conceivable. What drove early-Holocene inactivation? The present surface movement pattern and ground ice distribution is correlated with its coarse-debris cover which is favourable for cold ground thermal regimes. Immediately below the major inner scarp at 2550 m a.s.l. that separates the transitional-inactive early-Holocene from the overriding active late-Holocene generation, we observe surface speeds of 30–40 cm a$^{-1}$ (magnitudes similar to the upper active lobes), and en-bloc movement of an entire morphologically delineated



lobe (unit III, Fig. 4). First, this viscous deformation pattern indicates stress transmission over >100 m and requires the presence of (excess) ground ice – notably beneath an early–middle Holocene boulder mantle that predates the activity phase of the upper lobes by several thousand years. Although the high, active-lobe-like deformation susceptibility might be in part explained by higher ground temperatures at lower elevations, mechanical loading exerted by the overriding younger generation (unit IV, Figs. 4, 5a), and greater thickness of the two-generation stack of lobes, the ice content in the lower,

transitional-inactive lobes must be significant (and mapped as ice-rich permafrost type by Kenner et al. (2019)). Second, a fresh lateral boulder apron (Fig. 5c) shows that the lobe (unit III) began to pour over its own lateral, vegetated outer ridge. This implies recently renewed motion after a long quiet phase where soil could develop, i.e. a forced reactivation (cf. Kirkbride and Brazier, 1995) most likely due to mechanical interaction of lobes. The age of the inferred ground ice is not known and can be original as well as recharged interstitial ice (Giardino and Vitek, 1988; Colucci et al., 2019). From the

impression of a still intact-looking micro-topography (Fig. 5b) we believe that the ice is as old as its covering debris mantle and of early Holocene age, otherwise the surface must look more disrupted and collapsed. Self-preservation of ground ice thanks to its high heat capacity and the cooling effect of air circulation in the overlying coarse-debris mantle is a commonly observed phenomenon in rock glaciers (Guodong et al., 2007, Colucci et al., 2019; Kellerer-Pirklbauer, 2019). For example, preservation of early-Holocene subsurface ice through the mid-Holocene warm period is interpreted by Krainer et al. (2015)

for the the Lazaun rock glacier. In contrast, unlike most of the lower rock-glacier lobes, the rock-fall deposits ("radiolarite protrusion", Figs. 3, 5a) show no movement and remained stable despite the mechanical load of the advancing 'rapid lobe'. These parts likely contain no ground ice. As these parts are below fine-grained surface cover (fractured radiolarite, soil, grassy vegetation; Fig. 12), the local ground thermal regime is less susceptible to permafrost conditions (Harris and Pedersen, 1998; Schneider et al., 2012). This has likely enabled substantial or complete ground ice loss in the past.

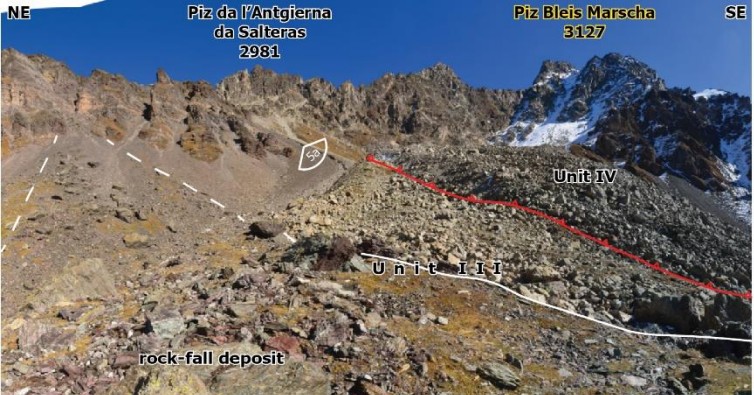

**Figure 12: View upstream (to E) from the radiolarite debris-fall deposits. Coarse debris-covered unit IV (late-Holocene age, 'rapid lobe') rapidly advances over unit III (early-middle Holocene). Although overall moving and therefore ground-ice bearing as well, unit III is stable in this corner where it is covered with finer debris of shattered radiolarite and soil (cf. Fig. 9). The boundary between moving and stable areas coincides with the different substrate types. The ground thermal regime under the fine material has not been favourable for ground ice preservation.**





Rock glacier degradation is conditioned by the energy fluxes towards the ice-rich permafrost body. Water plays a crucial role as a coupling agent: Water contributes to thermal degradation (thawing) via rapid heat advection and enhances motion (up to mechanical destabilisation) by increasing pore water pressure (Sorg et al., 2015; Wirz et al., 2016; Cicoira et al., 2019). For the low-elevation, early Holocene lobes of Bleis Marscha rock glacier, inactivation extended over several thousand years after emplacement, and is still incomplete. Ongoing movement and the intact micro-topography shows that the relict status is

by far not attained. Only the collapsing southernmost corner shows signs of advanced permafrost degradation and ice loss. To stress the slow transition towards a relict state (degradation), we classified the low-elevation lobes as "transitional-inactive" (cf. Giardino and Vitek, 1988). We explain the slow degradation and inactivation of the low-elevation lobes by (i) thermal decoupling from the atmosphere (coarse-debris mantle) and (ii) partial hydrological decoupling to upslope area and neighbouring talus (as the lobes rise above adjacent talus slopes).

Two reasonings answer the question "What drove early-Holocene inactivation?". First, the presence of azonal ground ice below the -2°C isotherm at 2540 m a.s.l. speaks against climatic inactivation (Fig. 3, lower boundary of discontinuous permafrost (Haeberli, 1985; Colucci et al., 2019)). Second, given the enormous post-YD debris flux likely exceeding the supply rate, it is plausible that the advance ceased as debris storage became exhausted (supply-limited system). Our hypothesis of dominantly dynamic inactivation agrees with modelling studies that find that debris supply governs the long-

term development (thickness and advance rate) of rock glaciers (Müller et al., 2016).

**5.4 Late Holocene: Intermittent glacier–rock glacier interaction**

In the late Holocene, after approximately 4 ka, climate cooled, timberline moved to lower elevations, and glacier advances became more frequent, longer and more severe compared to the middle Holocene (Fig. 11) (Joerin et al., 2006; Ivy-Ochs et al., 2009; Le Roy et al., 2015; Badino et al., 2018 and references therein). The rapidly fluctuating glaciers were forced by

climatic instabilities on a centennial timescale, superimposed on a multimillennial-scale cooling trend towards increased glacier extents culminating in the LIA between 1350 and 1850 (Solomina et al., 2016). Increasingly wetter conditions after 4 ka (Zerathe et al., 2014), more frequent glacier advances (Badino et al., 2018), air temperature oscillations and concomitant freeze-thaw cycles weakened the headwall and increased frost shattering and debris production, enhanced by the tectonically weakened fault zone in the headwall (Figs. 3–5a).

Due to the decrease of horizontal speeds with depth and the frontal ice melt-out, the rock-glacier surface moves faster than the advancing landform itself ('conveyor belt'-like advance mechanism, Haeberli et al., 1998; Frauenfelder et al., 2005; Kääb and Reichmuth, 2005). The maximum surface exposure age, measured atop the rock glacier front, gives a minimum landform age (e.g. Scapozza et al., 2014). The oldest exposure age (Err11) atop the front of the active upper lobe indicates that the most recent phase of activity phase likely began at or just before $2.8 \pm 0.1$ ka and lasts until today. According to the

dual-threshold model presented in Kirkbride and Brazier (1995), initiation of a new rock-glacier lobe occurs when an external climate threshold and an internal shear-stress threshold are crossed simultaneously. As the debris reservoir in the



Bleis Marscha cirque was replenished since inactivation of the first generation, sufficient internal ice could segregate and super-saturate a sufficiently thick debris accumulation with some lag after onset of the late Holocene cooling. At the moment of onset of viscous creep, a new, second-generation rock-glacier lobe was formed.

Long-term average debris fluxes of the late-Holocene generation rock-glacier lobe are in the order of 370–810 m$^3$ a$^{-1}$, calculated from mean speeds of 480 m/2800 a (length of lobe divided by oldest exposure age) and 0.3 m a$^{-1}$ (average speed of last 1200 years), respectively, and a rock-glacier width of 120 m (<150 m), a height of 30 m, and a debris fraction of 0.6. These fluxes correspond to head-wall erosion rates of ~1.2–2.6 mm a$^{-1}$. Given the massive debris accumulation in the cirque (Fig. 5a), the system must have been transport-limited during the late Holocene active phase, i.e. the debris evacuation rate

cannot have exceeded the supply rate by head-wall retreat. In contrast to these long-term average and therefore sustainable debris fluxes, the currently inferred flux on the 'rapid lobe' is likely much higher, $q \approx 120$ m $\times$ 30 m $\times$ 0.6 $\times$ (0.5 to 0.6 m a$^{-1}$) $\approx$ (1080 to 1300 m$^3$ a$^{-1}$). The discrepancy between currently observed rock glacier fluxes with respect to inferred long-term talus supply rate can be explained by kinematic wave theory (Degenhardt and Giardino, 2003; Müller et al., 2016). The 'rapid lobe' (frontal parts of unit IV) is possibly a surge package (Kenner et al., 2014) or a wave of increased discharge,

possibly amplified by the terrain drop-off below the cirque lip (Fig. 5a). It is expressed as a down-stream travelling bulge that propagates faster than the mean surface velocity (Degenhardt and Giardino, 2003). Currently observed surface speeds are temporarily higher than the life-time average and the travel time could be underestimated (Fig. 10). This would bring the outlier sample Err11 towards the 1:1 line (Fig. 10).

During the late Holocene cold phases, the Bleis Marscha cirque was likely intermittently occupied at the most by a perennial

ice patch or a glacieret (Frauenfelder et al, 2005), similar to the LIA extent as mapped in the mid-19$^{th}$ century (Dufour, 1853; Siegfried, 1887; cf. Badino et al., 2018). The ice patch likely never extended beyond the margins of the cirque (Fig. 4). The micro-topography of the lower part has been well preserved throughout the entire Holocene, as it was apparently beneath the lowest limits of any subsequent glacier advances.

The intact morphology of the active rock-glacier lobes (units IV–V), the continuous exposure age progression and the

consistent and moderate (~30 cm a$^{-1}$) magnitudes of short- and long-term creep rates suggest uninterrupted creep for the last ~1200 years without disruptive changes in the dynamics or thermal state. Permafrost conditions have likely been relatively stable in the high-elevation (>2700 m a.s.l.) cirque despite the climate oscillations of the last 1200 years. This is plausible given the fact that the shadowed cirque floor is in the continuous permafrost belt even today (Gruber et al., 2006; Boeckli et al., 2012). Although the debris provenance of the Bleis Marscha rock glacier (Err granodiorite) is in the headwall behind the

now-gone glacieret and there must have been glacier–rock glacier interaction, the rock glacier is clearly older than the LIA glacieret. Despite the likely incorporation of sedimentary ice and glacial debris, the Bleis Marscha rock glacier is not a glacier-derived rock glacier that has transformed from a LIA ice glacier. Instead, weak morphological evidence of the presence of the glacieret suggests that the small glacieret was stagnant or thin. The late Holocene cirque glaciers could not
destroy the rock glacier; permafrost and the pre-existing rock glacier outlasted beneath the ice patch and were not disrupted
by the presence of the glacieret, agreeing with Frauenfelder et al. (2001, 2005) and Kenner et al. (2018). The reason for these
stable permafrost conditions during the entire lifetime of the Bleis Marscha active rock-glacier lobes might be the very
permafrost-friendly, shadowy, high-elevation cirque with sparse glaciation during the late Holocene (Dufour, 1853). The
Bleis Marscha rock glacier root at an elevation above 2650 m a.s.l. is even now (in a relatively warm climate) above the
permafrost limit and must have been so during extended periods of the late Holocene (Gruber et al., 2006; Boeckli et al.,
2012). During LIA-like cold pulses of the late Holocene, the lower limit of permafrost was depressed by many tens of meters
(Frauenfelder et al., 2001). This is a very favourable environment for ice preservation and slow, steady rock glacier growth.
With the late Holocene climatic oscillations (oscillating ELA), the dominant coarse-debris transport mechanism in the rear
part of Bleis Marscha cirque alternated between a "conveyor-belt like" glacial transport during cold phases and permafrost
creep during glacial ice-free "mild" phases (Zasadni, 2005; Kenner et al., 2018). Ice of glacial and non-glacial sources might
have been incorporated. During the late Holocene, the continuously growing rock glacier intermittently coexisted with a
fluctuating cirque glacieret.

## 6. Conclusions

We constrained activity phases and reconstructed the morphodynamic development of the Bleis Marscha rock glacier (Val
d'Err, eastern Switzerland) with 15 cosmogenic nuclide exposure ages (14 $^{10}$Be and 1 $^{36}$Cl sample) and present (2003/2012)
surface creep quantification from aerial image correlation. Bleis Marscha is a 1100 m long, polymorphic talus rock glacier at
an elevation range of 2400 to 2700 m a.s.l., with active upper lobes and transitional-inactive lower lobes. Morphological
discontinuities (steep scarps) coinciding with kinematic discontinuities (decorrelation gaps, velocity jumps), and two distinct
exposure age populations indicate that the Bleis Marscha rock glacier is composed of two distinct groups of lobes linked to
two activity phases in the early and late Holocene, separated by a middle Holocene period of inactivation and quiescence.

The low-elevation older generation formed during the late Younger Dryas and persisted into the early Holocene in a short
time frame of a favourable succession of climatic conditions: First, an aridification at still cold conditions during the late
Younger Dryas, leading to glacier starvation, retreat, and permafrost growth, later followed by significant warming at around
10 ka. Crosscutting relationships in the field suggest that the rock glacier formed after the retreat of the Egesen Bleis
Marscha cirque glacier. The six $^{10}$Be ages for the older generation lobes have a large dispersion and show no correlation to
downstream distance. Exposure age dispersion and inversions for a single inactive lobe reflects settling and decay during
inactivation, affecting the nuclide inventory of boulders randomly and independently. We interpret these ages as inactivity or
stabilisation ages. The oldest age from a boulder atop the lowermost front places the end of the activity phase at no later than
8.9 ± 0.5 ka. The rapid emplacement (1100 m within ~2500 years) of the early-Holocene generation was enabled by high
debris fluxes likely exceeding a sustainable supply rate. It evacuated previously accumulated Egesen-glacier derived debris,





whose exhaustion ultimately lead to stagnation and cessation of the advance. The pulse-like behaviour in the early Holocene warming climate and the incorporation of glacial and non-glacial material agrees with previous studies on the Bleis Marscha rock glacier.

Ongoing coherent surface deformation and settling governed by the small-scale micro-topography on the morphologically transitional-inactive lobes requires the presence of ground ice as stress transmitter beneath the early Holocene boulder
mantle. Although these lobes are below the current -2°C isotherm, we believe from the still intact micro-topography that the ice is original, preserved from the early Holocene. Thermally driven inactivation of this azonal, ice-rich permafrost body, in contrast to its rapid emplacement, has been slow and is still incomplete. Therefore, dynamic rather than climatic inactivation caused the rock glacier to stall. Due to the high heat capacity of ice-rich material, the thermal decoupling and ground cooling provided by the coarse-debris mantle, sufficiently stable permafrost conditions were maintained even during the middle
Holocene warm phase.

The high-elevation younger generation developed from the late Holocene until today. The oldest exposure age atop the front of the active upper lobe indicates that the most recent phase of activity phase likely began before 2.8 ± 0.1 ka. This late-Holocene rock glacier advanced by 460 m in 3000 years and involved moderate and most likely sustainable debris fluxes. The nine exposure ages show a small dispersion and correlate linearly with down-stream distance. This suggests undisturbed
creep at an average rate of 30 cm $a^{-1}$ during the last ~1200 years, a rate that is consistent with present short-term 2003/2012 creep rates we determined. Furthermore, travel time estimates from streamline interpolations agree well with the exposure ages. The rock-glacier lobes developed uninterruptedly under stable permafrost conditions despite the climatic oscillations during the Medieval Warm Period, LIA, and the current warming up to now. Due to favourable topo-climatic conditions in the shadowy, high-elevation cirque (>2650 m a.s.l., above the -2°C isotherm) and polythermal-cold based cirque glacieret,
ground thermal conditions in the rock-glacier root zone remained cryotic despite late Holocene climate oscillations. Our data from the active lobes of the Bleis Marscha rock glacier show that exposure ages for rock-glacier lobes that are presently active and have not experienced disruptive dynamic changes during their lifetime can be interpreted as (minimum) travel times. Notably, it also shows that pre-travel nuclide build-up is insignificant. The rock glacier is sourced in the during late Holocene intermittently glacierized Bleis Marscha cirque. Even during the LIA glacier high stand, interactions with the
cirque glacieret were not disruptive for the rock glacier development. The Bleis Marscha rock glacier predates the LIA glacieret, which however does not exclude the incorporation of glacier-derived material (debris and ice).

Although the two rock-glacier generations originated in the same cirque, we find contrasting responses to external forcing by comparing typical advance rates and debris flux estimates for both rock glacier generations. This very different dynamic history of the two rock-glacier generations is expressed in the exposure age distributions of the different lobes.




*Code availability.* The code used for flowline and strain-rate calculations is available by request from the corresponding author.

*Author contributions.* S.I.-O., O.S., and M.F. designed the study. D.A., S.I.-O., O.S. and M.F. conducted the field work. O.S and D.A. carried out the $^{10}$Be extraction, O.S. and C.V. the $^{36}$Cl extraction, and M.C. the AMS measurements. D.A. computed the image correlation. D.A. and M.F. did the finite-element modelling. D.A. prepared the figures and the manuscript which were edited by S.I.-O., O.S. and M.F.

*Acknowledgements.* We are grateful for the field support by Ueli Steinemann, Reto Grischott and Jonas von Wartburg. We thank Kristina Hippe and Ewelina Bros for their assistance during sample preparation and the Laboratory of Ion Beam Physics group for excellent AMS measurements. We thank Armando Janett (local gamekeeper), Carole Müller, and Peter Kaiser for their logistical support.

*Financial support.* D. Amschwand received financial support for the fieldwork by the Swiss Society for Quaternary Research. This work was completed in the framework of Swiss National Science Foundation project (SNF) 175794.

*Competing interests.* The authors declare that they have no conflict of interest.

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
