# Peer review of "Deciphering the evolution of the Bleis Marscha rock glacier (Val d'Err, eastern Switzerland) with cosmogenic nuclide exposure dating, aerial image correlation, and finite element modelling"

_The Cryosphere, 2020_

## Referee Comment (RC1) · Jakob Heyman (Referee) · 10 Sep 2020

General comments

The paper presents a study of a rock glacier in eastern Switzerland combining cosmogenic dating, recent velocity estimation from aerial imagery, and simulation of the rock glacier deformation in an effort to investigate and reconstruct the post-glacial rock glacier evolution. The questions addressed appear relevant for The Cryosphere and

the study presents a novel and interesting concept combining multiple chronological and numerical techniques. The paper is clearly presented and it is mostly easy to understand. I find the paper highly interesting and mostly very good. There are however a number of issues, primarily related to the interpretation of the exposure ages, that should be addressed before publication. In particular, several interpretations of the exposure ages appear to be not supported by the data, and for some interpretations the underlying assumptions have not been spelled out. In the specific comments and technical corrections below these issues (and other more minor issues) are presented in more detail.

Specific comments

1. Exposure age interpretations 1 In section 4.2, you write: "The exposure ages in general anticorrelate with elevation and correlate with down-flow distance". I agree with this but an important part here is "in general". A few lines further down you write "The boulders remained exposed at the surface and were passively transported at the rock glacier surface between talus and top of front slope [—]. We conclude that the exposure ages on the active lobe have negligible systematic errors and no overall age shift from inheritance/pre-exposure (systematically "too old") or nuclide loss/incomplete exposure (systematically "too young")". This statement is in my mind incorrect and it should be changed. It is clear from Figure 6 that a maximum of three out of seven samples overlap within uncertainties with the linear regression line. This implies that at least four out of the seven samples (plus sample Err8 and Err11 which are not included in the linear regression) have experienced either prior or incomplete exposure. The absolute numbers of the age "error" are small (<1 ka) but the relative errors are not. For example, sample Err6 has an exposure age that is less than half of the age expected from the linear regression. Such an error cannot be seen as "negligible" if trying to use the age to constrain the velocity of the rock glacier. I suggest that you are more open with the fact that the samples may have experienced both prior and incomplete exposure and that it is really difficult to identify how much each of these

processes have affected the exposure ages. If assuming that the linear regression actually represent the rock glacier transport an underlying assumption is that prior and incomplete exposure have affected the group of exposure ages by an equal amount. This assumption should be properly spelled out in the text. It is easy to argue that both prior and incomplete exposure may very well have affected the exposure ages. We will likely get prior exposure unless the rock-wall erosion is really deep/rapid and we can easily get incomplete exposure if the rocks on top of the rock glacier move around and topple every now and then. The scatter of exposure ages around the regression line could perhaps be used to try to estimate an uncertainty of the rock glacier velocity.

2. Exposure age interpretations 2 Further down in section 4.2, you write: "the 10Be ages of Err2 and Err15 are significantly younger than the further up-slope sampled Err14. It is unlikely that this apparent age inversion reflects internal creep, as the laminar flow behaviour of a rock glacier makes mixing impossible and contradicts our finding of linear relationship between exposure age and down-flow distance on the active lobes". I agree with the statement about samples Err2 and Err15. However, just the same can be said about samples Err6 and Err7 on the active lobe which are younger than samples Err5 and Err9 further up-slope. The difference in the interpretation of the exposure ages on the active lobe and the inactive lobe is strange and not supported by the data, and I think that it should be changed.

3. Exposure age interpretation 3 In section 5.1 (L 517-518) your write about Figure 10: "The exposure ages for the samples located on the presently active upper lobes are slightly less than the travel times by about 10-100 years, which is within the uncertainty margins". This statement is wrong and it should be changed. Of the seven samples falling below the 1:1 line, only one sample (Err5) overlap with the line. You further write: "The measured surface velocity field is representative, and the discrepancy is smaller the closer the samples are to the central flow line (Err3, 4, 5, 10), where the influence of lateral viscous drag is smallest." It is correct that the age discrepancy is smaller for samples 3,4,5,10 compared to samples 6,7,9, but the age discrepancy is still large for

samples Err3 and Err4. Sample Err3 has an exposure age of 229 years compared to an estimated travel time of 800 years. Sample Err4 has an exposure age of 556 years compared to an estimated travel time of 1000 years. The statement about the discrepancy should be changed.

4. Exposure age interpretation 4 Further down in section 5.1 (L 529-531), you write: "Our data on the active lobe of Bleis Marscha rock glacier show that the exposure ages for an undisturbed rock-glacier lobe can be addressed as (minimum) travel times. While travelling during the active phase, boulders remain mutually interlocked within the clast-supported framework and the age dispersion remains small. Notably, it also shows that pre-travel nuclide build-up, inheritance, and pre-exposure in the rock wall or on the talus are insignificant." These statements are poorly supported by the data and they should be changed. The age dispersion is NOT small for the question of rock glacier velocity (several hundred years for multiple samples!). As the ages DON'T cluster and overlap with the estimated travel times, there MUST have been some prior and/or incomplete exposure causing the exposure age scatter (unless the scatter comes from the measurements or some other source of error).

5. Exposure age interpretation 5 In lines 550-553 you write: "Intermittent boulder instability during inactivation has been affecting the nuclide inventory of each boulder individually, leading to incomplete exposure by self-shielding and nuclide loss by weathering or spalling, and ultimately to tendentially underestimated true exposure times ('apparent rejuvenation'). This results in a large dispersion of exposure ages and exposure age inversions on the low-elevation rock-glacier lobes." I largely agree with these statements about the exposure ages from the inactive part of the rock glacier. However, we cannot really be sure that NO prior exposure has affected the exposure ages. Further, from Figure 10 it is clear that if lifting up the 1:1 line the samples from the inactive part could fit the line much better than the samples on the active part. You never discuss this fact in the manuscript and I think that you should. You could fit a regression line to the inactive samples in the same way as to the active samples and

my guess is that you would get a similarly good match to the data. I imagine that such an approach could potentially be used to try to estimate the velocity of the presently inactive part during its active phase (although this will of course involve several questionable assumptions...). In any case, I think you should at least mention more clearly the correlation of exposure age against down-flow distance also on the inactive part of the rock glacier and try to offer some explanation for it.

6. Exposure age interpretation 6 In lines 564-568 you write: "We find that exposure ages need to be interpreted according to the dynamic history of the sampled rock-glacier lobe. On undisturbed lobes where the current surface creep is concordant with the micro-topography, exposure ages represent (minimum) travel times. This applies to active lobes without destabilisation. On 'discordant' lobes where micro-topography reflects different stress conditions and must have undergone disruptive dynamic changes, the exposure ages represent time elapsed since the major dynamic change. Inactivation or destabilisation are such disruptive events." I find these statements too categorical and not really well-supported by the data. First, if the statement about undisturbed lobes is correct and the estimated 36Cl age of sample Err 8 is roughly correct, the minimum travel time of the active lobe is significantly longer than what you state elsewhere in the manuscript. Second, if the statement about 'discordant' lobes is correct, the "major dynamic change" must have been in the period 8.9 to 4.8 ka. That is a long period (c. 4000 years) for a "disruptive event". Third, with data from one specific rock glacier it seems a bit too bold to draw conclusions about exposure dating of rock glaciers in general (which is how I interpret text).

7. L 135 "using the CRONUS-EARTH online calculator". You should state which version of the calculator you have used. Further, I would suggest using the updated version 3 calculator and perhaps also use the Chironico 10Be reference production rate directly instead of the northeast North America production rate. In the v. 3 calculator you can pick the Chironico input data from the ICE-D database to calibrate the production rate and then use that to calculate exposure ages. Perhaps you should also

exclude the sample with the lowest 10Be concentration because with that included you get a P-value of only 0.033. However, all this is not really important for the outcome so just disregard it if you don't want to do the extra work.

8. Table 1. I suggest changing "calibrated to the 07KNSTD standard" to "calibrated to the S2007N standard" because that is what is stated in section 3.2. I also suggest that you add the down-profile distance for each sample that you use in Figure 6. That data is not included anywhere else in the manuscript and I think that it should be included somewhere.

9. L 673-674 "The oldest exposure age (Err11) atop the front of the active upper lobe indicates that the most recent phase of activity phase likely began at or just before $2.8 \pm 0.1$ ka and lasts until today." This statement is correct only under the assumption that the sample has not experienced prior or incomplete exposure, and that assumption should be clearly expressed. See also lines 751-752 where you write almost exactly the same but "before 2.8" instead of "at or just before 2.8".

10. L 756-757 "Furthermore, travel time estimates from streamline interpolations agree well with the exposure ages." This is incorrect - see points 3-4 above.

11. L 771-772 "Code availability. The code used for flowline and strain-rate calculations is available by request from the corresponding author." Why not upload the code to a repository and add a link to it or include it as a supplement? That would make it easier for anyone who is interested in the code.

Technical corrections

12. L 13 I suggest changing "The results suggest" to "The results indicate".

13. L 65-66 "Long-term effects on rock glacier development must remain unresolved (Kenner and Magnusson, 2017)" This sentence sounds a bit strange to me (in particular "must remain") and I suggest rephrasing it.

14. L 68-69 "Cosmogenic radionuclides exposure dating is a unique tool because it

directly measures a chronometric, numerical exposure age of the landform surface". I suggest rephrasing this. Exposure dating does not directly measure an exposure age - it measures the cosmogenic nuclide concentration which we can use to calculate an exposure age (involving several critical assumptions).

15. L 74 "at an elevation range of 2400–2700 m a.s.l. Previous studies". I don't know what is correct writing here, but it should perhaps be one more punctuation mark after "m a.s.l." to show that the sentence ends.

16. L 116 I suggest changing "the surface exposure dating method" to "surface exposure dating".

17. Figure 2 Disregard these comments if you disagree (they regard the visual impression and are not really important). In figure a, the text with the white edges does not look good so I suggest trying to use just white text or to make the white edges thinner or perhaps bright grey. In figure b, I suggest making  and $\mu$ smaller (they look too big). It also looks a bit strange with having "surface", "boundaries", and "base" underlined.

18. Figure 3 I suggest adding units I to V to this map. For the three texts with white background, I suggest trying to make the white edges thinner or perhaps bright grey.

19. Figure 4 It is a bit strange to have this map shifted 90 degrees compared to Figure 3 and I would probably try to keep the same orientation in both figures (although this is not really important so disregard this comment if you wish). For the black texts with white background, I suggest trying to make the white edges thinner or perhaps bright grey.

20. Figure 5 The text in the photos is in several cases a bit difficult to read. I suggest trying to change the black text with white background to just white text or perhaps white text with black or grey background (but thinner background text than in the present version).

21. L 331 "ages ranging from ∼4.8 to ∼9.5 ka". The age 9.5 ka here seems to be too

high as the oldest age is 8.9 ka.

22. L 504-505 "The active high-elevation lobes (2550–2700 m a.s.l.) with surface speeds up to 60 cm a-1 display clustered exposure ages within 0.23 $\pm$ 0.04 to 2.8 $\pm$ 0.1 ka". The exposure ages on the active lobe should NOT be called "clustered". See major points 1 and 2 above.

23. Figure 10. You state in the figure caption that the exposure age uncertainties are the "analytical uncertainties (internal errors)". Although it won't really matter for the outcome, you should rather use the external uncertainties including production rate uncertainty as you are comparing the exposure ages with travel time estimates that are independent from the exposure ages.

———————————————————

---

## Referee Comment (RC2) · Leif Anderson (Referee) · 5 Oct 2020

Review of "Deciphering the evolution of the Bleis Marscha rock glacier (Val d'Err, eastern Switzerland) with cosmogenic nuclide exposure dating, aerial image correlation, and finite-element modelling" by Amschwand et al..

By Leif Anderson

[Figure]

The manuscript considers both the modern and past states of Bleis Marscha rock glacier in Switzerland. The authors use field observations, surface velocity estimates, finite-element modeling, and surface exposure ages to explore the evolution of the rock glacier. Ultimately the authors use their datasets to interpret the Holocene history of the rock glacier as related to climate and erosion rate from the headwall.

General Comments

This is truly an exceptional amount of work and it is really an ambitious project and manuscript! The authors should be proud of this achievement. I am especially impressed with the bringing together of the methods of a glaciologist/physicist with the methods of a geomorphologist/paleoclimatologist. The comments below are mostly related to the presentation of the work, with some suggestions which could improve the analysis.

It would be great if the surface velocity estimates, the field observations, and the model could be better used to justify the interpretation of the surface exposure ages. Right now the manuscript seems to touch on these different features and then transition into the Holocene history of the Bleis Marscha rock glacier without too clear of a connection between the modern and paleo perspectives. It would better honor all the work in the manuscript if logic behind the assumptions being made was laid out in connection to the modern analysis of the rock glacier. This is no easy task, but I think one that will really highlight the broad scope of methods brought to bear in this body of work.

I think that the uncertainty in the surface velocity estimates could re-done using off-rock glacier velocities adjacent to the rock glacier itself. Right now I suspect that the surface velocity error is too small. Comments below explain this more fully.

The modeling analyses could be improved with some sensitivity tests exploring assumed parameters and the ice-rock ratio but I am not sure they are necessary. Really a more clear statement of which parameters are assumed and how those parameters where chosen is enough.

I suggest that the authors go through the full manuscript with a discerning eye for which observations and analyses are really needed to support the main take homes from the manuscript. There are various classifications provided of the rock glacier surface Units and surface zones on the glacier that do not coincide. These multiple classifications are difficult to keep straight so some simplification would be helpful for the reader. As can be seen in the detailed comments below I suggest in some cases for material to be moved into a supplemental section and for some figures to be simplified. Perhaps figures 4 and 7 could be combined into a 3 panel figure? The text should be simplified to improve the reader's experience and their ability to access the science. Terminology needs to be used consistently throughout the manuscript especially regarding section titles from the methods, to the results and discussion.

Overall this manuscript will be great contribution to the field, but it needs a clean up in terms of presentation, readability, and connection between the diverse datasets produced.

Line-by-line comments

Line 11. "2003/2011" is unclear to me. Perhaps note that repeated surface velocities measurements were made.

line 12. I am not sure what "orthophoto orientation correlation" means here.

Line 19. Consider rewording as I am not completely sure what is meant. "Nuclide loss from boulder erosion, affecting the nuclide inventory of boulders independently"

line 31. change 'is' to 'are'

line 32. revise this sentence

line 33. remove 'resources'

line 35-50. You might emphasize the considerable differences in the timescale between these two concepts.

Line 55. I am not aware of 'dynamic inactivation' is there a citation for this term? It confuses me a bit because I tend to think of 'dynamics' as the flow of the rock glacier body and not a processes related to changing headwall erosion rate to the rock glacier.

Line 63. It is not clear what 'interactions with glaciers' refers to.

Line 65-66. Consider revising to make the meaning more clear. Could new methods / approaches help resolve this issue though?

Line 73. Why this rock glacier? A sentence about why you chose this rock glacier would resolve this.

Line 78. instead of 'present' maybe use 'modern'

Line 106-7. The materials and methods section would benefit from a bit more generalized and expanded text here more broadly introducing the methods. A road map into the diverse methods applied in this study would help. This is easily fixed!

Line 109. Perhaps a bit more detail about the mapping performed? What sort of mapping did you conduct? For what purpose what the mapping conducted?

Line 115. It would be nice to have a map of the rock glacier with the locations of CRN sampled boulders referenced in this section if not before this section.

Line 116. Expand or combine this paragraph as it is only one sentence. This sentence itself can be simplified as well.

Line 146-147. How were these erosion rate values chosen? How much does these assumed erosion rates effect your exposure ages and your conclusions? Perhaps a few extra sentences would help here as well as some citations to support the assumption.

Line 150 . It might be easier for the reader if the heading here is 'surface velocity' or 'surface creep rate estimates.'

Line 152. reword 'The used'

Line 163-165. "We estimate the uncertainty by correlating a reference area in the valley floor considered as stable."

I suspect that this will under estimate the error on the rock glacier because the area. Is there reason to expect that the off-rock glacier areas in figure 7a are moving at 10 cm/a? It seems to me that a more robust estimate of the error of the surface velocities would come from the large off-rock glacier areas in figure 7a that show velocities up to 20 cm/a. This is also consistent with the lack of correlation between slope and velocity on the rock glacier in figure 7b.

Line 166-177. These paragraphs are interesting and well written but I am not sure why they are being included. How does calculation of strain rates relate to the larger framework of the manuscript? Maybe a few sentences of introduction to the section at line 151 could provide a road map for the calculations made related to the surface velocity/creep estimates.

183-4. Is this the only process by which rock glaciers move? How about translation along shear zones/ sliding? Does it make sense to state that you assume that movement of this rock glacier occurs by internal deformation?

202-204. It seems you should state clearly that you assume that this 3 layer structure applies to Bleis Marscha rock glacier.

209-10. If you give a few general observations here the reader does not need to search for the justification in the later section.

211-12. "a 3 m thick basal low-viscosity shear zone (constant)." Does this not also contribute to the movement of the rock glacier? How is the viscosity in the low-viscosity shear zone constrained? If this low viscosity portion of the rock glacier is included here then it seems it should be discussed above where you mention processes leading to rock glacier movement.

Furthermore it seems that the actual values of effective viscosity produced by the

model are highly dependent on the assumed relationship between the viscosity and the viscosity in the shear layer at the base of the rock glacier. How important is the assumption that the shear layer contains a viscosity 10% of the rest of the rock-ice mixture? I don't think this has a large bearing on the main results of the manuscript though.

215. What is the Salteras terrace? Is it composed of river gravel or bedrock or till?

219-225. Is there a local justification for the 60% ice by volume? If not this should be stated as an assumption. Or further down a sensitivity test should be shown to highlight how much your results depend on this assumption.

Section 4.1 It would help the reader if the approach to mapping was outlined in the methods section. Right now there is scant mention of mapping methods, despite a rather large results section dedicated to it.

While there are valuable observations from the field here. I find that the section contains a lot of details that I am not sure how they connect to the rest of the exciting work presented in this manuscript. Perhaps it could be simplified and only the most necessary observations included. Other additional observations could be moved to the supplemental materials.

246. It would help the reader if you simplified the section title here.

250. Maybe describe what the estimate of volume was based on (i.e. what was the assumed mean thickness)?

Also I am not sure 'Internal' is needed here.

258. I am not sure what 'well-localized' means.

261-2. I am not sure I totally follow the reasoning here. It would help to spell it out more clearly.

266-69. If the ice patch was not flowing then it is not clear why geomorphological

evidence in the landscape would be expected.

293. Is there data to support this observation?

298-9. It is not immediately clear why this is calculated or how it ties into the rest of the manuscript.

351-2. Here I think you need to describe what those processes are. Boulder rolling seems like an important potential process on rock glaciers. You might see Crump et al., 2017 as well.

380. Simplifying the section heading will benefit the readability

386. What is the significant level based on? It seems that much of the off-rock glacier area in Fig. 7 is moving up to 10-15 cm/a. This makes me think that the error associated with the surface velocities should be higher.

390-412. It is hard for me to keep track of the different lobes as well as the newly presented creep rates here. Perhaps this section can be synthesized a bit more.

Section 4.4 This section could be simplified and maybe extra text moved into the supplemental section.

503-5. How can you be sure that these jumps aren't just associated with the steepening slopes at these lobe boundaries?

547-549. This suggests to me that these velocities are not the result of active flow but rather the motion of boulders due to surface processes, shadows, and spurious correlation. As suggested above I think the error uncertainties for the velocity estimates should be redone.

558. maybe 'preserves' instead of 'memorizes'

Line 560. "The exposure ages are rather inactivity or stabilisation ages than travel time estimates, as previously reported (Moran et al., 2016; Steinemann et al., 2020)." I don't

understand what is meant here.

560-63. I do not understand how the travel time is within the uncertainty of the exposure age on the lower part of the rock glacier. Maybe the travel time constitutes half of the exposure age for Err12 and 13, but certainty not the other samples.

564-70. This paragraph is hard for me to follow.

566-68. I do not understand how travel time can be neglected in this case. Perhaps the logic can be laid out more here.

599-602. I suggest that you state that these are 'back-of-th-envelope' estimates as a lot of assumptions go into them.

Conclusions : It would be good to see a bit more incorporation of the results from the velocities and model with the paleoclimate story.

Table 1. How sensitive are these results to the assumed surface erosion rate of the rock samples?

Tables 2 and 3. Perhaps move this table to the supplemental as the sample is assumed to be an outlier.

Figure 1. Very nice map and inset of Switzerland.

Figure 2. Panel (b) the '5 m' and '2 m' labels are for the boulder mantle and basal shear layer but that is not clear in the figure. The 0.1 x viscosity in the basal shear layer should be discussed in the methods portion of the manuscript and described as an assumed value.

Caption: what is the Salteras terrace? Maybe reference it as the lower geomorphological surface?

Figure 4. Making the fill less transparent for the CDN ages would improve legibility, as well as making the boxes around the CRN ages tighter. What does 'active highelevation lobes' refer to? I do not see any active lobate features

Figure 5. I find the caption difficult to follow. There is a not of information here, which is great, but I am not sure how it ties into the broader manuscript. Perhaps it could be moved into a supplemental section because it adds good background info.

Figure 6. This figure is a good synthesis of the different datasets produced. But I think the legibility of the figure can be substantially improved. Maybe the vertical dashed lines do not need to extend across the full height of the figure. It might instead work well to move this figure to the supplemental and then just include the exposure ages and the surface profile as a figure in the main text? It seems like the local topography and thrust activity are secondary controls that complicate the figure.

From the caption:

'This suggests that pre-travel nuclide concentrations are negligible.'

maybe add 'typically' in front of 'negligible.'

"Active thrusts coincide with sharp velocity gradients (cf. Fig. 7); this differential move-ment results in overriding lobes."

To me the assertion that the front of lobes can be positively linked to active thrusts is an interpretation here and throughout the manuscript.

Figure 7. Panel (a) The blue dots are not explained in the caption. The pink is hard to see. It seems that much of the off rock glacier area also produces significant velocities. Is this real motion?

Panel (b) the principle strain rates are very hard to read. Consider reducing the num-ber of plotted strain rates (same for Panel b arrows) or creating a raster of dominant compression versus extension areas of the rock glacier.

Figure 8. The colors between the panels should match otherwise it is very hard to read. It seems that velocities.

Panel (a) based on the histogram up to 20 cm/a

Below 20 cm/a there does not seem to be a positive correlation between velocity and surface slope. Based on the velocities from off the rock glacier of up to 15 cm/a does this not indicated that the below ∼ 20 cm/a the velocities could be noise?

Either use only 'surface velocity' or only 'surface creep rate' throughout the manuscript.

Figure 9. Lots of great information here but I would suggest just including the lower panel.

Figure 12. I would suggest that this figure be moved into a supplemental section.

Crump, Sarah E., et al. "Interpreting exposure ages from ice‐cored moraines: a Neoglacial case study on Baffin Island, Arctic Canada." Journal of Quaternary Science 32.8 (2017): 1049-1062.

---

## Author Comment (AC1) · 27 Nov 2020

Dear Jakob Heyman, Dear Leif Anderson, Dear Andreas Vieli, editor of The Cryosphere

We sincerely thank you and reviewers Jakob Heyman and Leif Anderson for considering our paper for publications in The Cryosphere. We thank both reviewers for their positive comments on our approach combining diverse methodologies to understand

the evolution of the Bleis Marscha rock glacier. This paper is one of the first to use cosmogenic nuclides on a fully active rock glacier. We obtained early Holocene exposure ages from boulders close to the toe of the rock glacier and ages of only a few hundred years close to the root zone. Both reviewers brought up questions on our interpretation of the exposure ages, notably that we used a different approach in interpreting the ages on the upper, more active lobe than on the lower lobe. These comments have led us to re-evaluate our interpretation of the ages on the lower lobe. In that light, we are convinced that the input from the two reviewers will lead to an improved paper and we look forward to having the opportunity to revise our manuscript along the lines suggested by the reviewers. Below we summarize the concerns brought up by the two reviewers and outline our planned approach to revising the paper. A number of points were raised by both reviewers. Comments relating to each specific review are given further below.

Both reviewers note that we have used different approaches to interpreting exposure ages for the upper versus the lower lobe. We thank the reviewers for pointing this out and will use a consistent approach in the revised version. Further details on this are given in our comments below on Reviewer 1's suggestions. In a similar vein, both reviewers point out that our thoughts on how exposure ages on active rock glaciers should be interpreted should be included. In a revised version, we will include a few sentences on what processes lead to too old ages and what lead to too young ages on a moving rock glacier. Adding such content to our manuscript will be an enlightened and incisive addition given that our ages actually cannot be interpreted as only time elapsed since stabilization, as in previous publications. Such a discussion will proceed specifically in light of the Bleis Marscha site.

Comments by Reviewer 2 additionally prompted a reassessment of the level of detection of the surface velocities from image correlation and clarifications on the assumptions embedded in the numerical model. We thank Reviewer 2 for these suggestions and will include these in the revised manuscript. See detailed response below.

Finally, Reviewer 2 states "It would be great if the surface velocity estimates, the field

observations, and the model could be better used to justify the interpretation of the surface exposure ages. Right now, the manuscript seems to touch on these different features and then transitions into the Holocene history of the Bleis Marscha rock glacier without too clear of a connection between the modern and paleo perspectives. It would better honor all the work in the manuscript if logic behind the assumptions being made was laid out in connection to the modern analysis of the rock glacier. This is no easy task, but I think one that will really highlight the broad scope of methods brought to bear in this body of work." In a revised version we will pay close attention to weaving the various different threads, geomorphology, isotopic dates and numerical modelling, together. Additionally, Reviewer 2 noted that the presence of unnecessary details leads to a digression from the main message, that the connection between observations/results and the interpreted Bleis Marscha rock-glacier history is not clear enough and that important thoughts on the interpretation of the exposure ages are not properly spelled out. Therefore, we will condense parts of the manuscript and move some of the details into a supplemental section. The re-interpretation of the ages on the lower lobe triggered by the reviewers' criticisms enables substantial simplifications in the revised manuscript. By spelling out the age model we also hope to clarify the connection between observations on Bleis Marscha and the interpreted rock-glacier evolution scenario. Both reviewers made a number of suggestions for improving the figures. These and all minor line-by-line points made by the reviewers will be taken into account in a revised version. Finally, as suggested by Reviewer 2, a roadmap that relates the different methods and results will be added. With these modifications, we hope to make the structure clearer, improve the readability, and resolve the reviewers' objections.

Reviewer 1, Jakob Heyman. Reviewer 1's comments focussed primarily on the interpretation of the exposure ages. He wrote "I suggest that you are more open with the fact that the samples may have experienced both prior and incomplete exposure". We thank him for his insight and are enthusiastic to include such considerations in our text. Reviewer 1's main concern is that the exposure age data were treated differently on the

upper lobe and the lower lobe. In a nutshell, if the upper lobe exposure data contain travel time, as shown by the ages becoming older the further away from the headwall scree slope, then consequently the ages on the lower lobe must also contain boulder travel time. This has never been shown before in any other rock glacier cosmogenic nuclide study and we are happy to consider in depth and to express this result. Reviewer 1 suggests that the degree of scatter on the ages on the upper and lower lobes is similar. This has led us to reconsider our interpretation of the exposure ages on the lower lobe and we will change the main interpretation accordingly. We will adopt the same interpretation scheme for all rock-glacier lobes, regardless of their age. Notably, this re-evaluation made us realize that the lower lobe of the rock glacier is not at all relict. The lower lobe is still active today, albeit only marginally. This is remarkable given its early-mid Holocene age and is in itself an important finding. As the actual velocity of the lower lobe can never be known (obtained velocities through aerial image correlation are for today), the amount of travel time contained in the nuclide inventory cannot really be known. We will incorporate such caveats into our revised manuscript. The new approach, that encompasses the fact that the exposure ages for the lower lobe do contain at least some component of travel time, allows the finite-element modelling and velocity determinations to be well merged with the exposure dating results.

Reviewer 1 suggests that we cannot strictly exclude bedrock rockwall pre-exposure. This is true but it is actually true for exposure dating of moraine boulders and landslide boulders. Only years of experience of dating moraines has shown that pre-exposure occurs in less than 2% of moraine boulders. For rockglaciers, such a body of data does not yet exist. Nevertheless, the fact that rock glaciers, and specifically the talus-derived Bleis Marscha, are fed by active talus production and frequent rock falls provides strong evidence against pre-exposure being acquired in the bedrock wall. Nevertheless, we remain open to this possibility and will cover this point in more detail in our revised version. On the other hand, we relax our previous assumption of 'passive transport' of the boulders on the active rock-glacier surface and readily acknowledge that exposure ages from single boulders may be 'too young' due to shifting or overturning. However,

based on our finding of a – in general – linear relationship between exposure age and down-flow distance allows us to glean information on the timing of development of the rock-glacier lobe from the boulder ages. In a revised manuscript, we will discuss these disturbances and limitations more openly.

Reviewer 2, Leif Anderson. Reviewer 2 brought up several of the same concerns as Reviewer 1, namely that the age data for the upper lobe cannot be treated differently than the age data for the lower lobe. We also thank him for the additional literature reference on boulder rolling as an important potential process on rock glaciers that we will consult. As described above these points led us to re-assess our interpretation of the ages on the lower lobe.

Reviewer 2 felt that too many details are included in the paper, which hampers its readability and accessibility. To streamline the paper, in the revised version we will delete some of this detail and place related text and figures in the supplemental material. For example, (o) Fourier analysis of surface topography to constrain the viscosity contrast between rock-glacier core and boulder mantle (viscosity ratio estimated with buckle-folding theory), (o) Error assessment of image correlation to estimate the level of detection (correlation of stable reference area in the valley floor), (o) Detailed image correlation results with strain-rate calculations and subdivision into kinematic domains. Similarly, a number of the different lobe classification labels will be deleted, to simplify the paper also in accordance with the re-assessed interpretation of the age of the lower lobe.

Reviewer 2 suspects that the level of detection of the surface velocity we use in the submitted version is too low (currently 5 cm/a) by pointing at the presumably stable off-rock glacier slopes that nonetheless show velocities that cluster at 5-10 cm/a. We partly agree and will discuss possible errors in the pre-processing and orthorectification of the orthophotos. In a revised version, we will adopt a more conservative level of detection of 10 cm/a. This change does not alter the picture of current Bleis Marscha kinematics.

Reviewer 2 asked for clarifications on: (i) how the deformation processes are treated in the finite-element model, and (ii) on the assumptions underpinning the construction of the numerical rock-glacier model (3-layer structure). First, all deformation mechanisms, from creep in the ice-rich rock-glacier core to tilting/sliding of the rock fragments in the boulder mantle, are treated as effectively viscous deformation (e.g. the stiff boulder mantle with a much higher viscosity). We will carefully discuss the limitations of this continuum approach. We are confident that this simplified model does capture the processes relevant for the conclusions we draw from it. Second, we agree that we have to separate better the assumptions coming from observations on Bleis Marscha from literature knowledge. More details on the reasoning behind the values chosen for specific parameters will be added either in the main text or in the supplemental section. Reviewer 2 mentions sensitivity tests for the numerical modelling parameters. We will check the relevant rock-glacier modelling literature to get a picture of the range of values chosen in these studies, and do sensitivity tests for poorly constrained, uncertain or disputed values.

We hope that we have provided some constructive comments that address the reviewers' concerns and hope to incorporate these into an eventual future revision. Thank you for your consideration and we look forward to receiving your direction regarding the next steps of review.

Kind regards, Dominik Amschwand, Susan Ivy-Ochs, and Olivia Steinemann on behalf of all co-authors

---

## Author Response (AR1)

Authors' response to comments by Jakob Heyman (Referee #1)

Dear Jakob Heyman,
Thank you very much for your careful review of our manuscript. Your detailed and insightful comments led us to profoundly reconsider our interpretation of the exposure ages. Indeed, we became enlightened to the fact that the exposure ages agree well in general with their position on the rock glacier and our image correlation results, when one views them in the framework of the rock glacier still being active. We no longer take the approach that boulder ages measure time since final stabilization. Such an approach is warranted when dating 'dead' landforms (moraines, rock avalanche deposits). Bleis Marscha is moving (albeit parts of it only slowly). We embrace the concept that the nuclide inventory can include bedrock exposure (inheritance in the strict sense), time the boulder lies at the foot of the talus slope, the travel time on the rock glacier, as well as time after final stabilization (if the rock glacier becomes relict). We have revised substantial portions of the manuscript and added a new section to the Discussion (section 5.2) that sets out in detail our refined approach to the ages. With our new interpretation of the ages on the lower lobe the approach for lower and upper lobes is now completely consistent.

Here, we summarily respond to the main issues. Please see our point-by-point responses below.

Point 1: "Several interpretations of the exposure ages appear to be not supported by the data, and for some interpretations the underlying assumptions have not been spelled out. I suggest that you are more open with the fact that the samples may have experienced both prior and incomplete exposure and that it is really difficult to identify how much each of these processes have affected the exposure ages."
Thank you, we completely agree with this statement. We have added a new section to the Discussion (section 5.2) and a new table (Table 4) that gives all details of our approach to the ages in the revised manuscript. We relax our idea of passive transport of the boulders and acknowledge that the exposure ages from single boulders might be biased from both prior and incomplete exposure.

Point 2: "The difference in the interpretation of the exposure ages on the active lobe and the inactive lobe is strange and not supported by the data".
You have correctly pointed out that the relative age dispersion is similar on all lobes, regardless of their age. We now interpret the exposure ages in light of the geomorphology instead of the modern kinematics. In our revised manuscript, the approach for the lower and upper lobes is completely consistent.

We thank you very much again for the detailed and inspiring review of our manuscript.
Dominik Amschwand, Susan Ivy-Ochs and Olivia Steinemann, on behalf of all the co-authors
* * *
Reviewer's comment (in blue)
Reply to comment (in green)
Modification in the revised manuscript (in black)

**I. General Comments**

The paper presents a study of a rock glacier in eastern Switzerland combining cosmogenic dating, recent velocity estimation from aerial imagery, and simulation of the rock glacier deformation in an effort to investigate and reconstruct the post-glacial rock glacier evolution. The questions addressed appear relevant for The Cryosphere and the study presents a novel and interesting concept combining multiple chronological and numerical techniques. The paper is clearly presented and it is mostly easy to understand. I find the paper highly interesting and mostly very good. Thank you.
There are however a number of issues, primarily related to the interpretation of the exposure ages, that should be addressed before publication. In particular, several interpretations of the exposure ages appear to be not supported by the data, and for some interpretations the underlying assumptions have not been spelled out. In the specific comments and technical corrections below these issues (and other more minor issues) are presented in more detail.

Triggered by your comments we completely re-evaluated our interpretation of the exposure ages, especially for the lower elevation lobes. This is discussed in detail in the following paragraphs.

**II. Specific comments**

1. Exposure age interpretations 1. In section 4.2, you write: "The exposure ages in general anticorrelate with elevation and correlate with down-flow distance." I agree with this but an important part here is "in general". A few lines further down you write "The boulders remained exposed at the surface and were passively transported at the rock glacier surface between talus and top of front slope [—]. We conclude that the exposure ages on the active lobe have negligible systematic errors and no overall age shift from inheritance/pre-exposure (systematically 'too old') or nuclide loss/incomplete exposure (systematically 'too young')". This statement is in my mind incorrect and it should be changed.
We agree and deleted this text.

It is clear from Figure 6 that a maximum of three out of seven samples overlap within uncertainties with the linear regression line. This implies that at least four out of the seven samples (plus sample Err8 and Err11 which are not included in the linear regression) have experienced either prior or incomplete exposure. The absolute numbers of the age 'error' are small (<1 ka) but the relative errors are not. For example, sample Err6 has an exposure age that is less than half of the age expected from the linear regression. Such an error cannot be seen as "negligible" if trying to use the age to constrain the velocity of the rock glacier. I suggest that you are more open with the fact that the samples may have experienced both prior and incomplete exposure and that it is really difficult to identify how much each of these processes have affected the exposure ages.

We have completely revised the manuscript and now give a detailed description of how we view the cosmogenic nuclide inventories in the rock glacier boulders.

We agree with Reviewer #1's opinion that measured nuclide concentrations can be impacted by several factors. To clarify our approach, we added a completely new section to the Discussion section 5.2 with an accompanying table that sets out where nuclides build-up in the Bleis Marscha setting and where the concentrations will be lower than expected.

Added to the Results section 4.2:

"In stark contrast to nearly all exposure dating projects whereby the cosmogenic nuclide concentration records the time elapsed since the boulder as reached its final position (on a moraine, in a landslide deposit), we have dated an active, moving landform. This requires a different way of looking at the exposure dates. The cosmogenic nuclide concentrations comprise all the nuclides acquired during exposure of the sampled surfaces. This includes in the bedrock (inheritance), at the talus foot, on the rock glacier surface including while it is moving and at the final boulder position. Effects already discussed in other studies (Heyman, 2011) such as boulder rolling, toppling or spalling can also affect the nuclide concentration."

This excerpt from Discussion section 5.2 summarizes our up-dated approach to deciphering exposure ages on an active rock glacier:

"On an active rock glacier, the cosmogenic nuclide concentration adds up all of the following periods of exposure (Table 4):

1. Pre-exposure in the headwall (bedrock inheritance),
2. Transit time in the talus or during intermediate storage upstream in the talus,
3. Transport on the moving (active) rock glacier (travel time),
4. At the (quasi-)stabilized position (inactive rock glacier or relict rock glacier deposit)."

The associated table is entitled:

"Table 4: Processes that affect measured cosmogenic nuclide concentration in a boulder on an active rock glacier (see text for discussion)."

If assuming that the linear regression actually represents the rock glacier transport an underlying assumption is that prior and incomplete exposure have affected the group of exposure ages by an equal amount. This assumption should be properly spelled out in the text. It is easy to argue that both prior and incomplete exposure may very well have affected the exposure ages. We will likely get prior exposure unless the rock-wall erosion is really deep/rapid and we can easily get incomplete exposure if the rocks on top of the rock glacier move around and topple every now and then. The scatter of exposure

ages around the regression line could perhaps be used to try to estimate an uncertainty of the rock glacier velocity.

As you point out here, it is unlikely that prior and incomplete exposure have affected the exposure ages by an equal amount. We agree with this and as a result of our new view on interpreting the exposure ages we have deleted the associated text.

We argue that rock wall erosion is deep and rapid, which is a cornerstone of our revised age interpretation. The nuclide inventory is made up of time at the talus foot slope and travel time on the rock glacier. It is not impossible that inheritance was acquired in the bedrock wall, but we consider this effect to be small, see text section added here. As suggested by Reviewer #2, we consulted the Crump et al. paper, wherein it was pointed out that large boulders are less likely to contain (bedrock) inheritance. This is explicitly addressed in the text added to the Discussion 5.2 as follows:

"Significant pre-exposure in the cirque headwalls seems unlikely because for talus rock glaciers like Bleis Marscha, fed by scree from the retreating headwall, headwall erosion is rapid (cf. Mohadjer et al., 2020; Steinemann et al., 2020). Back-of-the-envelope estimates of long-term average debris fluxes of the rock glacier lobe IV and V are in the order of 720 (<900) $m^3$ $a^{-1}$, calculated from mean speeds of 400 m/1200 a (length of lobe divided by exposure age), and a rock-glacier width of 120 m (<150 m), a height of 30 m, and a debris fraction of 0.6. These debris fluxes correspond to head-wall erosion rates of ~2.3–2.9 mm $a^{-1}$. At this accumulation rate meters of talus are built-up at the foot slope in decades to centuries as testified by the abundant fresh talus cones within the footprint of the LIA Bleis Marscha cirque glacier (Fig. 5a). With respect to their ice-cored moraine study, Crump et al. (2017) point out that it may be difficult to detect low levels of $^{10}$Be inheritance, but that it becomes unlikely for boulders larger than ca. 1 m side length because of strong self-shielding in the bedrock wall (rapid drop-off of 10Be production with depth)."

We deleted the linear regression because too many statistical assumptions are not met:
- The scatter is not symmetric around the mean as the ages are tendentially skewed towards 'too young',
- The predictor variable, the distance along the profile, is not error free. Off-profile samples are perpendicularly projected onto the profile instead along (unknown) isochrones (which we think is sufficient for visual representation of the samples in Fig. 7).
- The sample group comes from two different morphological units IV and V that possibly moved with different (paleo-)velocities.

2. Exposure age interpretations 2. Further down in section 4.2, you write: "the $^{10}$Be ages of Err2 and Err15 are significantly younger than the further up-slope sampled Err14. It is unlikely that this apparent age inversion reflects internal creep, as the laminar flow behaviour of a rock glacier makes mixing impossible and contradicts our finding of linear relationship between exposure age and down-flow distance on the active lobes". I agree with the statement about samples Err2 and Err15. However, just the same can be said about samples Err6 and Err7 on the active lobe which are younger than samples Err5 and Err9 further up-slope. The difference in the interpretation of the exposure ages on the active lobe and the inactive lobe is strange and not supported by the data, and I think that it should be changed. Thank you for pointing this out. We have revised our interpretation of the ages from units I, II and III. Our approach is now consistent with that of units IV and V. The aforementioned sentence has been deleted. There is scatter in the ages. We address this explicitly in the revised text in Discussion sections 5.1 and 5.2.

3. Exposure age interpretation 3. In section 5.1 (L 517-518) your write about Figure 10: "The exposure ages for the samples located on the presently active upper lobes are slightly less than the travel times by about 10-100 years, which is within the uncertainty margins". This statement is wrong and it should be changed. Of the seven samples falling below the 1:1 line, only one sample (Err5) overlap with the line. You further write: "The measured surface velocity field is representative, and the discrepancy is smaller the closer the samples are to the central flow line (Err3, 4, 5, 10), where the influence of lateral viscous drag is smallest." It is correct that the age discrepancy is smaller for samples 3,4,5,10 compared to samples 6,7,9, but the age discrepancy is still large for samples Err3 and Err4. Sample Err3 has an exposure age of 229 years compared to an estimated travel time of 800 years. Sample Err4 has an

exposure age of 556 years compared to an estimated travel time of 1000 years. The statement about the discrepancy should be changed.

We deleted Figure 10. The interpretation of travel time was too speculative as it depended on the velocities we measured with the image correlation (2003-2012) being a representative average for the whole life of the lobe (~1200 years, Err10). There is no reason that this should a priori be true. Reviewer #2's comments are what made us realize this. We deleted Fig. 10 and related text. As pointed out by Reviewer #2, the nuclide inventories are affected by many factors. Most importantly, the lobe on which a boulder is found can have slowed down or sped up over the whole exposure. We have added a new section in the Discussion that outlines our approach to this in detail. Please also see point 1 above.

4. Exposure age interpretation 4. Further down in section 5.1 (L 529-531), you write: "Our data on the active lobe of Bleis Marscha rock glacier show that the exposure ages for an undisturbed rock-glacier lobe can be addressed as (minimum) travel times. While travelling during the active phase, boulders remain mutually interlocked within the clast-supported framework and the age dispersion remains small. […] Notably, it also shows that pre-travel nuclide build-up, inheritance, and pre-exposure in the rock wall or on the talus are insignificant." These statements are poorly supported by the data and they should be changed. We agree, and deleted this sentence. The age dispersion is NOT small for the question of rock glacier velocity (several hundred years for multiple samples!). As the ages DON'T cluster and overlap with the estimated travel times, there MUST have been some prior and/or incomplete exposure causing the exposure age scatter (unless the scatter comes from the measurements or some other source of error).

As discussed in detail in our revised manuscript, we consider the scatter in ages largely due to post-incorporation processes (see new Table 4 in the Discussion). In our revised manuscript, we interpret the nuclide concentrations to include travel time of the boulder on the rock glacier. Travel time is not inheritance.

See also response to number 3 above. We have deleted the comparison of exposure age to interpreted 'travel time' over the whole life of a boulder, as the latter is much too poorly constrained. The scatter in the ages is due to many factors. We added new text to both the Results and the Discussion dealing with this in detail. We also added an accompanying table (Table 4 in Discussion) that sets out where nuclides build-up in the Bleis Marscha setting and where the concentrations will be lowered by the specified processes. Please also see response to comment 1 above.

5. Exposure age interpretation 5. In lines 550-553 you write: "Intermittent boulder instability during inactivation has been affecting the nuclide inventory of each boulder individually, leading to incomplete exposure by self-shielding and nuclide loss by weathering or spalling, and ultimately to tendentially underestimated true exposure times ('apparent rejuvenation'). This results in a large dispersion of exposure ages and exposure age inversions on the low-elevation rock-glacier lobes." I largely agree with these statements about the exposure ages from the inactive part of the rock glacier. However, we cannot really be sure that NO prior exposure has affected the exposure ages.

As travel time on the rock glacier is included into the nuclide inventory (i.e. travel time is not inheritance), the only inheritance is that acquired in the rock wall. As stated above we believe the evidence supports the idea that rock wall erosion is deep and rapid, and bedrock inheritance (especially for [10]Be) unlikely. Please also see response to point 1 above.

Further, from Figure 10 it is clear that if lifting up the 1:1 line the samples from the inactive part could fit the line much better than the samples on the active part. You never discuss this fact in the manuscript and I think that you should. You could fit a regression line to the inactive samples in the same way as to the active samples and my guess is that you would get a similarly good match to the data. I imagine that such an approach could potentially be used to try to estimate the velocity of the presently inactive part during its active phase (although this will of course involve several questionable assumptions...). In any case, I think you should at least mention more clearly the correlation of exposure age against down-flow distance also on the inactive part of the rock glacier and try to offer some explanation for it.

As stated in response to point 4 above, travel time estimates can be wrong in the sense that it is highly unlikely that current 2003-2012 velocity field is representative over the lifetime of these boulders on the rock glacier. We no longer use this plot to assess prior and/or incomplete exposure because the uncertainties in the travel time are too large. We deleted Fig. 10. Nevertheless, this comment led us to

see that the scatter across all five lobes is similar and that all ages can be interpreted on the same basis, which we have done in the revised manuscript.

6. Exposure age interpretation 6. In lines 564-568 you write: "We find that exposure ages need to be interpreted according to the dynamic history of the sampled rock glacier lobe. On undisturbed lobes where the current surface creep is concordant with the micro-topography, exposure ages represent (minimum) travel times. This applies to active lobes without destabilisation. On 'discordant' lobes where microtopography reflects different stress conditions and must have undergone disruptive dynamic changes, the exposure ages represent time elapsed since the major dynamic change. Inactivation or destabilisation are such disruptive events." I find these statements too categorical and not really well-supported by the data. First, if the statement about undisturbed lobes is correct and the estimated $^{36}$Cl age of sample Err8 is roughly correct, the minimum travel time of the active lobe is significantly longer than what you state elsewhere in the manuscript. Second, if the statement about 'discordant' lobes is correct, the "major dynamic change" must have been in the period 8.9 to 4.8 ka. That is a long period (c. 4000 years) for a "disruptive event".

The revised approach we take on interpretation of our exposure ages agrees well with our image correlation velocity data. Our exposure ages do not represent time since stabilization. The boulders have not finally stabilized as they are on a moving body. We have deleted all text related to 'major dynamic change.'

Third, with data from one specific rock glacier it seems a bit too bold to draw conclusions about exposure dating of rock glaciers in general (which is how I interpret text).

We agree. We have deleted these sentences and our inferences are now more site specific.

7. L 135. "…using the CRONUS-EARTH online calculator". You should state which version of the calculator you have used. Further, I would suggest using the updated version 3 calculator and perhaps also use the Chironico $^{10}$Be reference production rate directly instead of the northeast North America production rate. In the v. 3 calculator you can pick the Chironico input data from the ICE-D database to calibrate the

production rate and then use that to calculate exposure ages. Perhaps you should also exclude the sample with the lowest $^{10}$Be concentration because with that included you get a P-value of only 0.033. However, all this is not really important for the outcome so just disregard it if you don't want to do the extra work.

Thank you for the suggestion. However, ages differ by at most a few percent between the different calculators.

8. Table 1. I suggest changing "calibrated to the 07KNSTD standard" to "calibrated to the S2007N standard" because that is what is stated in section 3.2.

We modified the footnote as follows:

"AMS measurement errors are at 1σ level and include the AMS analytical uncertainties and the error of the subtracted blank. Measured ratios were measured against the in-house standard S2007N which is calibrated to 07KNSTD."

Text and Table 1 footnotes are now consistent.

I also suggest that you add the down-profile distance for each sample that you use in Figure 6 [Fig. 7 in revised manuscript]. That data is not included anywhere else in the manuscript and I think that it should be included somewhere.

We prefer not to adopt this suggestion. In Fig. 6b, sample locations are approximate as the samples are projected orthogonally onto the profile line, not along isochrones, and the transition between rock-glacier root and talus is difficult to pin down. The down-profile distance is not a firm number. Giving distances would impart a deceptive amount of accuracy to such numbers, as additional factors such as lateral drag are difficult to include. Please see also response to point 1 above.

9. L 673-674 "The oldest exposure age (Err11) atop the front of the active upper lobe indicates that the most recent phase of activity phase likely began at or just before 2.8 ± 0.1 ka and lasts until today." This statement is correct only under the assumption that the sample has not experienced prior or incomplete exposure, and that assumption should be clearly expressed.

We have added text to clearly express our assumptions. Our approach to our age interpretation is given briefly in the Results 4.2 and in more detail in Discussion 5.1 and 5.2.

See also lines 751-752 where you write almost exactly the same but "before 2.8" instead of "at or just before 2.8".

This has been corrected. See response to point 1 above.

10. L 756-757. "Furthermore, travel time estimates from streamline interpolations agree well with the exposure ages." This is incorrect – see points 3-4 above.

This sentence was deleted, as was the travel time analysis. See response to above points.

11. L 771-772 "Code availability. The code used for flowline and strain-rate calculations is available by request from the corresponding author." Why not upload the code to a repository and add a link to it or include it as a supplement? That would make it easier for anyone who is interested in the code.

Thank you for this suggestion. We have deleted the strain-rate and streamlines. All other code is readily available in the given references.

**III. Technical corrections**

12. L 13. I suggest changing "The results suggest" to "The results indicate".

This part of the abstract was completely revised.

13. L 65-66. "Long-term effects on rock glacier development must remain unresolved (Kenner and Magnusson, 2017)" This sentence sounds a bit strange to me (in particular "must remain") and I suggest rephrasing it.

We agree that this sentence is (hopefully) incorrect, and was rephrased.

> "Historical records are too short compared to typical rock-glacier lifetimes, activity phases and response periods. To resolve long-term effects on rock glacier development (Kenner and Magnusson, 2017) and to put the present-day morphology reflecting the lifelong dynamic history of active rock glaciers and (relict) rock glacier deposits in a climate-sensitivity context (Frauenfelder and Kääb, 2000), their activity phases need to be placed in a chronological framework."

14. L 68-69. "Cosmogenic radionuclides exposure dating is a unique tool because it directly measures a chronometric, numerical exposure age of the landform surface". I suggest rephrasing this. Exposure dating does not directly measure an exposure age – it measures the cosmogenic nuclide concentration which we can use to calculate an exposure age (involving several critical assumptions).

We agree and rephrased. Additional to the assumptions needed to convert nuclide concentrations to exposure ages, the distinction between the abstract landform surface and the actually exposure dated boulder surfaces on that landform surface is crucial.

> We rephrased to: "Cosmogenic radionuclide concentrations record all periods of exposure of the rock surface to cosmic rays. In principle, they are a suitable tool for deriving numerical exposure ages for boulders on the landform surface."

15. L 74. "at an elevation range of 2400–2700 m a.s.l. Previous studies". I don't know what is correct writing here, but it should perhaps be one more punctuation mark after 'm a.s.l.' to show that the sentence ends.

The second period not required.

16. L 116. I suggest changing "the surface exposure dating method" to "surface exposure dating".

This sentence was absorbed into a road map at the beginning of the methods section (suggestion of reviewer #2), and rephrased as suggested.

17. Figure 2. Disregard these comments if you disagree (they regard the visual impression and are not really important). In figure a, the text with the white edges does not look good so I suggest trying to use just white text or to make the white edges thinner or perhaps bright grey. In figure b, I suggest making the Greek letters 'rho' and 'mu' smaller (they look too big). It also looks a bit strange with having "surface", "boundaries", and "base" underlined.

White edges on text in all figures were removed, font sized unified, and underlining of words removed.

18. Figure 3. I suggest adding units I to V to this map. For the three texts with white background, I suggest trying to make the white edges thinner or perhaps bright grey.
We added the labels with the 5 units (I–V) to the map. We added cosmogenic nuclide sample locations. For better readability, white edges on text in all figures were removed, and font size of contour lines and isotherms were increased.

19. Figure 4. It is a bit strange to have this map shifted 90 degrees compared to Figure 3 and I would probably try to keep the same orientation in both figures (although this is not really important so disregard this comment if you wish). For the black texts with white background, I suggest trying to make the white edges thinner or perhaps bright grey.
We would like to keep the orientation of the Figure as is. White edges on text in all figures were removed.

20. Figure 5. The text in the photos is in several cases a bit difficult to read. I suggest trying to change the black text with white background to just white text or perhaps white text with black or grey background (but thinner background text than in the present version).
We corrected that so the text is now easier to read.

21. L 331 "ages ranging from ~4.8 to ~9.5 ka". The age 9.5 ka here seems to be too high as the oldest age is 8.9 ka.
We changed it to 8.9 ka.

22. L 504-505. "The active high-elevation lobes (2550–2700 m a.s.l.) with surface speeds up to 60 cm a$^{-1}$ display clustered exposure ages within $0.23 \pm 0.04$ to $2.8 \pm 0.1$ ka". The exposure ages on the active lobe should NOT be called "clustered". See major points 1 and 2 above.
We agree and deleted the word 'clustered'.

23. Figure 10. You state in the figure caption that the exposure age uncertainties are the "analytical uncertainties (internal errors)". Although it won't really matter for the outcome, you should rather use the external uncertainties including production rate uncertainty as you are comparing the exposure ages with travel time estimates that are independent from the exposure ages.
We now show the external errors.

Crump, S. E., Miller, G. H., and Anderson, R. S.: Interpreting exposure ages from ice-cored moraines: a Neoglacial case study on Baffin Island, Arctic Canada, J Quaternary Sci, 32, 1049–1062, https://doi.org/10.1002/jqs.2979, 2017.

Authors' response to comments by Leif Anderson (Referee #2)

Dear Leif Anderson,
Thank you very much for your careful and extensive review of our manuscript. We address your comments concerning (i) manuscript organization, (ii) exposure age interpretation, (iii) image correlation, and (iv) finite-element modelling in detail below. The issues related to the interpretation of the exposure ages have also been brought up by Reviewer #1 in more detail. We apologize that we respond to these issues in more detail in his reply. Please see responses to Reviewer #1.

In the revised version of the manuscript, we have addressed your comments. Please see our point-wise replies below. The major changes regarding your suggestions are briefly outlined here:

- Consistent Bleis Marscha subdivision imposed throughout the manuscript and figures, based on morphological rock-glacier subdivisions (units I–IV). Sect. 4.1, morphological observations, rewritten to agree with discussion of exposure ages (upstream, forward in time in terms of lobe succession).
- Reassessment of the exposure age interpretation (also brought up by Reviewer #1), now closely connected to the morphology (added Sects. 5.1–5.2, additional panel figure showing the sampled boulders relevant to the discussion, new table with exposure age interpretation approach).
- Reframing of the contribution by finite-element modelling, more closely connected to the kinematic part: Separation of topographic from material control on surface creep rate, rather than semi-qualitative ground-ice content estimates; shortening of discussion (Sect. 4.4).
- Clarifications concerning mechanical rock-glacier layering and associated deformation mechanisms, clearer separation of assumptions from local observations.
- Shortening of the image correlation sections, deletion of obsolete paragraphs on strain rates and flow lines.
- Reassessment of the uncertainty of the image correlation: adoption of 15 cm a$^{-1}$ as a conservative level of detection based on adjacent off-rock glacier velocities (as suggested).
- Addition of a short paragraph on mapping (in Sect. 3.1).
- Addition of a road map to direct the reader to the diverse methods (beginning of Sect. 3).
- Cleaning up of all the figures (Fig. 8a, 12 deleted; Fig. 9 (10 in revised manuscript) graphically condensed).

We thank you very much again for the detailed and inspiring review of our manuscript.
Dominik Amschwand, Susan Ivy-Ochs and Olivia Steinemann on behalf of all the co-authors
* * *
Reviewer's comment
Answer to comment
Modification in the revised manuscript

The manuscript considers both the modern and past states of Bleis Marscha rock glacier in Switzerland. The authors use field observations, surface velocity estimates, finite-element modeling, and surface exposure ages to explore the evolution of the rock glacier. Ultimately the authors use their datasets to interpret the Holocene history of the rock glacier as related to climate and erosion rate from the headwall.

**I. Manuscript organization, general comments**

This is truly an exceptional amount of work and it is really an ambitious project and manuscript! The authors should be proud of this achievement. I am especially impressed with the bringing together of the methods of a glaciologist/physicist with the methods of a geomorphologist/paleoclimatologist.
We thank you for this encouraging comment!
The comments below are mostly related to the presentation of the work, with some suggestions which could improve the analysis.

It would be great if the surface velocity estimates, the field observations, and the model could be better used to justify the interpretation of the surface exposure ages. Right now, the manuscript seems to touch on these different features and then transition into the Holocene history of the Bleis Marscha rock glacier without too clear of a connection between the modern and paleo perspectives. It would better honor all the work in the manuscript if logic behind the assumptions being made was laid out in connection to the modern analysis of the rock glacier. This is no easy task, but I think one that will really highlight the broad scope of methods brought to bear in this body of work.

We see that the manuscript was too long and that the reader might have lost track with different rock glacier surface units. We only use one unit designation now consistently throughout the manuscript (units I-V). We shortened the methods and result sections by condensing the text instead of moving it into a supplementary section. On the other hand, we added roadmaps to clarify how the different methods and results are related to each other. As a substantial addition, we spelled out the surface age model (how boulders are thought to move downstream on the surface etc.). These major changes are outlined above.

I think that the uncertainty in the surface velocity estimates could re-done using off-rock glacier velocities adjacent to the rock glacier itself. Right now, I suspect that the surface velocity error is too small. Comments below explain this more fully.

Yes, we agree, cf. our reply to lines 163-165 and 386. In the revised manuscript, we adopted a much higher, more robust level of detection of 15 cm/a. Figure 8 has been modified accordingly.

The modeling analyses could be improved with some sensitivity tests exploring assumed parameters and the ice-rock ratio but I am not sure they are necessary. Really a clearer statement of which parameters are assumed and how those parameters where chosen is enough.

We separated field observations from assumptions more carefully. We however did not add sensitivity tests.

I suggest that the authors go through the full manuscript with a discerning eye for which observations and analyses are really needed to support the main take homes from the manuscript. There are various classifications provided of the rock glacier surface units and surface zones on the glacier that do not coincide. These multiple classifications are difficult to keep straight so some simplification would be helpful for the reader.

We only kept the morphological rock-glacier subdivisions (units I-IV) for clarity.

As can be seen in the detailed comments below I suggest in some cases for material to be moved into a supplemental section and for some figures to be simplified. Perhaps figures 4 and 7 could be combined into a 3-panel figure? The text should be simplified to improve the reader's experience and their ability to access the science. Terminology needs to be used consistently throughout the manuscript especially regarding section titles from the methods, to the results and discussion.

We deleted panel (b) of Fig. 7, but decided not to combine Fig. 4 and Fig. 7 to keep the order as they are mentioned in the text. We simplified terminology (consistent use, neutral instead of interpretative terms).

**II. Line-by-line comments**

Line 11. "2003/2011" is unclear to me. Perhaps note that repeated surface velocities measurements were made.

Line 12. I am not sure what "orthophoto orientation correlation" means here.

"…horizontal surface creep rate quantification by correlating two orthophotos from 2003 and 2012".

Line 19. Consider rewording as I am not completely sure what is meant. "Nuclide loss from boulder erosion, affecting the nuclide inventory of boulders independently"

This part of the abstract was completely revised. We deleted the terms 'nuclide loss'.

Line 35-50. You might emphasize the considerable differences in the timescale between these two concepts. ['self-preservation'/climate resilience vs. rapid response and destabilization]

Yes, we agree that the timescales are different, millennia versus decades. However, we hypothesize that short-lived events like boulder mantle destabilization can rapidly change the morphology of a rock glacier in a way that irreversibly alters the subsequent evolution (e.g. rupture of insulating boulder mantle that leads to rapid degradation while a nearby intact rock glacier maybe just accelerates due to warming). Such disruptive events may also leave behind long-lasting traces (e.g. scarps, patchy lichen cover). We have replaced "contradicting" with "different", as it is not strictly contradicting but relates to different time scales.

> "However, in the literature, different views on the climate sensitivity of…"

Line 55. I am not aware of 'dynamic inactivation' is there a citation for this term? It confuses me a bit because I tend to think of 'dynamics' as the flow of the rock glacier body and not processes related to changing headwall erosion rate to the rock glacier.

We deleted this paragraph (terminology came from Barsch, 1996).

Line 63. It is not clear what 'interactions with glaciers' refers to.

We have the situation in the Bleis Marscha cirque in mind where the rock glacier intermittently coexisted with a cirque glacier. 'Interactions with glaciers' refers to the glacier's impact on ground thermal condition (insulating effect of the ice), on debris supply (direct connection to talus intermittently lost, debris supply via glacier?), and on stresses/forces (overriding or pushing of the advancing glacier ice). For clarity, we have included these examples in the revised text.

> We added concrete examples and changed the phrasing to "…and interactions with glaciers (e.g. pushing ice, altered thermal conditions/insulation, glacial debris sources)"

Line 65-66. Consider revising to make the meaning clearer. Could new methods/approaches help resolve this issue though? [Also raised by reviewer #1] ["Long-term effects of climate variability on rock glacier development must remain unresolved."]

We agree that this sentence is (hopefully) incorrect, and was rephrased.

> "Historical records are too short compared to typical rock-glacier lifetimes, activity phases and response periods. To resolve long-term effects on rock glacier development (Kenner and Magnusson, 2017) and to put the present-day morphology reflecting the lifelong dynamic history of active rock glaciers and (relict) rock glacier deposits in a climate-sensitivity context (Frauenfelder and Kääb, 2000), their activity phases need to be placed in a chronological framework."

Line 73. Why this rock glacier? A sentence about why you chose this rock glacier would resolve this.

Its multiple, stacked lobes immediately suggested that the history of Bleis Marscha could span millennia. This impression was corroborated by relative Schmidt-hammer dating by Frauenfelder et al. (2005) that gave mean R-values as low as 42 at the lowermost front, a value that typically points to early Holocene (cf. Fig. 4 in Böhlert et al., 2011b).

> "Previous relative dating studies based on Schmidt-hammer rebound values as well as thickness and chemical composition of weathering rinds suggest a Holocene-long development (Frauenfelder et al., 2001; 2005; Laustela et al., 2003)."

Line 78. instead of 'present' maybe use 'modern'.

We changed the wording accordingly.

Line 106-7. The materials and methods section would benefit from a bit more generalized and expanded text here more broadly introducing the methods. A road map into the diverse methods applied in this study would help. This is easily fixed!

We agree that the readability benefits from such a road map and added it in the revised manuscript.

> We added: "Topographic maps, aerial photographs, digital terrain models, geomorphological field mapping, surface-exposure dating, and finite-element (FE) modelling form the data basis to reconstruct the Bleis Marscha morphodynamic development. The morphology of rock glaciers, where processes and form are intrinsically linked via deformation by creep, largely

preserves the cumulative deformation history over the lifetime of the landform (cf. Frauenfelder and Kääb, 2000). Concretely, the concept of periglacial rock-glacier formation and the 'conveyor-belt' advance mechanism form the theoretical groundwork on how surface boulders move compared to the rock glacier as a whole. The subdivision in geomorphologically defined units of the polymorphic (sensu Frauenfelder and Kääb, 2000) Bleis Marscha, assisted with an estimate of modern surface creep rates, provides a framework for the discussion and interpretation of the exposure ages. Surface exposure ages of 15 boulders of the rock-glacier deposit were determined with the cosmogenic radionuclides $^{10}$Be and $^{36}$Cl. FE modelling relates the surface movement to semi-quantitative ground-ice estimates at depth."

Line 109. Perhaps a bit more detail about the mapping performed? What sort of mapping did you conduct? For what purpose was the mapping conducted?
Field mapping provided morphological information on the rock glacier, the number and succession of lobes and an idea on number and relative age of activity phases. This is an indispensable framework for interpreting the exposure ages. We added a corresponding paragraph in Section 3.1.

Line 115. It would be nice to have a map of the rock glacier with the locations of CRN sampled boulders referenced in this section if not before this section.
We added the sampling locations in the geomorphological map.

Line 116. Expand or combine this paragraph as it is only one sentence. This sentence itself can be simplified as well.
This sentence is absorbed into the road map and deleted here.

Line 146-147. How were these erosion rate values chosen? How much does these assumed erosion rates effect your exposure ages and your conclusions? Perhaps a few extra sentences would help here as well as some citations to support the assumption.
We briefly discuss the erosion-rate uncertainty in Section 4.2 in the revised manuscript. The rate of 1 mm/ka comes from André (2002), cited in Steinemann et al. (2020). According to André (2002), rates of granular disintegration are independent of climate conditions, i.e. her values found in Lapland are applicable in the Alps. The effect on the exposure ages is anyway very small. For example, using 3 mm/ka as in Böhlert et al. (2011a) for the same lithology (our sampled 'Err granodiorite' is identical to their 'Julier granite'), we get a maximum deviation of 135 years (1.5%) for the oldest age of 8948 a. This is well within the methodological age uncertainties.

Line 150. It might be easier for the reader if the heading here is 'surface velocity' or 'surface creep rate estimates.'
We changed the heading to "Estimation of modern surface creep rates".

Line 152. Reword 'The used'
We deleted the word.

Line 163-165. "We estimate the uncertainty by correlating a reference area in the valley floor considered as stable." I suspect that this will underestimate the error on the rock glacier because the area. Is there reason to expect that the off-rock glacier areas in figure 7a are moving at 10 cm/a? It seems to me that a more robust estimate of the error of the surface velocities would come from the large off-rock glacier areas in figure 7a that show velocities up to 20 cm/a. This is also consistent with the lack of correlation between slope and velocity on the rock glacier in figure 7b.
We thank you for your suggestion, and agree that our previous approach might have underestimated pre-processing or orthorectification errors from topographical distortions. A flat area might not be a robust enough reference. We now take the presumably stable, adjacent off-rock glacier areas as the reference, which brings about much more conservative level of detection of 15 cm/a.

> "The modal displacement of the presumably stable adjacent off-rock glacier areas defined the significance level, that is the threshold below which any measured displacement is not distinguishable from immobility."

Line 166-177. These paragraphs are interesting and well written but I am not sure why they are being included. How does calculation of strain rates relate to the larger framework of the manuscript? Maybe a few sentences of introduction to the section at line 151 could provide a road map for the calculations made related to the surface velocity/creep estimates.
We deleted the strain rates and the streamlines as they are no longer necessary.

183-4. Is this the only process by which rock glaciers move [by gravity-driven steady-state creep of its ice-rich core]? How about translation along shear zones/sliding? Does it make sense to state that you assume that movement of this rock glacier occurs by internal deformation?
We thank you for this enlightening question. There are multiple deformation mechanisms related to the composition of the layers, depth, and water content. They are concisely outlined in the recent contribution by Cicoira et al. (2020).
The contribution of the boulder mantle to total deformation is negligible. The contribution of the shear layer is important (60–90% of total deformation). Its higher deformation susceptibility also arises from liquid water at grain boundaries, more related to grain size and pore pressure than ice content. This is especially the case for 'temperate' permafrost close to its melting point, which we must assume for the lobes below the -2°C MAAT isotherm. We accounted this weakening by a lower viscosity of the shear layer. While we do not seek to accurately reproduce the deformation mechanism at a process level, we are confident that we can simulate the displacement for the entire rock glacier to a first order. Our approach is similar to previous modelling studies (e.g. Müller et al., 2016) and yields deformation profiles in the "synthetic boreholes" that resemble observed ones. As a consequence, we reframed the contribution of the FE modelling: The goal is to disentangle topographical (slope, thickness) from material control on observed surface creep rates, not to infer ice content.

202-204. It seems you should state clearly that you assume that this 3 layer structure applies to Bleis Marscha rock glacier.
We agree. The boulder mantle is observed at the (inner) fronts, exposing the shallow part of the stratigraphy, but the deeper basal low-viscosity layer is not. We stated now more clearly that no boreholes have been drilled on Bleis Marscha rock glacier (currently only later in line 452) and separate assumptions from observations more clearly. Cf. reply for lines 209-10 below.

209-10. If you give a few general observations here the reader does not need to search for the justification in the later section.
We added these hints, and formulated it more cautiously: Strictly, we observed the upper two layers, boulder mantle over rock-glacier core. The basal shear layer is assumed based on the general knowledge about rock glaciers (see previous comment).

> "Over-steepened terrain steps, exposing the uppermost few meters of the stratigraphy, show the coarse boulder mantle over the finer-grained rock-glacier core (cf. field observations in Section 4.1). Although no borehole deformation data is available for Bleis Marscha that could evidence the basal shear layer, we are confident that this typical rock-glacier feature does exist there."

211-12. "a 3 m thick basal low-viscosity shear zone (constant)." Does this not also contribute to the movement of the rock glacier? How is the viscosity in the low-viscosity shear zone constrained? If this low viscosity portion of the rock glacier is included here then it seems it should be discussed above where you mention processes leading to rock glacier movement.
Furthermore, it seems that the actual values of effective viscosity produced by the model are highly dependent on the assumed relationship between the viscosity and the viscosity in the shear layer at the base of the rock glacier. How important is the assumption that the shear layer contains a viscosity 10% of the rest of the rock-ice mixture? I don't think this has a large bearing on the main results of the manuscript though.
Deformation in the basal low-viscosity shear zone contributes substantially to the overall deformation. I realized that the text in line 183 was incorrect, it should read "creep of its ice rich interior" that includes the shear layer, not only the "ice-rich core". This is corrected by rephrasing the paragraph on the mechanical layering, together with a more suitable reference (Müller et al., 2016, instead of Arenson et

al., 2016) and a clearer statement on which rock glacier parts contribute to deformation. We re-organized the paragraph on lines 203-209 accordingly.

In fact, in our model, the basal shear layer accommodates 60% of total modelled displacement, a value that agrees with borehole deformation measurements on other rock glaciers (60-90%, Cicoira et al., 2020). Apart from surface slope (known from terrain model) and the thickness (reasonably assessable), the viscosity ratio between core and low-viscosity shear zone has the largest influence on surface velocity. We assume a viscosity ratio of 10, a reasonable number derived from the literature (Cicoira et al., 2020).

As we compare the viscosity along the rock glacier only in a relative sense, the actual viscosity contrast is not critical. Critical is the assumption that the contrast does not vary along the rock glacier, i.e. in lobes of very different age. We argue that the weakening of this basal layer is largely owed to finer grain sizes and not to higher ice contents (which may or may not change throughout a rock-glacier's history), so that even shear layers from millennia-old rock-glacier lobes retain the weakening effect.

> "The first-order deformation of rock glaciers is governed by gravity-driven steady-state creep of its ice-bonded interior (Müller et al., 2016)."
>
> Lines 203-209: "Direct evidence from over-steepened rock glacier fronts, borehole deformation measurements (Arenson et al., 2002) and indirect geophysical investigations (Springman et al., 2012) suggest a pronounced thermo-mechanical layering of rock glaciers. A robust finding is that the deforming part can be divided onto a sequence of three layers, surface boulder mantle, rock-glacier core, and shear layer (Haeberli et al., 2006; Frehner et al, 2015; Cicoira et al., 2020). (1) The seasonally frozen ice-free/poor surface layer consists of a matrix-poor, clast-supported framework of large, interlocked boulders (~active layer). Discrete movements of blocks in the boulder mantle are negligible compared to the total surface movement (but not with respect of inventories of cosmogenic nuclides, cf. Section 5.1). Deformation is accommodated by the ice-bearing interior that comprises the rock-glacier core and the basal shear layer. (2) The ice-rich rock glacier core consists of a of a perennially frozen mixture of ice, debris, and fine material. (3) A fine-grained, few meters thick shear layer concentrates 60-90% of the total displacement. Its higher deformation susceptibility, arising from the weakening effect of liquid water at grain boundaries, is in the model accounted by a lower viscosity. Boulder mantle, core and shear layer lie on an immobile substratum of debris or bedrock."

215. What is the Salteras terrace? Is it composed of river gravel or bedrock or till?

The Salteras terrace is composed of glacial sediment. We used the term 'terrace' loosely, referring to the gently sloping surface of the trough-shoulder in a typical glacial valley. To avoid misinterpretation, we changed the wording accordingly.

> We changed the wording to 'Salteras trough shoulder'.

219-225. Is there a local justification for the 60% ice by volume? If not, this should be stated as an assumption. Or further down a sensitivity test should be shown to highlight how much your results depend on this assumption.

No, there is no local justification for the 60% ice content. The number is a typical value from the literature also used in other modelling studies (e.g. Müller et al., 2010). Sensitivity tests showed that creep rates are not very sensitive to density. Plausible density variations are small (30-90%vol ice, range of 2163-1089 kg m$^{-3}$) compared to viscosity variations (orders of magnitude).

Section 4.1 [Field observations/geomorphology]. It would help the reader if the approach to mapping was outlined in the methods section. Right now, there is scant mention of mapping methods, despite a rather large results section dedicated to it.

While there are valuable observations from the field here. I find that the section contains a lot of details that I am not sure how they connect to the rest of the exciting work presented in this manuscript. Perhaps it could be simplified and only the most necessary observations included. Other additional observations could be moved to the supplemental materials.

We added a paragraph that outlines the mapping methods (in Section. 3.1). We think that the field observations are necessary for the interpretation of the exposure ages. Instead of moving parts of this text into a supplemental section, we re-organized the section (now going upstream, as we discuss the

exposure ages) and concentrated on observations that are used later on in the manuscript to distinguish the different lobes (Section 5.1).

246. It would help the reader if you simplified the section title here.
We simplified the section title to '4 Results and interpretation'.

250. Maybe describe what the estimate of volume was based on (i.e. what was the assumed mean thickness)? Also, I am not sure 'Internal' is needed here.
We assumed a mean thickness of 30-40 m.
> "Based on an assumed mean thickness of 30-40 m, we estimate a total volume of $\sim(7–10) \times 10^6$ $m^3$."

258. I am not sure what 'well-localized' means.
We want to emphasize that the Err Granodiorite abundantly found on the rock-glacier surface is outcropping only in the back of the Bleis Marscha cirque. This is important for estimating the length of the old-generation rock glacier and its interactions with the cirque glaciers. Cf. our reply for line 261-2.

261-2. I am not sure I totally follow the reasoning here. It would help to spell it out more clearly.
This is the same line of arguments as for line 258, and hopefully spelled out more clearly:
> "All lobes share the same source of Err Granodiorite, located in the back of the cirque. As no (large enough) cirque glacier likely existed throughout the Holocene to move ice and debris out of the cirque, all lobes must have been directly connected to the source of Err Granodiorite in the Bleis Marscha cirque. The lower lobes must extend further upslope to the talus, i.e. their uppermost segments are buried beneath the now-active, high elevation lobes."

266-69. If the ice patch was not flowing then it is not clear why geomorphological evidence in the landscape would be expected.
We agree and rephrased.
> "We observed neither signs of glacier activity nor any surface ice remaining from the recent glaciation of the cirque (1853 Dufour, 1887 Siegfried maps: Dufour, 1853; Siegfried, 1887). The "small ice patch" mentioned by Frauenfelder et al. (2005) was no longer visible in 2017."

293. Is there data to support this observation? [of ice loss in the southernmost fringe of unit I]
The argument is based on our morphological observation and kinematic data. The reasoning is now spelled out more clearly.
> "The southernmost fringe of the rock glacier is slowly subsiding. Debris slides or aprons do not follow along the entire scarp. The slow, interior deformation must originate at depth rather than at shallow levels, what would push loose surface debris over the edge. The body is sagging instead of laterally advancing, for which ice loss is a possible explanation."

Later in the manuscript, where kinematic and FE-modelling results becomes available, we expand this argument further: The external parts of unit I move more slowly than potentially possible given the comparable surface slope and thickness as unit II, with outwards gradually decreasing velocities instead of sharp lateral contrasts (like shear margins) as observed on the 'healthy' unit II.

298-9. It is not immediately clear why this is calculated or how it ties into the rest of the manuscript.
We use buckle-folding theory to constrain the viscosity ratio between boulder mantle and rock glacier core. We moved this paragraph to the FEM methods (Sect. 3.4).

351-2. Here I think you need to describe what those processes are. Boulder rolling seems like an important potential process on rock glaciers. You might see Crump et al., 2017 as well.
Thank you for this literature suggestion, which did provide additional insight to interpreting exposure ages on non-stable landforms. We revised this section thoroughly, and added substantial content where we discuss these various possible impacts on nuclide concentrations in boulder surfaces in detail (Sects. 5.1, 5.2).

380. Simplifying the section heading will benefit the readability ['2003/2012 surface creep rates']

The section heading now reads "Modern surface creep rates".

386. What is the significance level based on? It seems that much of the off-rock glacier area in Fig. 7 is moving up to 10-15 cm/a. This makes me think that the error associated with the surface velocities should be higher.
The significance level was based on the modal velocity from the stable area. We agree with your concerns and raised the significance level to 15 cm/a based on the off-rock glacier areas. Please see our reply to line 163-165.

> "As considerable presumably stable adjacent off-rock glacier areas show apparent surface movements of up to 15 cm a$^{-1}$, we adopt this conservative value as the significance level. Areas with speeds lower than this level of detection are classified as non-moving (shaded areas in Fig. 7)."

390-412. It is hard for me to keep track of the different lobes as well as the newly presented creep rates here. Perhaps this section can be synthesized a bit more.
We acknowledge that the multiple classification of the rock-glacier surface is confusing. In the revised manuscript, we no longer refer to the kinematic classification and use only the geomorphological one (units I–V). Also, the image-correlation results are synthesized, and parts deleted.

Section 4.4 This section [discussion of FEM] could be simplified and maybe extra text moved into the supplemental section.
We synthesized this section and shortened lengthy discussions about forward and inverse problem uncertainties. The message is now that the deformation of units II–IV is topographically controlled rather than by material parameters (viscosity), which ties more closely to what we found from the slope-creep rate scatter plot (Section 4.3).

503-5. How can you be sure that these [velocity] jumps aren't just associated with the steepening slopes at these lobe boundaries?
This is an important question, thank you! The concept of a link between processes and form is central. You point out correctly that many velocity jumps seem to be associated with the local steep slope, because the speed decreases immediately upslope with decreasing slope. There are also 'inactive' terrain steps without a velocity jump, e.g. at 600 m (Fig. 10b), right below sample Err12 (belonging to the same mid-Holocene lobe, no 'time gap'). We agree that the sentence was overly generic. However, each of the units III and IV seem to as a whole, with sharp lateral velocity jumps to the adjacent immobile terrain. These lobes have overall higher speeds than thee downstream next one, i.e. are overriding each other. This is what we think these frontal velocity jumps are related to, what is also visible in the field as terrain steps with toppling boulders. We see this also on the lowermost front (unit I), where the presented kinematic data alone are not significant (below detection limit). We now specifically refer to the separations between the three lobes, where morphological and kinematic discontinuities do coincide.

> "Coinciding morphological discontinuities (steep terrain steps, Fig. 4), 'time gaps', and kinematic discontinuities (velocity jumps, data gaps; Fig. 8) separate units I–II, III, and IV–V from each other. We interpret that Bleis Marscha is not a continuous 'stream', but a stack of three overriding lobes, each with its own, discrete formation phase."

547-549. This suggests to me that these velocities are not the result of active flow but rather the motion of boulders due to surface processes, shadows, and spurious correlation. As suggested above I think the error uncertainties for the velocity estimates should be redone.
This part of the manuscript was thoroughly revised, and the uncertainties reassessed.

558. Maybe 'preserves' instead of 'memorizes'
We 'de-humanized' the wording in the revised manuscript.

Line 560. "The exposure ages are rather inactivity or stabilisation ages than travel time estimates, as previously reported (Moran et al., 2016; Steinemann et al., 2020)." I don't understand what is meant here.

We revised the exposure age interpretation scheme. This part of the manuscript was thoroughly revised. The same applies to lines 560-568.

560-63. I do not understand how the travel time is within the uncertainty of the exposure age on the lower part of the rock glacier. Maybe the travel time constitutes half of the exposure age for Err12 and 13, but certainty not the other samples.
564-70. This paragraph is hard for me to follow [the interpretation scheme].
566-68. I do not understand how travel time can be neglected in this case. Perhaps the logic can be laid out more here.
Thank you, you are correct, travel time is indeed included. We have revised our interpretation of the exposure ages which is now completely aligned with the image correlation results. We apologize that we do not repeat the detailed answer given in response to Reviewer #1 who brought up the same issues as you did. Your concerns were justified and led us to thoroughly revise the manuscript with respect to the exposure age interpretation. Please see also our detailed response to Reviewer #1.

599-602. I suggest that you state that these are 'back-of-the-envelope' estimates as a lot of assumptions go into them.
Yes, these are very rough estimates, and we made this clearer in the text.

Conclusions: It would be good to see a bit more incorporation of the results from the velocities and model with the paleoclimate story.
We added sections (5.1, 5.2) and several minor changes to show the links better, also in connection with the figures and photos. We shortened digressing discussions of the kinematic and modelling results. We completely rewrote the discussion chapter to present a significantly improved integration of results from all methods. Paleoclimatic paragraphs are slightly shortened and rearranged to set different development phases of the Bleis Marscha in context, to which an entire section '5.3 Implications' is devoted to. In the conclusions, the climate history and development of Bleis Marscha are now better interwoven.

Table 1. How sensitive are these results to the assumed surface erosion rate of the rock samples?
The exposure ages are not sensitive to slow weathering. Tripling the erosion rate changes the exposure ages by <2%. Cf. our reply to lines 146–147.

Tables 2 and 3. Perhaps move these tables [$^{36}$Cl data from Err8] to the supplemental as the sample is assumed to be an outlier.
No, we prefer to keep all the information required to calculate chlorine age in main manuscript.

Figure 1. Very nice map and inset of Switzerland.
Thank you.

Figure 2. Panel (b), the '5 m' and '2 m' labels are for the boulder mantle and basal shear layer but that is not clear in the figure. The 0.1 x viscosity in the basal shear layer should be discussed in the methods portion of the manuscript and described as an assumed value.
We clarified the figure caption. The '0.1x viscosity' is now clearly stated as an assumption from the literature, with Cicoira et al. (2020) as the reference. Cf. our reply to comment for lines 211-12.

Caption: what is the Salteras terrace? Maybe reference it as the lower geomorphological surface?
Please see our reply to comment for line 215.

Figure 4. Making the fill less transparent for the CRN ages would improve legibility, as well as making the boxes around the CRN ages tighter. What does 'active high-elevation lobes' refer to? I do not see any active lobate features.
We cleaned this figure as suggested, tightened and scaled down the boxes that covered a part of the rock glacier, deleted obsolete labels, and report external exposure age errors (rather than internal ones).

Figure 5 [panorama images]. I find the caption difficult to follow. There is a lot of information here, which is great, but I am not sure how it ties into the broader manuscript. Perhaps it could be moved into a supplemental section because it adds good background info.

Instead of moving the panorama images into a supplemental section, we hope that we could tie it better to the text (Section 5.1). These figures are needed to portray the terrain steps that are key features that define the different lobes. We shortened the figure caption.

Figure 6 [longitudinal transect]. This figure is a good synthesis of the different datasets produced. But I think the legibility of the figure can be substantially improved. Maybe the vertical dashed lines do not need to extend across the full height of the figure. It might instead work well to move this figure to the supplemental and then just include the exposure ages and the surface profile as a figure in the main text? It seems like the local topography and thrust activity are secondary controls that complicate the figure.

No, we do think that the local topography and thrust activity (wording changed to 'kinematic discontinuity', see below caption Fig. 6) is necessary background to interpret the exposure ages. We tried to carve out these links in the revised text with a newly added section (Section 5.1). For the rest, we cleaned up the figure as suggested.

From the caption of Fig. 6: "This suggests that pre-travel nuclide concentrations are negligible." maybe add 'typically' in front of 'negligible.'

Yes, we agree that individual boulders can have inheritance (Err8). We deleted the linear regression and this sentence.

From the caption of Fig. 6: "Active thrusts coincide with sharp velocity gradients (cf. Fig. 7); this differential movement results in overriding lobes." To me the assertion that the front of lobes can be positively linked to active thrusts is an interpretation here and throughout the manuscript.

We agree that since we do not have direct data on the Bleis Marscha interior, we do not know the exact nature of the lobe separations at depth. Instead of the tectonic term 'thrust' (now reserved for the thrust in the headwall), we now use the neutral terms 'morphological discontinuity', 'terrain step', or 'kinematic discontinuity'. We also formulated it more specifically, cf. our response on lines 503-5 above. The link between morphologically separated lobes of different ages with their own surface motion pattern is an important concept in our interpretation. In the revised manuscript, we discuss this close relationship between process and form explicitly (Sections 3.2, 5.1).

Figure 7. Panel (a) The blue dots are not explained in the caption. The pink is hard to see. It seems that much of the off rock glacier area also produces significant velocities. Is this real motion?

Panel (b) the principle strain rates are very hard to read. Consider reducing the number of plotted strain rates (same for Panel b arrows) or creating a raster of dominant compression versus extension areas of the rock glacier.

We adopted a more conservative level-of-detection threshold of 15 cm/a; cf. our reply to comment on lines 163-165. Virtually all off-rock glacier areas are then below the level of detection. We updated the caption of panel (a) accordingly. Areas below the level of detection are dark shaded, uncorrelated areas (gaps) are transparent. A legend for the blue dots (sample locations) is added. We deleted the strain-rate plot (panel b) because it is no longer necessary for our interpretation.

Figure 8. The colors between the panels should match otherwise it is very hard to read. It seems that velocities. Panel (a) based on the histogram up to 20 cm/a. Below 20 cm/a there does not seem to be a positive correlation between velocity and surface slope. Based on the velocities from off the rock glacier of up to 15 cm/a does this not indicated that the below ~20 cm/a the velocities could be noise?

We deleted the now needless panel (a).

We agree that the level of detection is 15 cm/a, see comments above. The loss of correlation between surface slope and creep rate concerns the uppermost and lowermost parts of the rock glacier. In the root zone, viscous creep is possibly not fully developed. In the lowermost frontal area (lowermost ~100 m), incipient stabilization and degradation hampers creep. This agrees with field observations.

Either use only 'surface velocity' or only 'surface creep rate' throughout the manuscript.

We ensured consistent wording throughout the revised manuscript: 'creep rate' whenever it refers to the measured quantity, 'velocity' when it refers to abstract, synthetic data from the finite-element modelling.

Figure 9. Lots of great information here but I would suggest just including the lower panel.
We think that the upper panel nicely shows the FE-modelling results, gives a more intuitive impression of the model domain, and shows the "virtual boreholes". We prefer to keep both panels, but made the figure graphically tighter, e.g. by removing unnecessary legends.

Figure 12. I would suggest that this figure be moved into a supplemental section.
We deleted figure 12.

André, M.-F.: Rates of postglacial rock weathering on glacially scoured outcrops (Abisko-Riksgränsen area, 68°N). Geogr. Ann. Phys. Geogr. 84 (3/4), 139-150. https://doi.org/10.1111/j.0435-3676.2002.00168.x, 2002.

Barsch, D.: Rockglaciers. Indicators for the Present and Former Geoecology in High Mountain Environments, Springer series in physical environment vol. 16, Springer, Berlin, Heidelberg, https://doi.org/10.1007/978-3-642-80093-1, 1996

Böhlert, R., Compeer, M., Egli, M., Brandová, D., Maisch, M., W Kubik, P., and Haeberli, W.: A combination of relative-numerical dating methods indicates two high Alpine rock glacier activity phases after the glacier advance of the Younger Dryas, The Open Geography Journal, 4, 115–130, https://doi.org/10.2174/1874923201003010115, 2011a.

Böhlert, R., Egli, M., Maisch, M., Brandová, D., Ivy-Ochs, S., Kubik, P. W., and Haeberli, W.: Application of a combination of dating techniques to reconstruct the Lateglacial and early Holocene landscape history of the Albula region (eastern Switzerland), Geomorphology, 127, 1–13, https://doi.org/10.1016/j.geomorph.2010.10.034, 2011b.

Cicoira, A., Marcer, M., Gärtner-Roer, I., Bodin, X., Arenson, L., and Vieli, A.: A general theory of rock glacier creep based on in-situ and remote sensing observations, Permafrost Periglac, 1–13, https://doi.org/10.1002/ppp.2090, 2020.

Crump, S. E., Miller, G. H., and Anderson, R. S.: Interpreting exposure ages from ice-cored moraines: a Neoglacial case study on Baffin Island, Arctic Canada, J Quaternary Sci, 32, 1049–1062, https://doi.org/10.1002/jqs.2979, 2017.

Frauenfelder, R. and Kääb, A.: Towards a palaeoclimatic model of rock-glacier formation in the Swiss Alps, Ann Glaciol, 31, 281–286, https://doi.org/10.3189/172756400781820264, 2000.

Müller, J., Vieli, A., and Gärtner-Roer, I.: Rock glaciers on the run – understanding rock glacier landform evolution and recent changes from numerical flow modeling, The Cryosphere, 10, 2865–2886, https://doi.org/10.5194/tc-10-2865-2016, 2016.

Steinemann, O., Reitner, J. M., Ivy-Ochs, S., Christl, M., and Synal, H.-A.: Tracking rockglacier activity in the Eastern Alps from the Lateglacial to the early Holocene, Quat Sci Rev, 241, https://doi.org/10.1016/j.quascirev.2020.106424, 2020.

---

## Referee Report (RR1)

Review by Leif Anderson

The authors done an admirable job improving the manuscript and addressed the reviewers' comments and suggestions in a comprehensive manner. I am happy to recommend this substantial contribution for publication after a few minor presentation issues are addressed.

The figures are very clear and communicate the results well. The manuscript will be even more well received if a bit more time is spent on the text. Below I suggest in some places splitting long paragraphs in two and synthesizing and simplifying text in some places.

It was a pleasure reviewing your research. Well done!

Minor suggestions

Line 32-50 consider breaking this into two paragraphs for easier reading and communication of the main points here.

Line 33. suggest a rewrite from "as diagnostic for" to "to be diagnostic of"

Line 37. consider 'in a delayed fashion' instead of 'attenduated and delayed'

Line 66 perhaps consider reword of 'Holocene-long development' to 'development through the entire Holocene'

Line 98 This sentence would be more effective if the list came second and the use of the methods came first.

Line 118-19. Very slick!

196. Consider breaking this paragraph up into a couple.

209. 'evidence' does not seem to be the right word here. Maybe 'to confirm the presence of a shear layer'? Also a citation to other work where a shear layers are present would be good here.

Figure 2. I think a figure title is needed here. Plus unbolded text I believe for figure captions.

Figure 3. Very nice figure.

Figure 4. the utm distances on the x and y are hard to read as are the contour labels. I would suggest removing the utm labels as there is a scale bar in the figure already. Photo views are really well done here.

359. 'looking at' is a bit casual, perhaps 'interpreting' ?

Table 1. If there is space adding a column for the unit number would be helpful. But not necessary.

390. Consider a more technical term for 'matched' here

Table 4 is very helpful for the reader.

548. All the info is good here but it I wonder if the take homes can be simplified a bit here. It is not totally clear after going through the paragraph.

685. Need an 'a' between 'as' and 'stress'

Line 757. Anderson, L.S. should be inserted as the second author here.

---

## Author Response (AR2)

Dear Leif Anderson
Dear Andreas Vieli, editor of The Cryosphere

We thank you for the very positive reception of our substantially revised manuscript. We hope we have addressed the few minor corrections raised by the referee and the editor by breaking up paragraphs or longer sentences and by ensuring consistent wording and spelling (e.g. "middle Holocene" instead of "mid Holocene"). Please see our point-by-point responses below.

We thank you very much again for your careful review of our manuscript.
Dominik Amschwand, Susan Ivy-Ochs and Olivia Steinemann, on behalf of all the co-authors

**List of minor suggestions by Referee Leif Anderson**

Line 32-50. Consider breaking this into two paragraphs for easier reading and communication of the main points here.
We broke the paragraph at "In the literature, …" and at "Another concept is the synchronous…".

Line 33. Suggest a rewrite from "as diagnostic for" to "to be diagnostic of".
We modified the text accordingly.

Line 37. Consider 'in a delayed fashion' instead of 'attenuated and delayed'.
We modified the text accordingly.

Line 66. Perhaps consider reword of 'Holocene-long development' to 'development through the entire Holocene'.
We modified the text accordingly.

Line 98. This sentence would be more effective if the list came second and the use of the methods came first.
We modified the text accordingly.

Line 118-19. Very slick!
Thank you!

Line 196. Consider breaking this paragraph up into a couple.
We broke the paragraph at "Over-steepened terrain steps, …" to separate general rock-glacier stratigraphy from Bleis Marscha observations and model implementation.

Line 209. 'evidence' does not seem to be the right word here. Maybe 'to confirm the presence of a shear layer'? Also a citation to other work where a shear layers are present would be good here.
We modified the text accordingly and added Arenson et al. (2002).

Figure 2. I think a figure title is needed here. Plus unbolded text I believe for figure captions.
We modified all figure captions accordingly. Yes, text is unbolded except for the "Figure X."

Figure 3. Very nice figure. Thank you!
Figure 4. the UTM distances on the x and y are hard to read as are the contour labels. I would suggest removing the UTM labels as there is a scale bar in the figure already. Photo views are really well done here.
We removed all UTM labels except of two for each axis for reference, and those in a slightly larger font size.

Line 359. 'looking at' is a bit casual, perhaps 'interpreting'?
We modified the text accordingly.

Table 1. If there is space adding a column for the unit number would be helpful. But not necessary.
No, unfortunately, there is not enough space. We hope that the units are gleaned from Figs. 7 and 11.

Line 390. Consider a more technical term for 'matched' here
We changed the wording to "correlated" and rewrote the text as follows:
> "By correlating two orthophotos from late summer 2003 and 2012, we obtain a noise-filtered horizontal surface creep-rate field for the Bleis Marscha rock glacier and its immediate surroundings."

Table 4 is very helpful for the reader.
Thank you!

Line 548. All the info is good here but it I wonder if the take homes can be simplified a bit here. It is not totally clear after going through the paragraph.
We added a sentence "It is within this framework that we interpret the exposure ages" to clarify the purpose and reformulated the statement more concretely:
> The cumulative effect of material loss and small under-exposure from boulder instabilities and rotation added up over centuries–millennia likely is the primary contributor to the 'geologic scatter'.

Line 685. Need an 'a' between 'as' and 'stress'
We modified the text accordingly.

Line 757. Anderson, L.S. should be inserted as the second author here
We apologize for the omission. We completed the citation.

**Minor comments by editor Andreas Vieli**

Line 46-47: Just a suggestion, liquid water input (besides temp and snow) seems also important for seasonal variations, maybe integrate this here (see Cicoira et al 2019, Earth Planet Sc Lett).
Yes, thank you for this suggestion. We include the citation and modified the text accordingly.

Line 99: you could use the abbreviation FE here as you already introduced it above on line 70.
We modified the text accordingly.

Line 121: average error of 1-3m. is this the vertical or horizontal position error?
This is the position error for all three dimensions (Kartenblatt 1236, Ausgabe 2017). We modified the text accordingly.

Line 195: '… in two dimensions along a central flowline with….'
We modified the text accordingly with this clarifying apposition.

Line 282: I would delete 'apparently'
No, we would like to emphasize that the lower part only looks inactive, but is actually slowly moving as revealed by the kinematic analysis (comparison with older aerial image from 1988, cf. caption to Fig. 8).

Fig. 4 caption: add age unit of exposure ages: '…exposure ages in ka BP'.

We added the units (ka, not ka BP), modified the caption accordingly and clarified that the uncertainties are 1σ external errors. 'BP' is only used for calibrated radiocarbon dates.

Fig. 6 caption: similar add a sentence what the numbers are on the foto: exposure ages in ka BP.
We modified the caption accordingly.

Fig. 7 caption: add age unit of exposure ages: '…exposure ages in ka BP' (yes it is on the right axes already, but the labels in the figures have no units!).
We modified the caption accordingly.

Line 523ff: relatively small wobble is also the case of relatively fast flowing rockglaciers in the Mattertal (see Wirz et al 2016 Earth Surf. Dynam., 4, 103-123, 2016 www.earth-surf-dynam.net/4/103/2016/ (based on continuous inclinometer data and GPS)
Thank you for this literature suggestion! We added this citation and modified the text accordingly.

Line 569: 'most upstream'
We modified the text accordingly.

Line 663: style, '…to earlier research…'
We modified the text accordingly.

Acknowledgments: please also acknowledge the input by the two referees.
We sincerely thank both reviewers and the editor for their work. Of course, we added:

An earlier draft greatly benefitted from the critical reviews by Jakob Heyman and Leif Anderson. We appreciate as well the numerous insightful suggestions given by handling editor Andreas Vieli.